# CONSTRAINED POSTERIOR SAMPLING: TIME SERIES GENERATION WITH HARD CONSTRAINTS

## ABSTRACT

Generating realistic time series samples is crucial for stress-testing models and protecting user privacy by using synthetic data. In engineering and safety-critical applications, these samples must meet certain hard constraints that are domain-specific or naturally imposed by physics or nature. Consider, for example, generating electricity demand patterns with constraints on peak demand times. This can be used to stress-test the functioning of power grids during adverse weather conditions. Existing approaches for generating constrained time series are either not scalable or degrade sample quality. To address these challenges, we introduce Constrained Posterior Sampling (CPS), a diffusion-based sampling algorithm that aims to project the posterior mean estimate into the constraint set after each denoising update. Notably, CPS scales to a large number of constraints ($\sim 100$) without requiring additional training. We provide theoretical justifications highlighting the impact of our projection step on sampling. Empirically, CPS outperforms state-of-the-art methods in sample quality and similarity to real time series by around 10% and 42%, respectively, on real-world stocks, traffic, and air quality datasets.

## 1 INTRODUCTION

Synthesizing realistic time series samples can aid in "what-if" scenario analysis, stress-testing machine learning (ML) models (Rizzato et al., 2022; Gowal et al., 2021), anonymizing private user data (Yoon et al., 2020), etc. Current approaches for time series generation use state-of-the-art (SOTA) generative models, such as Generative Adversarial Networks (GANs) (Yoon et al., 2019; Donahue et al., 2018) and Diffusion Models (DMs) (Tashiro et al., 2021; Alcaraz & Strodthoff, 2023; Narasimhan et al., 2024), to generate high-fidelity time series samples.

However, generating realistic and high-fidelity time series samples requires strict adherence to various domain-specific constraints. For example, consider generating the daily Open-high-low-close (OHLC) chart for the stock price of an S&P 500 company. The generated time series samples should have opening and closing stock prices bounded by the high and low values. Similarly, consider generating stock price time series with a user-specified measure of volatility to stress-test trading strategies. If the generated samples do not have the exact volatility, the stress testing results might not be accurate.

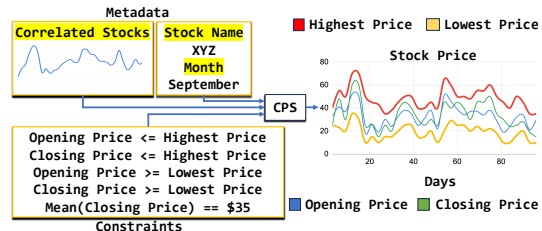

Figure 1: **Our Proposed Constrained Posterior Sampling (CPS) Approach.** CPS is a novel diffusion-based sampling approach to generate time series samples that adhere to hard constraints. Here, we show an example of generating the daily stock price time series, where CPS ensures that the generated stock prices adhere to natural constraints such as the bounds on the opening and closing prices of the stock.

On a more general note, the advent of large-scale generative models for language and vision, like GPT-4 (Bubeck et al., 2023) and Stable Diffusion (Podell et al., 2023), has increased the focus on constraining the outputs from these models, owing to usefulness and privacy reasons. Note that we cannot clearly define the notion of a constraint set in these domains. For example, verifying if the image of a hand has 6 fingers is practically hard, as all deep-learned perception models for this task have associated prediction errors. However, our key insight is that we can describe a time series through statistical features computed using well-defined

functions. These features can be imposed as constraints, and we can accurately verify the constraint satisfaction. Hence, the time series domain allows for the development of a new class of constrained generation algorithms. We first outline the qualities of an ideal constrained time series generator.

1. **Training-free approach to include multiple constraints:** Training the generative model for a specific constraint, as in the case of Loss-DiffTime (Coletta et al., 2024), is not scalable. A model trained to generate samples with specified mean constraints cannot adapt to argmax constraints.
2. **Independence from external realism enforcers:** Generally, prior works involve a projection step to a feasible set defined by a set of constraints, which often destroys the sample quality. To address this, prior approaches (Coletta et al., 2024) rely on external models to enforce realism, in addition to the generative model, resulting in additional training and complex sampling procedures.
3. **Hyperparameter-free approach to constrained generation:** The choice of guidance weights in guidance-based approaches with DMs significantly affects the sample quality. Optimizing for guidance weights becomes combinatorially hard while dealing with hundreds of constraints.

Given the following requirements, we propose **Constrained Posterior Sampling (CPS)**, a novel sampling procedure for diffusion-based generative models (check Fig. 1). CPS introduces a projection step that aims to project the posterior mean estimate into the constraint set after each diffusion denoising update. We rely on off-the-shelf optimization routines, thereby providing a training and hyperparameter-free approach to include multiple constraints. Additionally, CPS does not require external models to enforce realism, as the key intuition in our approach is that the subsequent denoising steps rectify the adverse effects of the projection steps toward sample quality. To this end, our contributions in this paper are:

1. We present Constrained Posterior Sampling ((CPS), Fig. 1), a scalable diffusion sampling process that generates realistic time series samples that belong to a constraint set. Without any additional training, CPS can handle a large number of constraints without sacrificing sample quality (Fig. 3).
2. We provide a detailed theoretical analysis of the effect of modifying the traditional diffusion sampling process with CPS. Additionally, we perform convergence analysis for well-studied settings, such as convex constraint sets and Gaussian prior data distribution, to draw useful insights for the practical implementation of CPS.
3. Through extensive experiments on six diverse real-world and simulated datasets spanning finance, traffic, and environmental monitoring, we demonstrate that CPS outperforms state-of-the-art approaches (SOTA) on sample quality, similarity, and constraint adherence metrics (check Fig. 2).

## 2 PRELIMINARIES

**Notations:** We denote a time series sample by $x \in \mathbb{R}^{K \times L}$. Here, $K$ and $L$ refer to the number of channels and the horizon, respectively. A dataset is defined as $\mathcal{D} = \{x^1, \ldots, x^{N_D}\}$, where the superscript $i \in [1, \ldots N_D]$ refers to the sample number, and $N_D$ is the total number of samples in the dataset. $P_{\text{data}}$ denotes the real time series data distribution. $x^i$ is the realization of the random vector $X^i$, where $X^1, \ldots X^{N_D} \sim P_{\text{data}}$. The Probability Density Function (PDF) associated with $P_{\text{data}}$ is represented by $p_{\text{data}} : \mathbb{R}^{K \times L} \to \mathbb{R}$, where $\int p_{\text{data}}(x)dx = 1$. Here, $\int$ refers to the integration operator over $\mathbb{R}^{K \times L}$. The notation $\mathcal{N}(\mu, \Sigma)$ refers to the Gaussian distribution with mean $\mu$ and covariance matrix $\Sigma$. Similarly, $\mathcal{U}(a, b)$ indicates the uniform distribution with non-zero density from $a$ to $b$. $\| \cdot \|_2$ is overloaded and indicates the $l_2$ norm in the case of a vector and the spectral norm in the case of a matrix. We denote the constraint set $\mathcal{C}$ as $\mathcal{C} = \mathcal{C}_1 \bigcap \mathcal{C}_2, \ldots, \bigcap \mathcal{C}_{N_C}$, where $N_C$ is the total number of constraints and $\bigcap$ denotes intersection. Here, $\mathcal{C}_i = \{x \mid f_{c_i}(x) \le 0\}$

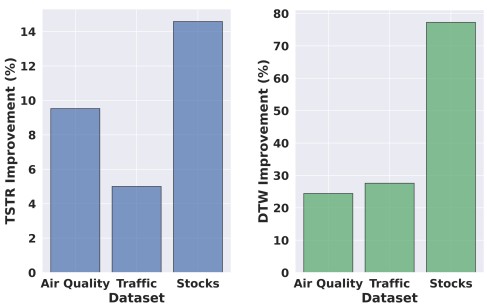

Figure 2: **CPS outperforms existing approaches on real-world datasets.** Dynamic Time Warping (DTW) measures the similarity between the real and the generated time series. The Train on Synthetic and Test on Real (TSTR) evaluates a task model on real test data when the model was trained on synthetic data. Improved TSTR indicates high generated sample quality. CPS provides 42% and 10% improvements for DTW and TSTR, respectively, over SOTA methods.

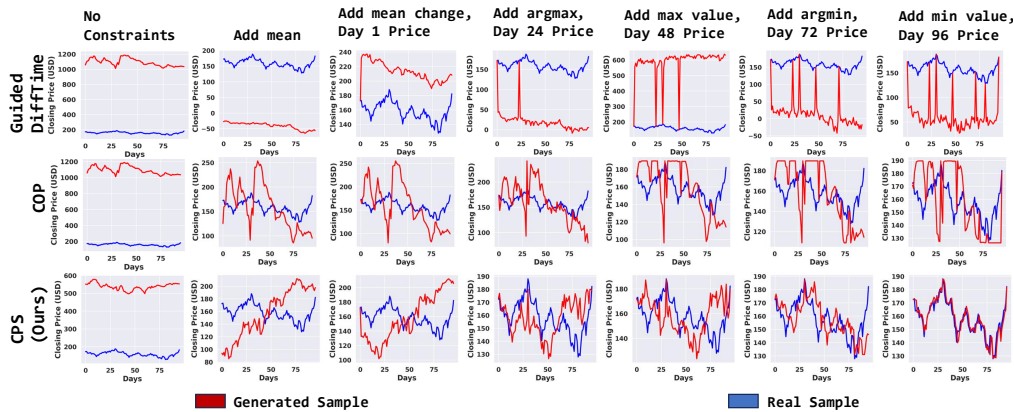

Figure 3: **CPS tracks the real data samples as the number of constraints increases.** Increasing the number of constraints reduces the size of the constraint set, and an ideal approach should effectively generate samples that resemble the real time series samples that belong to the constraint set. Here, we show a qualitative example from the Stocks dataset. Observe that CPS accurately tracks the real sample that concurs with the specified constraints while other approaches suffer.

with $f_{c_i} : \mathbb{R}^{K \times L} \to \mathbb{R} \; \forall \; c_i \in [1, \dots, N_C]$. $\lambda_{\max}(M)$ and $\lambda_{\min}(M)$ refer to the largest and the smallest eigen values of the square matrix $M$. The rank of the matrix $M$ is indicated by $rank(M)$.

**Example:** The stocks dataset has 6 channels ($K = 6$) with 96 timestamps in each channel ($L = 96$). The first 4 channels represent the opening price ($o$), the highest price ($h$), the lowest price ($l$), and the closing price ($c$), and each timestamp represents a day. The OHLC constraint, *i.e.,* the opening and closing prices should lie between the highest and the lowest prices, is given by $o - h \leq 0, c - h \leq 0$, $l - o \leq 0$, and $l - c \leq 0$. Additionally, a mean equality constraint on the closing price is expressed as $\frac{1}{L} \left( \sum_{u=1}^{L} c(u) \right) - \mu_c \leq 0$ and $\mu_c - \frac{1}{L} \left( \sum_{u=1}^{L} c(u) \right) \leq 0$, where $\mu_c$ is the required mean.

## 2.1 BACKGROUND AND RELATED WORK

GANs (Goodfellow et al., 2014) have been the popular choice for time series generation (Yoon et al., 2019; Donahue et al., 2018; Srinivasan & Knottenbelt, 2022; Ni et al., 2021). Recently, DMs have dominated the landscape of image, video, and audio generation (Rombach et al., 2022; Ho et al., 2022; Kong et al., 2020). Denoising DMs (Ho et al., 2020; Dhariwal & Nichol, 2021) generate samples by learning to gradually denoise clean data, sampled from the data distribution $P_{\text{data}}$, corrupted with Gaussian noise. Denoising Diffusion Probabilistic Models (DDPMs) (Ho et al., 2020) define a Markovian forward noising process, where the clean data sample $x$, referred to as $z_0$, is transformed into $z_T$ with iterative Gaussian corruption for $T$ noising steps, such that $z_T \sim \mathcal{N}(\mathbf{0}, \mathbf{I})$. With abuse of notation, $\mathbf{0}$ represents zero mean, and $\mathbf{I}$ represents the identity covariance. The forward process introduces $T$ conditional Gaussian distributions with fixed covariance matrices governed by the diffusion coefficients $\bar{\alpha}_0, \dots, \bar{\alpha}_T$, where $\bar{\alpha}_t \in [0, 1], \bar{\alpha}_0 = 1, \bar{\alpha}_T = 0, \bar{\alpha}_{t-1} > \bar{\alpha}_t \; \forall \; t \in [1, T]$. Formally, $q_t(z_t \mid z_0)$ is the PDF of the conditional Gaussian distribution at the forward step $t$ with mean $\sqrt{\bar{\alpha}_t} z_0$ and covariance matrix $(1 - \bar{\alpha}_t)\mathbf{I}$. The PDF associated with the marginal distribution at $t = 0$ is given by $q_0 = p_{\text{data}}$.

The sample generation or the reverse process is also Markovian, where we autoregressively sample from $T$ Gaussian distributions with fixed covariance matrices, indicated by PDFs $p_{\theta,t}(z_{t-1} \mid z_t) \; \forall \; t \in [1, T]$, to get from $z_T$ to $z_0$, where $z_T \sim \mathcal{N}(\mathbf{0}, \mathbf{I})$. The means of $p_{\theta,t}(z_{t-1} \mid z_t)$ are learned using neural networks. DDPMs are trained to maximize the log-likelihood of observing the clean data, *i.e.,* $\log p_\theta(z_0)$, where $p_\theta(z_0) = \int p_\theta(z_{0:T}) dz_{1:T}$. The joint PDF $p_\theta(z_{0:T})$ can be factorized as $p(z_T) \prod_{t=1}^{T} p_{\theta,t}(z_{t-1} \mid z_t)$, due to the Markovian nature of the reverse process, with $p(z_T) = \mathcal{N}(\mathbf{0}, \mathbf{I})$. With successive reparametrizations, the training objective can be simplified into the following denoising objective:

$$\mathbb{E}_{z_0 \sim P_{\text{data}}, \epsilon \sim \mathcal{N}(\mathbf{0}, \mathbf{I}), t \sim \mathcal{U}(1,T)} [\|\epsilon - \epsilon_\theta(z_t, t)\|_2^2], \tag{1}$$

where $\epsilon_\theta(z_t, t)$ is trained to estimate the noise $\epsilon$ from $z_t$, and $z_t = \sqrt{\bar{\alpha}_t}z_0 + \sqrt{1 - \bar{\alpha}_t}\epsilon$, with $t$ ranging from $1$ to $T$. Denoising Diffusion Implicit Models (DDIMs) Song et al. (2022) propose a non-Markovian forward process and, accordingly, a novel mechanism for sample generation given by

$$z_{t-1} = \sqrt{\bar{\alpha}_{t-1}}\hat{z}_0(z_t; \epsilon_\theta) + \sqrt{1 - \bar{\alpha}_{t-1} - \sigma_t^2}\epsilon_\theta(z_t, t) + \sigma_t\epsilon. \tag{2}$$

Here, $\hat{z}_0(z_t; \epsilon_\theta) = \frac{z_t - \sqrt{1 - \bar{\alpha}_t}\epsilon_\theta(z_t, t)}{\sqrt{\bar{\alpha}_t}}$ is the posterior mean estimate, and $\sigma_t$ is a control parameter that dictates determinism in the sampling process. Song et al. (2022) show that Eq. 2 corresponds to the following reverse process:

$$p_{\theta,t}(z_{t-1} \mid z_t) = \begin{cases} p_{\theta,\text{init}}(z_0 \mid \hat{z}_0(z_1; \epsilon_\theta)) & \text{if } t = 1, \\ q_{\sigma,t}(z_{t-1} \mid z_t, \hat{z}_0(z_t; \epsilon_\theta)) & \text{otherwise,} \end{cases} \tag{3}$$

where $q_{\sigma,t}(z_{t-1} \mid z_t, \hat{z}_0(z_t; \epsilon_\theta))$ represents the PDF of the Gaussian distribution with mean $\sqrt{\bar{\alpha}_{t-1}}\hat{z}_0(z_t; \epsilon_\theta) + \sqrt{1 - \bar{\alpha}_{t-1} - \sigma_t^2}\epsilon_\theta(z_t, t)$ and covariance matrix $\sigma_t^2\mathbf{I}$. Similarly, $p_{\theta,\text{init}}(z_0 \mid \hat{z}_0(z_1; \epsilon_\theta))$ is the PDF of the Gaussian distribution with mean $\hat{z}_0(z_1; \epsilon_\theta)$ and covariance matrix $\sigma_1^2\mathbf{I}$. This reverse sampling process can be viewed as obtaining the posterior mean estimate $\hat{z}_0(z_t; \epsilon_\theta)$ and transforming it to the noise level for step $t - 1$. CPS builds on Eq. 2.

Sampling from a probability distribution supported on a constraint set is essential in various engineering fields, including material science and robotics. To address this, Frerix et al. (2020) introduce Variational AutoEncoders (VAEs) with additional trainable layers to enforce linear inequality constraints. Liu et al. (2023) and Fishman et al. (2023a;b) modify the forward noising process and the diffusion model training to satisfy the required constraints. Since these methods require additional training, they are less scalable when adapting to new constraints. To overcome this limitation, Christopher et al. (2024) propose Projected Diffusion Models (PDMs), a training-free approach that projects the intermediate noisy latents of the reverse process ($z_T, \ldots, z_0$) into the constraint set. Though training-free, this approach can impact sample quality and diversity, as detailed in Appendix G.

In the time series domain, Wang et al. (2024a;b) focus on generating constrained counterfactual explanations for classification and forecasting by perturbing selected time stamps of a synthesized seed sample. These approaches do not provide any mechanism to induce realism other than staying near the seed sample. Recently, Coletta et al. (2024) proposed three approaches - **Loss-DiffTime**, a training-based approach where constraint-specific samples are generated with constraints as conditional input to the generator, **Guided DiffTime**, which uses guidance gradients from differentiable constraint functions to guide the sample generation toward a constraint set, and **Constrained Optimization Problem** (COP), which projects a seed sample to the constraint set while using the critic function from any Wasserstein GAN (Arjovsky et al., 2017) as a realism enforcer. Loss-DiffTime is not scalable to new constraints without retraining, while Guided DiffTime and other guidance-based approaches like Diffusion-TS (Yuan & Qiao, 2024) do not guarantee constraint satisfaction even for convex constraint sets. We compare CPS against these approaches on many real-world datasets and highlight our advantages.

Finally, constrained generation can also be viewed as controlling the outputs of a generative model, which occurs in multiple formulations in the image domain, such as solving inverse problems (Rout et al., 2023a;b; 2024a; Chung et al., 2024), personalization (Rout et al., 2024b; Ruiz et al., 2022), text-to-image generation (Rombach et al., 2022; Ramesh et al., 2022), and text-based image editing (Kawar et al., 2023; Choi et al., 2023). We note that CPS (Sec. 3) can be viewed as a constraint satisfaction (through projection) approach for time series, in the same spirit as gradient-based image personalization through diffusions (Rout et al., 2024b). However, these works do not impose hard constraints, as described in the case of OHLC charts in Sec. 1. We refer the reader to Appendix F for a detailed discussion of the prior works. Formally, the constrained time series generation problem is defined as follows:

**Problem Setup.** Consider a dataset $\mathcal{D} = \{x^i\}_{i=1}^{N_\text{D}}$, where $N_\text{D}$ denotes the number of samples, $x^i \sim P_\text{data}$ with the density function $p_\text{data}$ and $x^i \in \mathbb{R}^{K \times L}$. The goal is to generate $x^\text{gen} \sim P_\text{data}$ such that $x^\text{gen}$ belongs to the constraint set $\mathcal{C} = \mathcal{C}_1 \bigcap \mathcal{C}_2, \ldots, \bigcap \mathcal{C}_{N_\text{C}}$, where $N_\text{C}$ denotes the number of constraints. Here, $\mathcal{C}_i = \{x \mid f_{c_i}(x) \leq 0\}$ with $f_{c_i} : \mathbb{R}^{K \times L} \to \mathbb{R}$. To put it more succinctly,

$$x^\text{gen} := \underset{x}{\arg\min} -\log p_\text{data}(x) \text{ s.t. } f_{c_i}(x) \leq 0, \ \forall \ c_i \in [1, N_\text{C}], \tag{4}$$

where the objective is to find a maximum likelihood sample in the constraint set.

## 3 CONSTRAINED POSTERIOR SAMPLING

To generate realistic samples with high likelihood, our approach assumes the availability of a pre-trained diffusion model trained on the dataset $\mathcal{D}$. Given the diffusion model $\epsilon_\theta$, we propose Constrained Posterior Sampling (CPS, check Fig. 4) to restrict the domain of a generated sample without sacrificing sample quality. Described in Algorithm 1, CPS effectively guides the diffusion denoising process towards the constraint set.

We follow the typical DDIM inference procedure. Starting with a sample from the standard normal distribution $\mathcal{N}(\mathbf{0}, \mathbf{I})$ (line 1), we perform sequential denoising (lines 2 to 10). Line 3 refers to the forward pass through the denoiser to obtain the noise estimate $\epsilon_\theta(z_t, t)$. After every denoising step, we obtain the posterior mean estimate $\hat{z}_0(z_t; \epsilon_\theta)$ (line 4). We then project this estimate towards the constraint set $\mathcal{C}$ to obtain the projected posterior mean estimate $\hat{z}_{0,\mathrm{pr}}(z_t; \epsilon_\theta)$ (line 5). Later, we perform a DDIM reverse sampling step with $\hat{z}_{0,\mathrm{pr}}(z_t; \epsilon_\theta)$ and $\epsilon_\theta(z_t, t)$ to obtain $z_{t-1}$ (lines 7-9).

The projection step in line 5 solves an optimization problem with the objective function $\frac{1}{2}(\|z - \hat{z}_0(z_t; \epsilon_\theta)\|_2^2 + \gamma(t)\Pi(z))$. The first term of the objective function ensures that $\hat{z}_{0,\mathrm{pr}}(z_t; \epsilon_\theta)$ is close to $\hat{z}_0(z_t; \epsilon_\theta)$, thereby ensuring that $z_{t-1}$ is not heavily perturbed for the denoiser to perform poorly. We define the constraint violation function $\Pi : \mathbb{R}^{K \times L} \to \mathbb{R}$ as $\Pi(z) = \sum_{i=1}^{N_C} \max(0, f_{c_i}(z))$, such that $\Pi(z) = 0$ if $z \in \mathcal{C}$ and $\Pi(z) > 0$ otherwise. For the denoising step $t$, the constraint violation function is scaled by a time-varying penalty coefficient $\gamma(t)$. Our key intuition is to design $\gamma(t)$ as a strictly decreasing function of $t$ that takes small values for the initial denoising steps ($t$ close to $T$) and tends to $\infty$ for the final de-

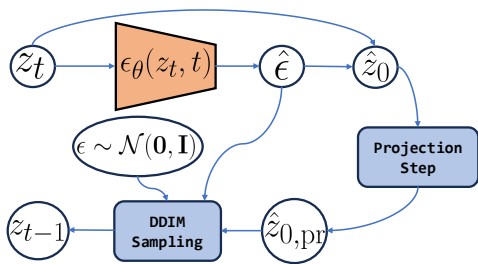

Figure 4: **Our proposed Constrained Posterior Sampling approach.** We show the graphical model for one step of denoising in CPS, as outlined in Algorithm 1.

noising steps. This ensures that the constraint satisfaction is not heavily enforced during the initial denoising steps when the signal-to-noise ratio in $z_t$ is very low. Given the requirements for the penalty coefficient, we choose $\gamma(t) = e^{1/(1-\bar{\alpha}_{t-1})}$ such that $\gamma(t)$ is close to 0 for the initial denoising steps ($\gamma(T) \simeq e$) and $\gamma(t) \to \infty$ for $t = 1$. Note that our choice of $\gamma(t)$ ensures that $\gamma(t)$ is strictly decreasing with respect to $t$ since $\bar{\alpha}_t$ strictly decreases with $t$.

Observe that CPS is the DDIM sampling process with one change. We replace the posterior mean estimate $\hat{z}_0(z_t; \epsilon_\theta)$ with the projected posterior mean estimate $\hat{z}_{0,\mathrm{pr}}(z_t; \epsilon_\theta)$. Additionally, CPS can be viewed similarly to the penalty-based methods to solve a constrained optimization problem. With each progressing denoising update, the penalty coefficient increases, thereby pushing the posterior mean estimate towards the constraint set.

We do not add noise after the final denoising step ($\sigma_1 = 0$). This ensures that the efforts of the final projection step towards constraint satisfaction are not compromised by additional noise. For convex constraint sets with assumptions on the convexity of the constraint definition functions $f_{c_i}$, we note that the projection step is an unconstrained minimization of a convex function with the optimal constraint violation value being 0 if $\gamma(1)$ tends to $\infty$. With a suitable choice of solvers (Diamond & Boyd, 2016), the optimal solution can be obtained for these cases, thereby ensuring constraint satisfaction ($\Pi(\hat{z}_{0,\mathrm{pr}}(z_1; \epsilon_\theta)) = 0$) when $\gamma(1)$ tends to $\infty$.

---

**Algorithm 1** Constrained Posterior Sampling

**Input:** Diffusion model $\epsilon_\theta$ with $T$ denoising steps, Noise coefficients $\{\bar{\alpha}_0, \ldots, \bar{\alpha}_T\}$, DDIM control parameters $\{\sigma_1, \ldots, \sigma_T\}$, Constraint violation function $\Pi$, Penalty coefficients $\{\gamma(1), \ldots, \gamma(T)\}$.

**Output:** Synthesized time series sample $x^{\mathrm{gen}}$.

1: Initialize $z_T \sim \mathcal{N}(\mathbf{0}, \mathbf{I})$
2: **for** $t$ from $T$ to 1 **do**
3:      Obtain $\hat{\epsilon} = \epsilon_\theta(z_t, t)$      ▷ Noise Estimation
4:      $\hat{z}_0(z_t; \epsilon_\theta) = \frac{z_t - \sqrt{1 - \bar{\alpha}_t}\hat{\epsilon}}{\sqrt{\bar{\alpha}_t}}$      ▷ Predicted $z_0$
5:      $\hat{z}_{0,\mathrm{pr}}(z_t; \epsilon_\theta) = \arg\min_z \frac{1}{2} \left\{ \begin{array}{l} \|z - \hat{z}_0(z_t; \epsilon_\theta)\|_2^2 \\ + \gamma(t)\Pi(z) \end{array} \right\}$
6:      ▷ Projection Step
7:      $z_{t-1} = \sqrt{\bar{\alpha}_{t-1}}\hat{z}_{0,\mathrm{pr}}(z_t; \epsilon_\theta) + \sqrt{1 - \bar{\alpha}_{t-1} - \sigma_t^2}\hat{\epsilon}$
8:      $\epsilon \sim \mathcal{N}(\mathbf{0}, \mathbf{I})$
9:      $z_{t-1} = z_{t-1} + \sigma_t \epsilon$      ▷ DDIM Steps
10: **end for**
11: $x^{\mathrm{gen}} = z_0$
12: **return** $x^{\mathrm{gen}}$

---

Note that CPS satisfies the key requirements of an ideal constrained generation approach. CPS can handle multiple constraints without any training requirements. Further, CPS does not require additional critics to enforce realism, as our key intuition is that the successive denoising steps address the adverse effects of the projection step. Finally, CPS is hyperparameter-free as off-the-shelf solvers can perform the unconstrained optimization step in line 5. Our key observation is that unlike heuristically setting the guidance weights (Coletta et al., 2024), we can choose the parameters of the solvers using principled approaches from the vast optimization literature (Nocedal & Wright, 1999).

### 3.1 THEORETICAL JUSTIFICATION

Now, we provide a detailed analysis of the effect of modifying the traditional DDIM sampling process with CPS. For ease of explanation, we consider $z \in \mathbb{R}^n$. We indicate the identity matrix in $\mathbb{R}^{n \times n}$ as $\mathbf{I}_n$. First, we describe the exact distribution from which the samples are generated. For this, we make the following assumption.

**Assumption 1.** *Let the constraint set be* $\mathcal{C} = \{z \mid f_{\mathcal{C}}(z) = 0\}$, *where* $f_{\mathcal{C}} : \mathbb{R}^n \to \mathbb{R}$ *and the penalty function* $\Pi(z) = \|f_{\mathcal{C}}(z)\|_2^2$ *has L-Lipschitz continuous gradients, i.e.,* $\|\nabla\Pi(u) - \nabla\Pi(v)\|_2 \leq L\|u - v\|_2 \, \forall \, u, v \in \mathbb{R}^n$.

**Theorem 1.** *Suppose Assumption 1 holds. Given a denoiser* $\epsilon_\theta : \mathbb{R}^n \to \mathbb{R}^n$ *for a diffusion process with noise coefficients* $\bar{\alpha}_0, \ldots, \bar{\alpha}_T$, *if* $\gamma(t) > 0 \, \forall \, t \in [1, T]$, *the denoising step in Algorithm 1 is equivalent to sampling from the following conditional distribution:*

$$p_{\theta,t}(z_{t-1} \mid z_t) = \begin{cases} p_{\theta,\mathrm{init}}(z_0 \mid \hat{z}_{0,\mathrm{pr}}(z_1; \epsilon_\theta)) & \text{if } t = 1, \\ q_{\sigma,t}(z_{t-1} \mid z_t, \hat{z}_{0,\mathrm{pr}}(z_t; \epsilon_\theta)) & \text{otherwise,} \end{cases} \quad (5)$$

*Here,* $p_{\theta,\mathrm{init}}(z_0 \mid \hat{z}_{0,\mathrm{pr}}(z_1; \epsilon_\theta))$ *indicates the PDF of* $\mathcal{N}\left(\hat{z}_{0,\mathrm{pr}}(z_1; \epsilon_\theta), \sigma_1^2 \mathbf{I}_n\right)$, *and* $q_{\sigma,t}(z_{t-1} \mid z_t, \hat{z}_{0,\mathrm{pr}}(z_t; \epsilon_\theta))$ *indicates the PDF of* $\mathcal{N}\left(\sqrt{\bar{\alpha}_{t-1}}\hat{z}_{0,\mathrm{pr}}(z_t; \epsilon_\theta) + \sqrt{1 - \bar{\alpha}_{t-1} - \sigma_t^2}\epsilon_\theta(z_t, t), \sigma_t^2 \mathbf{I}_n\right)$. $\sigma_1, \ldots, \sigma_T$ *denote the DDIM control parameters, and* $\gamma(t)$ *indicates the penalty coefficient for the denoising step* $t$ *in Algorithm 1.*

Intuitively, Algorithm 1 can be viewed as replacing $\hat{z}_0(z_t; \epsilon_\theta)$ with $\hat{z}_{0,\mathrm{pr}}(z_t; \epsilon_\theta)$ and following the DDIM sampling process. Therefore, the reverse process PDFs are obtained by replacing $\hat{z}_0(z_t; \epsilon_\theta)$ with $\hat{z}_{0,\mathrm{pr}}(z_t; \epsilon_\theta)$ in Eq. 3. More formally, under Assumption 1, the projection step (line 5) can be written as a series of gradient updates that transform $\hat{z}_0(z_t; \epsilon_\theta)$ to $\hat{z}_{0,\mathrm{pr}}(z_t; \epsilon_\theta)$. Having Lipschitz continuous gradients for $f_{\mathcal{C}}$ allows for fixed step sizes which can guarantee a reduction in the value of the objective function $\frac{1}{2}(\|z - \hat{z}_0(z_t; \epsilon_\theta)\|_2^2 + \gamma(t)\|f_{\mathcal{C}}(z)\|_2^2)$ with each gradient update. We refer the readers to Appendix A.1 for the detailed proof. Now, we investigate the convergence properties for Algorithm 1 under the following assumption.

**Assumption 2.** *The real data distribution is* $\mathcal{N}(\mu, \mathbf{I}_n)$, *where* $\mu \in \mathbb{R}^n$, *and the constraint set* $\mathcal{C}$ *is defined as* $\mathcal{C} = \{z \mid Az = y\}$ *with* $A \in \mathbb{R}^{m \times n}$ *such that* $rank(A) = n \leq m$. *Additionally, for the real data distribution* $\mathcal{N}(\mu, \mathbf{I}_n)$ *and the constraint set* $\mathcal{C} = \{z \mid Az = y\}$, *there exists a unique solution to Eq. 4, indicated by* $x^*$.

We note that Assumption 2 ensures the existence of a unique solution to the linear problem $Ax = y$. While there exist many efficient methods to solve such problems under this assumption, the focus of this paper is not on solving this problem efficiently. Instead, we use this well-studied problem as a framework to analyze the convergence properties of Algorithm 1, providing valuable insights for better practical performance.

**Theorem 2.** *Suppose Assumption 2 holds. For a diffusion process with noise coefficients* $\bar{\alpha}_0, \ldots, \bar{\alpha}_T$, *where* $\bar{\alpha}_0 = 1$, $\bar{\alpha}_T = 0$, $\bar{\alpha}_t \in [0, 1] \, \forall \, t \in [0, T]$, *if* $\bar{\alpha}_t < \bar{\alpha}_{t-1}$ *and* $\gamma(t) = \frac{2k(T-t+1)}{\lambda_{\min}(A^T A)}$ *with any design parameter* $k > 1$, *then in the limit as* $T \to \infty$, *Algorithm 1 returns* $x^{\mathrm{gen}}$ *such that:*

$$\|x^{\mathrm{gen}} - x^*\|_2 \leq \frac{\sqrt{\bar{\alpha}_1}}{k}\left(\|x^*\|_2 + \|\mu\|_2\right).$$

We refer the readers to Appendix A.2 for detailed proof. Briefly, the proof in Appendix A.2 indicates that the terminal error $\|x^{\mathrm{gen}} - x^*\|_2$ reduces to 0 as $T, k \to \infty$, thereby ensuring that Algorithm 1 converges to the true solution. From the proof, we observe that under Assumption 2, the convergence can be guaranteed when the penalty coefficient is set to very large values for the final denoising step. This is in accordance with our choice of penalty coefficients which assumes very large values for the final denoising step.

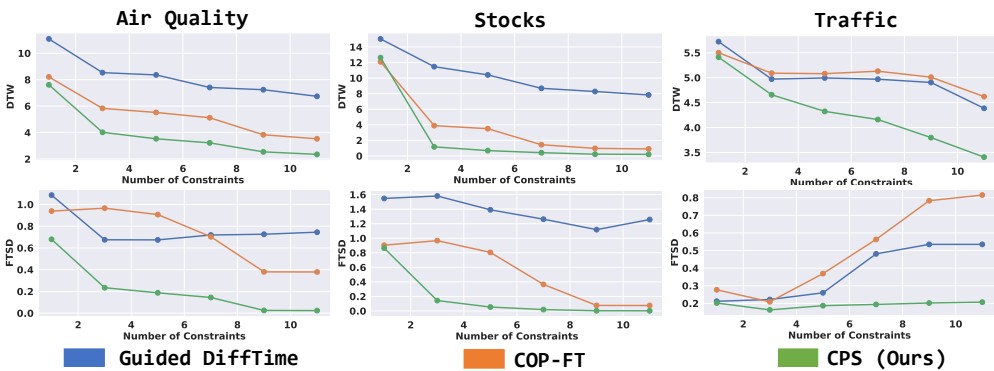

Figure 5: **CPS outperforms existing baselines with increasing number of constraints.** Note that constraints are the features extracted from real time series samples. We gradually increase the number of constraints imposed on the generative model. Observe that CPS achieves the lowest DTW score for any number of constraints while having the best sample quality, indicated by the lowest FTSD metric. This result is in accordance with the qualitative example shown in Fig. 3.

## 4 EXPERIMENTS

This section describes the experimental procedure, including the wide range of datasets and metrics used to evaluate CPS against the state-of-the-art constrained generation approaches.

**Datasets:** We use real-world datasets from different domains, such as Stocks (Yoon et al., 2019), Air Quality (Chen, 2019), and Traffic (Hogue, 2019). Specifically, we test the performance of CPS on both conditional and unconditional variants of these datasets. We also evaluate our approach on a simulated sinusoidal waveforms dataset to generate sinusoids with varying amplitudes, phases, and frequencies specified as constraints.

Our evaluation procedure is framed to test any approach for generating the maximum likelihood sample from a constraint set, such that the real time series samples from the constraint set were never seen during training. To achieve this, from every sample in the test dataset, we first extract an exhaustive set of features such that only one test sample exists per set of features. These features are considered constraints, which we impose on the generative model.

**Constraints:** We extract the following features to be used as constraints - *mean*, *mean consecutive change*, *argmax*, *argmin*, *value at argmax*, *value at argmin*, *values at timestamps 1, 24, 48, 72, & 96*. For the Stocks dataset, we additionally impose the natural OHLC constraint, *i.e.,* the opening and closing prices should be bounded by the highest and the lowest prices. Similarly, for the sinusoidal waveforms dataset, we extract the locations and values of the peaks and valleys and the trend from a peak to its adjacent valley. Note that these constraints can be written in the form $Ax \leq 0$. Projection to such constraint sets is easy and can be handled by numerous off-the-shelf solvers (Diamond & Boyd, 2016; Virtanen et al., 2020). This allows us to analyze the effect of the sampling process without worrying about the off-the-shelf solvers that influence the projection step. We provide a budget of 0.01 for constraint violation.

**Baselines:** We compare against the **Constrained Optimization Problem** (COP) approach (Coletta et al., 2024) and its fine-tuning variant, which is referred to as COP-FT. COP projects a random sample from the training dataset to the required set of constraints, whereas COP-FT projects a generated sample. Both these variants rely on a discriminator to enforce realism after perturbation. We also compare our approach against **Guided DiffTime** (Coletta et al., 2024), a guidance-based diffusion sampling approach. All baselines, except COP, utilize the same TIME WEAVER-CSDI denoiser backbone (Narasimhan et al., 2024) for fair comparison. Additionally, we compare CPS against **Projected Diffusion Models (PDM)** (Christopher et al., 2024), **Diffusion-TS** (Yuan & Qiao, 2024) (another guidance-based sampling approach), and **Loss-DiffTime**, the training-based approach from Coletta et al. (2024), in Appendix G.

**Metrics:** We evaluate the performance of CPS on three fronts - **sample quality, ability to track the test time series, and constraint violation**. For sample quality, we use the Frechet Time Series

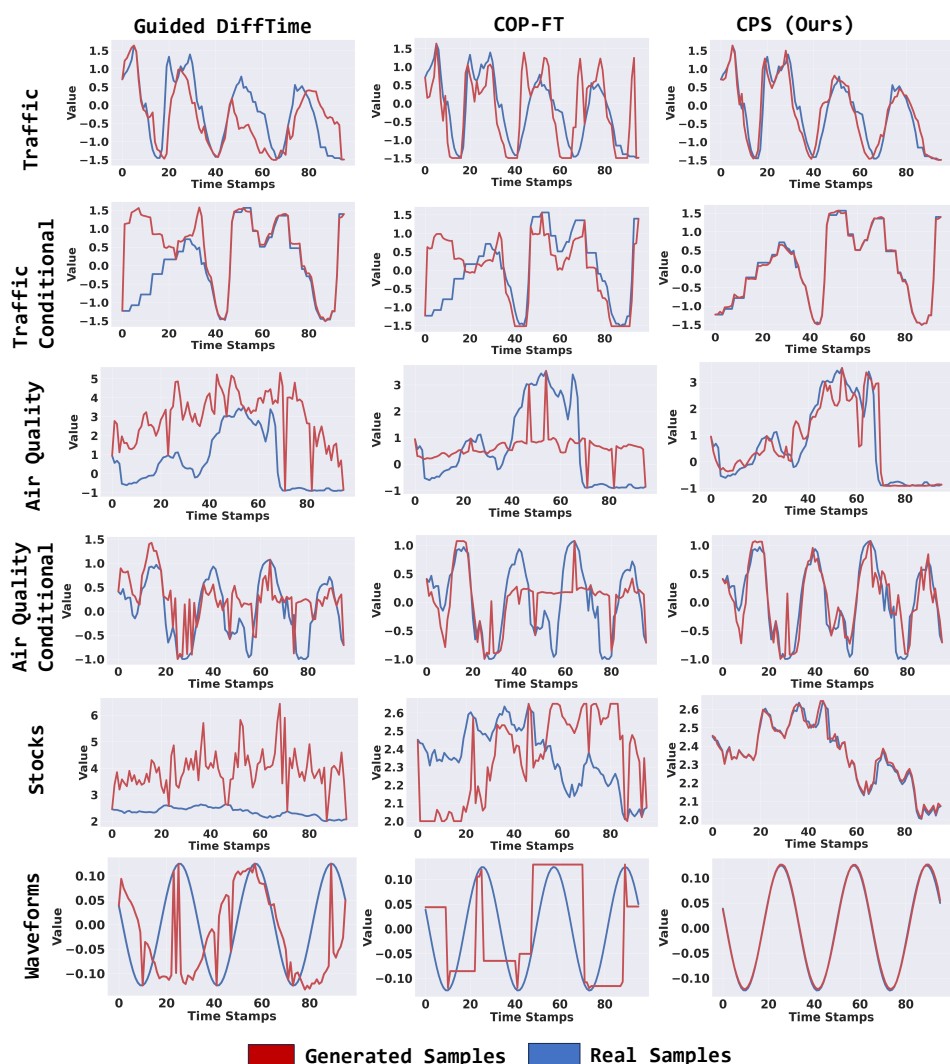

Figure 6: **CPS provides high-fidelity synthetic time series samples that match real time series data.** Here, we show a qualitative comparison between the baselines (Guided DiffTime and COP-FT) and CPS for six different experimental settings. As described in Sec. 4, the real test time series samples from which the constraints are extracted are shown in blue. Observe that across datasets, CPS generates high-fidelity samples that match the ground truth, while the baselines suffer to generate meaningful qualitative results.

Distance (**FTSD**) metric (Narasimhan et al., 2024; Paul et al., 2022) for the unconditional setting and the Joint Frechet Time Series Distance (**J-FTSD**) metric (Narasimhan et al., 2024) for the conditional setting. The FTSD metric is also referred to as **Context-FID** (Paul et al., 2022). For simplicity, we indicate both these metrics by Frechet Distance or FD. Additionally, we show the Train on Synthetic and Test on Real (**TSTR**) metric for sample quality. For TSTR, we choose random imputation as the task with 75% masking. We train the TimesNet model (Wu et al., 2023) for imputation on the synthesized training data, generated with constraints, and evaluate the trained model on the real test data for imputation performance. We report the mean squared error (MSE) on the real test set as the TSTR metric. Lower MSE indicates accurate modeling of the true data distribution.

From our evaluation procedure, note that we aim to enforce one test sample per set of constraints. Therefore, an ideal approach is expected to generate a sample that is similar to that single test sample. To estimate this, we report the Dynamic Time Warping (**DTW**) metric Müller (2007) and the Structural Similarity Index Measure (**SSIM**) metric Nilsson & Akenine-Möller (2020). Though SSIM is typically used for images, in essence, both these metrics capture the similarity between the

| METRIC | APPROACH | AIR QUALITY | AIR QUALITY (CONDITIONAL) | TRAFFIC | TRAFFIC (CONDITIONAL) | STOCKS | WAVEFORMS |
|---|---|---|---|---|---|---|---|
| FRECHET DISTANCE ($\downarrow$) | GUIDED DIFFTIME | 0.7457 | 3.1883 | 0.5351 | 0.5638 | 1.2575 | 0.3108 |
| | COP-FT | 0.3793 | 0.9931 | 0.8156 | 0.8135 | 0.0759 | 1.8419 |
| | COP | 0.2165 | 27.9425 | 0.9242 | 43.2472 | 0.0701 | 1.6627 |
| | CPS (OURS) | **0.0234** | **0.6039** | **0.2077** | **0.2812** | **0.0023** | **0.0029** |
| TSTR ($\downarrow$) | GUIDED DIFFTIME | 0.29±0.015 | 0.25±0.003 | 0.30±0.01 | 0.28±0.01 | 0.05±0.001 | **0.005±0.001** |
| | COP-FT | 0.23±0.005 | **0.19±0.002** | 0.32±0.01 | **0.28±0.01** | 0.048±0.001 | 0.023±0.001 |
| | COP | 0.22±0.002 | 0.22±0.003 | 0.33±0.01 | 0.32±0.01 | 0.048±0.001 | 0.024±0.001 |
| | CPS (OURS) | **0.19±0.003** | **0.19±0.003** | **0.29±0.01** | **0.28±0.01** | **0.041±0.001** | **0.005±0.001** |
| DTW ($\downarrow$) | GUIDED DIFFTIME | 6.74±8.18 | 4.28±5.66 | 4.38±1.25 | 1.31±1.01 | 7.84±7.24 | 1.67±1.15 |
| | COP-FT | 3.52±2.08 | 2.01±1.24 | 4.61±1.08 | 1.26±0.87 | 0.90±1.41 | 1.19±0.64 |
| | COP | 3.72 ± 2.14 | 3.72 ± 2.12 | 5.16 ± 1.34 | 4.94 ± 1.08 | 0.88 ± 1.39 | 1.16 ± 0.65 |
| | CPS (OURS) | **2.35±1.48** | **1.83±1.16** | **3.41±1.47** | **0.84±0.62** | **0.20±0.71** | **0.23±0.17** |
| SSIM ($\uparrow$) | GUIDED DIFFTIME | 0.18±0.13 | 0.38±0.18 | 0.16±0.16 | 0.9±0.11 | 0.09±0.09 | 0.37±0.3 |
| | COP-FT | 0.19±0.11 | 0.48±0.16 | 0.10±0.14 | 0.89±0.14 | 0.15±0.10 | 0.35±0.11 |
| | COP | 0.17±0.11 | 0.17±0.11 | 0.09±0.13 | 0.09±0.13 | 0.14±0.09 | 0.39±0.12 |
| | CPS (OURS) | **0.38±0.15** | **0.52±0.15** | **0.31±0.20** | **0.95±0.07** | **0.73±0.26** | **0.96±0.05** |
| CONSTRAINT VIOLATION RATE ($\downarrow$) | GUIDED DIFFTIME | 1.0 | 1.0 | 0.99 | 0.89 | 1.0 | 0.933 |
| | COP-FT | **0.0** | **0.0** | **0.0** | **0.0** | **0.0** | 0.003 |
| | COP | **0.0** | **0.0** | 0.005 | **0.0** | **0.0** | 0.008 |
| | CPS (OURS) | **0.0** | **0.0** | **0.0** | **0.0** | **0.0** | **0.0** |
| CONSTRAINT VIOLATION MAGNITUDE ($\downarrow$) | GUIDED DIFFTIME | 23.21 | 16.35 | 0.50 | 0.15 | 1128.22 | 5.23 |
| | COP-FT | **0.0** | **0.0** | **0.0** | **0.0** | **0.0** | 0.0002 |
| | COP | **0.0** | **0.0** | 0.0001 | **0.0** | **0.0** | 0.0003 |
| | CPS (OURS) | **0.0** | **0.0** | **0.0** | **0.0** | **0.0** | **0.0** |

Table 1: **CPS outperforms existing baselines on sample quality and similarity metrics.** Yellow corresponds to sample quality metrics, and orange and violet correspond to similarity and constraint violation metrics, respectively. The best approach is shown in bold for each metric. Overall, we observe that CPS maintains high sample quality (very low FD and TSTR values) and the highest similarity with real time series samples (best values for the DTW and SSIM metrics). Our key intuition is that the adverse effects of projection step are nullified by the subsequent denoising steps. Note that as the constraints are all convex, the COP variants and CPS can achieve very low constraint violation.

generated sample and the real test sample that belongs to the constraint set. Similarly, for constraint violation, we report the ratio of the generated samples that do not belong to the constraint set to the total number of test samples. We also report the average constraint violation magnitude.

A detailed discussion on the baselines, metrics, etc., is provided in Appendix D.3 and Appendix B, respectively. Across all metrics, CPS outperforms the baselines on real-world and simulated datasets in conditional and unconditional settings. We provide intuitive reasons, backed by empirical evidence, for these performance gains by answering the following key questions:

**How well does CPS generate realistic samples that belong to the constraint set?** We argue about the performance of CPS based on the sample quality and the constraint violation metrics in Table 1. As the constraint sets used in our experiments are convex, both CPS and COP variants can almost always ensure constraint satisfaction using off-the-shelf solvers. However, Guided DiffTime struggles severely to generate samples that belong to the constraint set. This is clearly observed in the Stocks dataset, where Guided DiffTime has an average constraint violation magnitude of 1128. With respect to sample quality, we observe that the CPS provides the lowest FD and TSTR values.

Even though Guided DiffTime provides comparable TSTR values for some settings, we note that the generated samples are very less likely to belong to the constraint set. Therefore, guidance gradients alone are insufficient to drive the sample generation process to the constraint set. Similarly, there exists a considerable difference in performance between both COP variants, specifically for the conditional setting. Here, our key observation is that conditional generation provides a seed sample for COP-FT that lies close to the constraint set. Therefore, projection does not degrade the sample quality by a lot. However, the sample quality degradation due to projection is significant for COP, and it can be observed through very high values of J-FTSD in Table 1. Therefore, our key insight is that COP is influenced by the choice of initial seed.

**Does CPS handle unnatural artifacts that typically occur due to the projection step?** While imposing constraints on the generation process, we note that even though Guided DiffTime generates a realistic sample, it fails to adhere to the constraints. On the other hand, COP variants adhere to

constraints but generate samples with unnatural artifacts induced by the projection step. However, our key intuition is that CPS circumvents such artifacts using the iterative projection and denoising updates, where the adverse effects of the projection step are nullified by the subsequent denoising steps. The difference between the baseline approaches and CPS is significantly pronounced specifically in the waveforms dataset (check Fig. 6). The stark contrast between the generated sinusoid from CPS and other baselines is empirically supported with a $100\times$ reduction in the FD value.

**How does CPS perform in comparison with baselines for a large number of constraints?** We consider the Stocks dataset as the OHLC condition introduces more than 400 constraints. With a large number of constraints, the feasible set size reduces, and this necessitates the requirement of accurate guidance to generate samples from such constraint sets. In such settings, Guided DiffTime performs poorly. This can be attributed to the interaction between gradients for each constraint violation. The combination of these gradients, if not scaled appropriately, leads to poor guidance. Additionally, finding the correct set of guidance weights is practically very hard for a large number of constraints. Similarly, projection to small constraint sets affects the sample quality of COP variants, specifically when the initial seed is far away from the constraint set. While the baselines suffer with an increasing number of constraints, CPS gets rid of these issues by alternating projection and denoising updates. We observe this through the qualitative example from the Stocks datasets in Fig. 3. Quantitatively, we observe $14.5\%$ improvement in the TSTR and $67\%$ improvement in the FD metric when compared against the best-performing baseline.

**Does CPS track the real test samples that adhere to the same set of constraints?** For a large number of constraints or a small constraint set, we expect the generated samples that satisfy the constraints to have a high degree of similarity with the real test samples from which we extract the constraints. To this end, we denote tracking real test samples as the property to have better similarity scores with the real sample as the number of constraints increases. In Fig. 5, we note that CPS outperforms all baselines in the DTW metric for any number of constraints, thereby showing higher similarity with the real test samples. Note that out of all approaches, CPS has the best reduction in the DTW scores as the number of constraints increases. Simultaneously, we also note that the sample quality is unaffected or even improves for CPS with increasing constraints (lower FD scores). We observe that CPS's performance is consistent across multiple real-world datasets, with significant improvements in the DTW values of around 33% for Air Quality, 77% for Stocks, and 22% for the Traffic dataset with respect to the best-performing baseline.

We refer the reader to Appendix G, where we show that CPS outperforms PDM (Christopher et al., 2024) on sample quality metrics. Additionally, we show that CPS beats Diffusion-TS (Yuan & Qiao, 2024) and Loss DiffTime (Coletta et al., 2024) on constraint violation metrics.

**Limitations.** Although CPS outperforms all the compared baselines in standard evaluation metrics, we note that the projection step (line 5) in Algorithm 1 can be time-consuming for some applications. This increases the overall sampling time of CPS, the trade-off being superior performance. In time-critical applications, the sampling time can be reduced further by leveraging higher order moments and different initialization schemes (Rout et al., 2024a). Additionally, the projection step is not necessary after every denoising step and can be adapted to the constraint violation magnitude.

## 5 CONCLUSION

We proposed Constrained Posterior Sampling – a novel training-free approach for constrained time series generation. CPS is designed such that it exploits off-the-shelf optimization routines to perform a projection step towards the constraint set after every denoising step. Through an array of sample quality and constraint violation metrics, we empirically show that CPS outperforms the state-of-the-art baselines in generating realistic samples that belong to a constraint set.

**Future work.** We aim to apply our approach for constrained trajectory generation in the robotics domain with dynamic constraints typically modeled by neural networks. Additionally, constrained time series generation readily applies to style transfer applications. Hence, we plan on extending the current work to perform style transfer from one time series to another by perturbing statistical features.

**Reproducibility.** The pseudo-code and hyper-parameter details have been provided in the Appendix to help reproduce the results reported in the paper. The source code will be released post publication.

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

APPENDIX

# A PROOFS

In this section, we provide the detailed proof for the theorems stated in the manuscript.

## A.1 PROOF OF THEOREM 1

We first describe the assumption on the constraint set. The constraint set is defined as $\mathcal{C} = \{z \mid f_{\mathcal{C}}(z) = 0\}$, where $f_{\mathcal{C}} : \mathbb{R}^n \to \mathbb{R}$, and the penalty function $\Pi(z) = \|f_{\mathcal{C}}(z)\|_2^2$ has $L$-Lipschitz continuous gradients, *i.e.,* $\|\nabla\Pi(u) - \nabla\Pi(v)\|_2 \leq L\|u - v\|_2 \ \forall \ u, v \in \mathbb{R}^n$.

Line 7 of the Algorithm 1 modifies the traditional DDIM sampling by replacing $\hat{z}_0(z_t; \epsilon_\theta)$ with $\hat{z}_{0,\mathrm{pr}}(z_t; \epsilon_\theta)$. Without this modification, the DDIM sampling denotes the following reverse process when started with $x_T \sim \mathcal{N}(\mathbf{0}_n, \mathbf{I}_n)$, where $\mathbf{0}_n$ indicates the zero mean vector in $\mathbb{R}^n$ and $\mathbf{I}_n$ is the identity matrix in $\mathbb{R}^{n \times n}$:

$$p_{\theta,t}(z_{t-1} \mid z_t) = \begin{cases} p_{\theta,\mathrm{init}}(z_0 \mid \hat{z}_0(z_1; \epsilon_\theta)) & \text{if } t = 1, \\ q_{\sigma,t}(z_{t-1} \mid z_t, \hat{z}_0(z_t; \epsilon_\theta)) & \text{otherwise,} \end{cases} \tag{6}$$

where $q_{\sigma,t}(z_{t-1} \mid z_t, \hat{z}_0(z_t; \epsilon_\theta))$ represents the PDF of the Gaussian distribution $\mathcal{N}\left(\sqrt{\bar{\alpha}_{t-1}}\hat{z}_0(z_t; \epsilon_\theta) + \sqrt{1 - \bar{\alpha}_{t-1} - \sigma_t^2}\epsilon_\theta(z_t, t), \sigma_t^2\mathbf{I}_n\right)$ with $\sigma_t$ as the DDIM control parameter. Similarly, $p_{\theta,\mathrm{init}}(z_0 \mid \hat{z}_0(z_1; \epsilon_\theta))$ is the PDF of the Gaussian distribution with mean $\hat{z}_0(z_1; \epsilon_\theta)$ and covariance matrix $\sigma_1^2\mathbf{I}_n$ (Song et al., 2022).

Note that sampling from $q_{\sigma,t}(z_{t-1} \mid z_t, \hat{z}_0(z_t; \epsilon_\theta))$ provides the DDIM sampling step (check Eq. 2).

We reiterate that the main modification with respect to the DDIM sampling approach is the projection step in line 5 of Algorithm 1. Therefore, we first analyze the projection step,

$$\hat{z}_{0,\mathrm{pr}}(z_t; \epsilon_\theta) = \arg\min_z \frac{1}{2}\left(\|z - \hat{z}_0(z_t; \epsilon_\theta)\|_2^2 + \gamma(t)\|f_{\mathcal{C}}(z)\|_2^2\right). \tag{7}$$

Here, $\hat{z}_0(z_t; \epsilon_\theta) = \frac{z_t - \sqrt{1 - \bar{\alpha}_t}\epsilon_\theta(z_t, t)}{\sqrt{\bar{\alpha}_t}}$ (line 3, predicted $z_0$). We will denote the objective function $\frac{1}{2}\left(\|z - \hat{z}_0(z_t; \epsilon_\theta)\|_2^2 + \gamma(t)\|f_{\mathcal{C}}(z)\|_2^2\right)$ as $g(z)$. Note that we replaced the constraint violation function $\Pi(z)$ by $\|f_{\mathcal{C}}(z)\|_2^2$ for this case. Given that $f_{\mathcal{C}}$ is a differentiable and convex with $\|f_{\mathcal{C}}\|_2^2$ having $L$-Lipschitz continuous gradients, Eq. 7 can be written as a series of gradient updates with a suitable step size such that the value of the objective function decreases for each gradient update.

From the statement, we observe that $\gamma(t) > 0 \ \forall \ t \in [1, T]$. Under this condition and Assumption 1, note that the function $g(z)$ is convex and has $\left(\frac{2 + \gamma(t)L}{2}\right)$-Lipschitz continuous gradients, as $\|z - \hat{z}_0(z_t; \epsilon_\theta)\|_2^2$ has 2-Lipschitz continuous gradients, $\gamma(t)\|f_{\mathcal{C}}(z)\|_2^2$ has $(\gamma(t)L)$-Lipschitz continuous gradients, and the fraction $\frac{1}{2}$ makes $g(z)$ to have $\left(\frac{2 + \gamma(t)L}{2}\right)$-Lipschitz continuous gradients. Let $\eta$ be the step size of the projection step. From Nocedal & Wright (1999), we know that $\eta \in (0, 2/(2 + \gamma(t)L))$ ensures that the objective function in Eq. 7 reduces after each gradient update. We denote the gradient update as:

$$^n\hat{z}_0(z_t; \epsilon_\theta) = \ ^{n-1}\hat{z}_0(z_t; \epsilon_\theta) - \eta\nabla_z(g(z))\big|_{^{n-1}\hat{z}_0(z_t;\epsilon_\theta)}, \tag{8}$$

where $^0\hat{z}_0(z_t; \epsilon_\theta) = \hat{z}_0(z_t; \epsilon_\theta)$ and $\hat{z}_{0,\mathrm{pr}}(z_t; \epsilon_\theta) = \ ^{N_{\mathrm{pr}}}\hat{z}_0(z_t; \epsilon_\theta)$. Here, $N_{\mathrm{pr}}$ is the total number of gradient update steps.

The iteration in Eq. 8 always leads to $\hat{z}_{0,\mathrm{pr}}(z_t; \epsilon_\theta)$ deterministically. Therefore, the projection step can be considered sampling from a Dirac delta distribution centered at $\hat{z}_{0,\mathrm{pr}}(z_t; \epsilon_\theta)$, *i.e.,* $\delta(z - \hat{z}_{0,\mathrm{pr}}(z_t; \epsilon_\theta))$. Consequently, using the law of total probability, the reverse process corresponding to the denoising step $t \ \forall \ t \in [2, T]$ in Algorithm 1 is given by

$$p_{\theta,t}(z_{t-1} \mid z_t) = \int p_{\theta,t}(z_{t-1}, \hat{z}_0 \mid z_t)d\hat{z}_0,$$

where $\hat{z}_0 \in \mathbb{R}^n$. This can be simplified using Bayes' rule as

$$p_{\theta,t}(z_{t-1} \mid z_t) = \int \delta(\hat{z}_0 - \hat{z}_{0,\mathrm{pr}}(z_t; \epsilon_\theta)) q_{\sigma,t}(z_{t-1} \mid z_t, \hat{z}_0) d\hat{z}_0.$$

The above equation stems from the fact that the distribution of $z_0$ conditioned on $z_t$ is a Dirac delta distribution centered at $\hat{z}_{0,\mathrm{pr}}(z_t; \epsilon_\theta)$. Since $\delta(x-y) = \delta(y-x)$ and using the sifting property of a Dirac delta function $\left(\int f(z)\delta(a-z)dz = f(a)\right)$, we get

$$p_{\theta,t}(z_{t-1} \mid z_t) = q_{\sigma,t}(z_{t-1} \mid z_t, \hat{z}_{0,\mathrm{pr}}(z_t; \epsilon_\theta)) \,\forall\, t \in [2, T]. \tag{9}$$

Similarly, we repeat the steps for $t = 1$,

$$p_{\theta,1}(z_0 \mid z_1) = \int p_{\theta,1}(z_0, \hat{z}_0 \mid z_t) d\hat{z}_0,$$

$$p_{\theta,1}(z_0 \mid z_1) = \int \delta(\hat{z}_0 - \hat{z}_{0,\mathrm{pr}}(z_1; \epsilon_\theta)) p_{\theta,\mathrm{init}}(z_0 \mid \hat{z}_0) d\hat{z}_0,$$

$$p_{\theta,1}(z_0 \mid z_1) = p_{\theta,\mathrm{init}}(z_0 \mid \hat{z}_{0,\mathrm{pr}}(z_1; \epsilon_\theta)).$$

Combining the two, we get

$$p_{\theta,t}(z_{t-1} \mid z_t) = \begin{cases} p_{\theta,\mathrm{init}}(z_0 \mid \hat{z}_{0,\mathrm{pr}}(z_1; \epsilon_\theta)) & \text{if } t = 1, \\ q_{\sigma,t}(z_{t-1} \mid z_t, \hat{z}_{0,\mathrm{pr}}(z_t; \epsilon_\theta)) & \text{otherwise,} \end{cases} \tag{10}$$

where $q_{\sigma,t}(z_{t-1} \mid z_t, \hat{z}_{0,\mathrm{pr}}(z_t; \epsilon_\theta))$ represents the PDF of the Gaussian distribution $\mathcal{N}(\sqrt{\bar{\alpha}_{t-1}}\hat{z}_{0,\mathrm{pr}}(z_t; \epsilon_\theta) + \sqrt{1 - \bar{\alpha}_{t-1} - \sigma_t^2}\epsilon_\theta(z_t, t), \sigma_t^2 \mathbf{I}_n)$ with $\sigma_t$ as the DDIM control parameter. Similarly, $p_{\theta,\mathrm{init}}(z_0 \mid \hat{z}_{0,\mathrm{pr}}(z_1; \epsilon_\theta))$ is the PDF of the Gaussian distribution with mean $\hat{z}_{0,\mathrm{pr}}(z_1; \epsilon_\theta)$ and covariance matrix $\sigma_1^2 \mathbf{I}_n$ (Song et al., 2022). This is the same as Eq. 5.

We note that the value of $\sigma_1$ is set to 0 in Algorithm 1. However, similar to (Song et al., 2022), for theoretical analysis, we consider a negligible value for $\sigma_1$ ($\sim 10^{-12}$) to ensure that the generative process is supported everywhere. In other words, $\sigma_1$ is chosen to be so low such that for $\sigma_1 \simeq 0$, $p_{\theta,\mathrm{init}}(z_0 \mid \hat{z}_{0,\mathrm{pr}}(z_1; \epsilon_\theta)) \simeq \delta(z_0 - \hat{z}_{0,\mathrm{pr}}(z_1; \epsilon_\theta))$.

Now, we show that the exact DDIM reverse process (check Eq. 6) can be obtained from Eq. 10 in the case where there are no constraints. Here, note that in the absence of any constraint, the projection step can be written as $\hat{z}_{0,\mathrm{pr}}(z_t; \epsilon_\theta) = \arg\min_z \frac{1}{2}\|z - \hat{z}_0(z_t; \epsilon_\theta)\|_2^2$, in which case $\hat{z}_{0,\mathrm{pr}}(z_t; \epsilon_\theta) = \hat{z}_0(z_t; \epsilon_\theta)$.

For $t \in [2, T]$, using the law of total probability, we get

$$p_{\theta,t}(z_{t-1} \mid z_t) = \int \delta(\hat{z}_0 - \hat{z}_0(z_t; \epsilon_\theta)) q_{\sigma,t}(z_{t-1} \mid z_t, \hat{z}_0) d\hat{z}_0, \tag{11}$$

which simplifies further to

$$p_{\theta,t}(z_{t-1} \mid z_t) = q_{\sigma,t}(z_{t-1} \mid z_t, \hat{z}_0(z_t; \epsilon_\theta)). \tag{12}$$

The above equation stems from the same sifting property of Dirac delta functions. The same applies to $t = 1$, except that after the projection step since there is no necessity for constraint satisfaction, we sample from $p_{\theta,\mathrm{init}}(z_0 \mid \hat{z}_0(z_1; \epsilon_\theta))$, which is a Gaussian distribution with mean $\hat{z}_0(z_1; \epsilon_\theta)$ and covariance matrix $\sigma_1^2 \mathbf{I}_n$.

Combining both cases, we observe that without any constraints the exact DDIM reverse process can be recovered from Algorithm 1 for all $t \in [1, T]$.

## A.2 PROOF OF THEOREM 2

We note that the intermediate samples in a $T$-step reverse sampling process are denoted as $z_T, \ldots, z_0$, where $z_0 = x^{\mathrm{gen}}$ and $z_T \sim \mathcal{N}(\mathbf{0}_n, \mathbf{I}_n)$. Once again, we reiterate the assumptions. We consider the real data distribution to be Gaussian with mean $\mu \in \mathbb{R}^n$ and covariance matrix $\mathbf{I}_n$, i.e., $\mathcal{N}(\mu, \mathbf{I}_n)$. The constraint set $\mathcal{C}$ is defined as $\mathcal{C} = \{z \mid Az = b\}$ with $A \in \mathbb{R}^{m \times n}$ such that $rank(A) = n$, where $m \geq n$. Additionally, for the real data distribution $\mathcal{N}(\mu, \mathbf{I}_n)$ and the constraint set $\mathcal{C} = \{z \mid Az = y\}$, there exists a unique solution to Eq. 4, indicated by $x^*$.

Given that $rank(A) = n$ for $A \in \mathbb{R}^{m \times n}$ with $m \geq n$, we note that $(A^T A)^{-1}$ exists. Consequently, $\lambda_{\min}(A^T A) > 0$.

From the theorem statement, we have $\gamma(t) = \frac{2k(T-t+1)}{\lambda_{\min}(A^T A)}$, with $k > 1$. Immediately, we note that for all $t \in [1, T]$, $\gamma(t) > 0$. More specifically, $t \in [1, T]$, $\gamma(t) > \frac{2}{\lambda_{\min}(A^T A)}$.

The proof is divided into 2 parts. First, we obtain the expression for $z_{t-1}$ in terms of $z_t$. Then, we obtain an upper bound for $\|z_0 - x^*\|_2$, which is the same as $\|x^{\text{gen}} - x^*\|_2$, as from Algorithm 1 we note that $z_0 = x^{\text{gen}}$.

First, we note that for deterministic sampling, we have the DDIM control parameters $\sigma_1 \dots \sigma_T = 0$. Therefore, the DDIM reverse sampling step from Algorithm 1 (line7) can be written as

$$z_{t-1} = \sqrt{\bar{\alpha}_{t-1}} \hat{z}_{0,\text{pr}}(z_t; \epsilon_\theta) + \sqrt{1 - \bar{\alpha}_{t-1}} \epsilon_\theta(z_t, t). \tag{13}$$

Since the true data distribution is Gaussian, the optimal denoiser $\epsilon^*(z_t, t)$ can be expressed analytically for any diffusion step $t$. Therefore, the deterministic sampling step can be written as

$$z_{t-1} = \sqrt{\bar{\alpha}_{t-1}} \hat{z}_{0,\text{pr}}(z_t; \epsilon^*) + \sqrt{1 - \bar{\alpha}_{t-1}} \epsilon^*(z_t, t).$$

We can obtain an analytical expression for the optimal denoiser from Lemma 1. Using Eq. 27 from Lemma 1, we note that the optimal denoiser at the diffusion step $t$ is

$$\epsilon^*(z_t, t) = -\sqrt{1 - \bar{\alpha}_t}(\sqrt{\bar{\alpha}_t}\mu - z_t). \tag{14}$$

Now, we obtain the expression for $\hat{z}_{0,\text{pr}}(z_t; \epsilon^*)$. Note that the constraint violation function is defined as $\Pi(z) = \|y - Az\|_2^2$. Consequently, we note that the objective function in line 5 of Algorithm 1, i.e., $\frac{1}{2}(\|z - \hat{z}_0(z_t; \epsilon_\theta)\|_2^2 + \gamma(t)\|y - Az\|_2^2)$, is convex with respect to $z$ for $\gamma(t) > 0$. As such, we use Lemmas 1 and 2 to obtain the expression for $\hat{z}_{0,\text{pr}}(z_t; \epsilon^*)$,

$$\hat{z}_{0,\text{pr}}(z_t; \epsilon^*) = [\mathbf{I}_n + \gamma(t)A^T A]^{-1}[\mu - \bar{\alpha}_t\mu + \sqrt{\bar{\alpha}_t}z_t + \gamma(t)A^T y]. \tag{15}$$

We substitute the expressions for $\epsilon^*(z_t, t)$ from Eq. 14 and $\hat{z}_{0,\text{pr}}(z_t; \epsilon^*)$ from Eq. 15, respectively, in addition to replacing $y$ with $Ax^*$, to obtain $z_{t-1}$ in terms of $z_t$:

$$
\begin{aligned}
z_{t-1} =& \sqrt{\bar{\alpha}_{t-1}} \left[\mathbf{I}_n + \gamma(t)A^T A\right]^{-1} \left[\mu - \bar{\alpha}_t\mu + \sqrt{\bar{\alpha}_t}z_t + \gamma(t)A^T y\right] \\
& + \sqrt{1 - \bar{\alpha}_{t-1}}(-\sqrt{1 - \bar{\alpha}_t}(\sqrt{\bar{\alpha}_t}\mu - z_t)), \\
z_{t-1} =& \sqrt{\bar{\alpha}_{t-1}} \left[\mathbf{I}_n + \gamma(t)A^T A\right]^{-1} \left[\mu - \bar{\alpha}_t\mu + \sqrt{\bar{\alpha}_t}z_t + \gamma(t)A^T y\right] \\
& - \sqrt{1 - \bar{\alpha}_{t-1}}\sqrt{1 - \bar{\alpha}_t}\sqrt{\bar{\alpha}_t}\mu + \sqrt{1 - \bar{\alpha}_{t-1}}\sqrt{1 - \bar{\alpha}_t}z_t, \\
z_{t-1} =& \sqrt{\bar{\alpha}_{t-1}} \left[\mathbf{I}_n + \gamma(t)A^T A\right]^{-1} \left[\mu - \bar{\alpha}_t\mu + \sqrt{\bar{\alpha}_t}z_t + \gamma(t)A^T Ax^*\right] \\
& - \sqrt{1 - \bar{\alpha}_{t-1}}\sqrt{1 - \bar{\alpha}_t}\sqrt{\bar{\alpha}_t}\mu + \sqrt{1 - \bar{\alpha}_{t-1}}\sqrt{1 - \bar{\alpha}_t}z_t, \\
z_{t-1} =& \left[\sqrt{\bar{\alpha}_{t-1}}\sqrt{\bar{\alpha}_t} \left[\mathbf{I}_n + \gamma(t)A^T A\right]^{-1} + \sqrt{1 - \bar{\alpha}_{t-1}}\sqrt{1 - \bar{\alpha}_t}\mathbf{I}_n\right] z_t \\
& + \left[\sqrt{\bar{\alpha}_{t-1}} \left[\mathbf{I}_n + \gamma(t)A^T A\right]^{-1} - \bar{\alpha}_t\sqrt{\bar{\alpha}_{t-1}} \left[\mathbf{I}_n + \gamma(t)A^T A\right]^{-1}\right] \mu \\
& - \left[\sqrt{1 - \bar{\alpha}_{t-1}}\sqrt{1 - \bar{\alpha}_t}\sqrt{\bar{\alpha}_t}\mathbf{I}_n\right] \mu + \gamma(t)\sqrt{\bar{\alpha}_{t-1}} \left[\mathbf{I}_n + \gamma(t)A^T A\right]^{-1} A^T Ax^*, \\
z_{t-1} =& \left[\sqrt{\bar{\alpha}_{t-1}}\sqrt{\bar{\alpha}_t} \left[\mathbf{I}_n + \gamma(t)A^T A\right]^{-1} + \sqrt{1 - \bar{\alpha}_{t-1}}\sqrt{1 - \bar{\alpha}_t}\mathbf{I}_n\right] z_t \\
& + \left[(1 - \bar{\alpha}_t)\sqrt{\bar{\alpha}_{t-1}} \left[\mathbf{I}_n + \gamma(t)A^T A\right]^{-1}\right] \mu - \left[\sqrt{1 - \bar{\alpha}_{t-1}}\sqrt{1 - \bar{\alpha}_t}\sqrt{\bar{\alpha}_t}\mathbf{I}_n\right] \mu \\
& + \gamma(t)\sqrt{\bar{\alpha}_{t-1}} \left[\mathbf{I}_n + \gamma(t)A^T A\right]^{-1} A^T Ax^*.
\end{aligned}
$$

On further simplification, we have

$$z_{t-1} = K_t z_t + E_t\mu - F_t\mu + \gamma(t)\sqrt{\bar{\alpha}_{t-1}} \left[\mathbf{I}_n + \gamma(t)A^T A\right]^{-1} A^T Ax^*.$$

where we have the following matrix definitions,

$$K_t = \left[\sqrt{\bar{\alpha}_{t-1}}\sqrt{\bar{\alpha}_t} \left[\mathbf{I}_n + \gamma(t)A^T A\right]^{-1} + \sqrt{1 - \bar{\alpha}_{t-1}}\sqrt{1 - \bar{\alpha}_t}\mathbf{I}_n\right], \tag{16}$$

$$E_t = \left[ (1 - \bar{\alpha}_t)\sqrt{\bar{\alpha}_{t-1}} \left[ \mathbf{I}_n + \gamma(t)A^T A \right]^{-1} \right], \tag{17}$$

$$F_t = \left[ \sqrt{1 - \bar{\alpha}_{t-1}}\sqrt{1 - \bar{\alpha}_t}\sqrt{\bar{\alpha}_t}\mathbf{I}_n \right]. \tag{18}$$

The goal is to obtain the upper bound for $\|x^{\text{gen}} - x^*\|_2$. Note that $\|x^{\text{gen}} - x^*\|_2 = \|z_0 - x^*\|_2$. So, first, we subtract $x^*$ from both sides to obtain

$$z_{t-1} - x^* = K_t z_t + E_t \mu - F_t \mu + \gamma(t)\sqrt{\bar{\alpha}_{t-1}} \left[ \mathbf{I}_n + \gamma(t)A^T A \right]^{-1} A^T A x^* - x^*.$$

Further, we add and subtract $K_t x^*$ to the right side to obtain

$$z_{t-1} - x^* = K_t z_t - K_t x^* + E_t \mu - F_t \mu + \gamma(t)\sqrt{\bar{\alpha}_{t-1}} \left[ \mathbf{I}_n + \gamma(t)A^T A \right]^{-1} A^T A x^* - x^* + K_t x^*.$$

We further simplify the above expression to obtain

$$z_{t-1} - x^* = K_t \left( z_t - x^* \right) + E_t \mu - F_t \mu + K_t x^* + D_t x^*,$$

where the matrix definition of $D_t$ is

$$D_t = \gamma(t)\sqrt{\bar{\alpha}_{t-1}} \left[ \mathbf{I}_n + \gamma(t)A^T A \right]^{-1} A^T A - \mathbf{I}_n. \tag{19}$$

Now, we obtain the expression for $\|z_{t-1} - x^*\|_2$ in terms of $\|z_t - x^*\|_2$.

$$\|z_{t-1} - x^*\|_2 = \|K_t(z_t - x^*) + E_t \mu - F_t \mu + K_t x^* + D_t x^*\|_2.$$

Applying the triangle inequality repeatedly, we get

$$\|z_{t-1} - x^*\|_2 \le \|K_t(z_t - x^*)\|_2 + \|K_t x^*\|_2 + \|D_t x^*\|_2 + \|E_t \mu\|_2 + \|F_t \mu\|_2. \tag{20}$$

Before obtaining the upperbound for $\|z_0 - x^*\|$, for $\gamma(t) > 0$, we will first show that $\|K_t\|_2, \|D_t\|_2, \|E_t\|_2, \|F_t\|_2 < 1 \,\forall\, t \in [1, T]$. Here $\|K_t\|_2$ refers to the spectral norm of the matrix $K_t$. To show this, we establish a few relationships that will be the recurring theme used in proving that $\|K_t\|_2, \|D_t\|_2, \|E_t\|_2, \|F_t\|_2 < 1 \,\forall\, t \in [1, T]$.

The spectral norm of the matrix M is defined as $\|M\|_2 = \max_{x \ne 0} \frac{\|Mx\|_2}{\|x\|_2}$. From this definition, we immediately note the following two inequalities.

- $\|Mx\|_2 \le \|M\|_2 \|x\|_2$ as $\|M\|_2 = \max_{x \ne 0} \frac{\|Mx\|_2}{\|x\|_2}$.
- $\|MN\|_2 = \max_{x \ne 0} \frac{\|MNx\|_2}{\|x\|_2} \le \max_{x \ne 0} \frac{\|M\|_2 \|Nx\|_2}{\|x\|_2} \le \max_{x \ne 0} \frac{\|M\|_2 \|N\|_2 \|x\|_2}{\|x\|_2} = \|M\|_2 \|N\|_2$.

Further, we note that the following are well-established properties for spectral norms and positive definite matrices. Consider a positive definite matrix $M$, *i.e.*, $M \succ 0$.

- $\|M\|_2$ is equal to the largest eigen value of $M$, *i.e.*, $\lambda_{\max}(M)$.
- $\|M^{-1}\|_2 = \frac{1}{\lambda_{\min}(M)}$ as the eigenvalues of $M^{-1}$ are the reciprocal of the eigenvalues of $M$.
- $\| - M\|_2 = \|M\|_2$.

We refer the readers to Lemmas 3, 8, and 10, where we show that $\|K_t\|_2, \|E_t\|_2, \|F_t\|_2 < 1 \,\forall\, t \in [1, T]$, if $\gamma(t) > 0$.

Similarly, Lemma 6 shows that $\|D_t\|_2 < 1 \,\forall\, t \in [1, T]$, if $\gamma(t) > \frac{2}{\lambda_{\min}A^T A}$.

We first apply the inequality $\|Mx\|_2 \le \|M\|_2 \|x\|_2$ to simplify Eq. 20 as follows.

$$\|z_{t-1} - x^*\|_2 \le \|K_t\|_2 \|z_t - x^*\|_2 + \|K_t x^*\|_2 + \|D_t x^*\|_2 + \|E_t \mu\|_2 + \|F_t \mu\|_2. \tag{21}$$

Therefore, we can recursively obtain the upper bound for $\|z_t - x^*\|_2$ in terms of $\|z_T - x^*\|_2$. This process, repeated $T$ times, provides the upper bound for $\|z_0 - x^*\|_2$.

$$
\begin{aligned}
\|z_0 - x^*\|_2 \le\ & \|K_1\|_2 \|K_2\|_2 \dots \|K_T\|_2 \|(z_T - x^*)\|_2 \\
& + (\|K_1\|_2 + \|K_1\|_2 \|K_2\|_2 + \dots + \|K_1\|_2 \|K_2\|_2 \dots \|K_{T-1}\|_2 \|K_T\|_2)\|x^*\|_2 \\
& + (\|D_1\|_2 + \|K_1\|_2 \|D_2\|_2 + \dots + \|K_1\|_2 \|K_2\|_2 \dots \|K_{T-1}\|_2 \|D_T\|_2)\|x^*\|_2 \\
& + (\|E_1\|_2 + \|K_1\|_2 \|E_2\|_2 + \dots + \|K_1\|_2 \|K_2\|_2 \dots \|K_{T-1}\|_2 \|E_T\|_2)\|\mu\|_2 \\
& + (\|F_1\|_2 + \|K_1\|_2 \|F_2\|_2 + \dots + \|K_1\|_2 \|K_2\|_2 \dots \|K_{T-1}\|_2 \|F_T\|_2)\|\mu\|_2.
\end{aligned} \tag{22}
$$

Let $\lambda_k = \max_t \left(\|K_1\|_2, \|K_2\|_2, \ldots, \|K_T\|_2\right)$. Since for $\gamma(t) > 0$, $\|K_1\|_2, \ldots, \|K_T\|_2 < 1$, we note that $\lambda_k < 1$.

Therefore, $\|K_1\|_2 \|K_2\|_2 \ldots \|K_T\|_2$ can be upper bounded by $\lambda_k^T$.

Additionally, note that $\|K_1\|_2 \|K_2\|_2 \leq \|K_1\|_2$ as $\|K_2\|_2 < 1$. Therefore, $(\|K_1\|_2 + \|K_1\|_2 \|K_2\|_2 + \cdots + \|K_1\|_2 \|K_2\|_2 \ldots \|K_{T-1}\|_2 \|K_T\|_2)$ can be upper bounded by $T\|K_1\|_2$.

Similarly, $(\|K_1\|_2 \|D_2\|_2 + \cdots + \|K_1\|_2 \|K_2\|_2 \ldots \|K_{T-1}\|_2 \|D_T\|_2)$ can be upperbounded by $(T - 1)\|K_1\|_2$.

The same applies to $(\|K_1\|_2 \|E_2\|_2 + \cdots + \|K_1\|_2 \|K_2\|_2 \ldots \|K_{T-1}\|_2 \|E_T\|_2)$ and $(\|K_1\|_2 \|F_2\|_2 + \cdots + \|K_1\|_2 \|K_2\|_2 \ldots \|K_{T-1}\|_2 \|F_T\|_2)$.

Therefore, the upper bound in Eq. 22 can be simplified as

$$\|z_0 - x^*\| \leq \lambda_k^T \|(z_T - x^*)\|_2 + T\|K_1\|_2 \|x^*\|_2 + (\|D_1\|_2 + (T - 1)\|K_1\|_2)\|x^*\|_2$$
$$+ (\|E_1\|_2 + (T - 1)\|K_1\|_2)\|\mu\|_2 + (\|F_1\|_2 + (T - 1)\|K_1\|_2)\|\mu\|_2. \quad (23)$$

Consequently, in Lemmas 4, 7, 9, 10, we show

$$\|K_1\|_2 \leq \frac{\sqrt{\bar{\alpha}_1}}{1 + \gamma(1)\lambda_{\min}(A^T A)} < 1 \text{ if } \gamma(1) > 0,$$

$$\|D_1\|_2 \leq \frac{1}{\gamma(1)\lambda_{\min}(A^T A)) - 1} < 1 \text{ if } \gamma(1) > \frac{2}{\lambda_{\min}(A^T A)},$$

$$\|E_1\|_2 \leq \frac{1 - \bar{\alpha}_1}{1 + \gamma(1)\lambda_{\min}(A^T A)} < 1 \text{ if } \gamma(1) > 0,$$

$$\|F_1\|_2 = 0. \quad (24)$$

For our choice of $\gamma(1) = \frac{2kT}{\lambda_{\min}(A^T A)}$, we first note that $\gamma(1) > 0$ and $\gamma(1) > \frac{2}{\lambda_{\max}(A^T A)}$ for $k > 1/T$. Therefore, we can rewrite the above inequalities as

$$\|K_1\|_2 \leq \frac{\sqrt{\bar{\alpha}_1}}{1 + 2kT},$$

$$\|D_1\|_2 \leq \frac{1}{2kT - 1},$$

$$\|E_1\|_2 \leq \frac{1 - \bar{\alpha}_1}{1 + 2kT},$$

$$\|F_1\|_2 = 0. \quad (25)$$

Therefore, Eq. 23 can be upper bounded using Eq. 25 as shown below:

$$\|z_0 - x^*\|_2 \leq \lambda_k^T \|(z_T - x^*)\|_2 + T\left(\frac{\sqrt{\bar{\alpha}_1}}{1 + 2kT}\right)\|x^*\|_2 + \left(\frac{1}{2kT - 1}\right)\|x^*\|_2 +$$
$$\left(\frac{1 - \bar{\alpha}_1}{1 + 2kT}\right)\|\mu\|_2 + (T - 1)\left(\frac{\sqrt{\bar{\alpha}_1}}{1 + 2kT}\right)\|x^*\|_2 + 2(T - 1)\left(\frac{\sqrt{\bar{\alpha}_1}}{1 + 2kT}\right)\|\mu\|_2. \quad (26)$$

As $T \to \infty$, we observe the following:

$$\lim_{T \to \infty} \lambda_k^T \|(z_T - x^*)\|_2 = 0 \quad (\lambda_k < 1),$$

$$\lim_{T \to \infty} T\left(\frac{\sqrt{\bar{\alpha}_1}}{1 + 2kT}\right)\|x^*\|_2 = \left(\frac{\sqrt{\bar{\alpha}_1}}{2k}\right)\|x^*\|_2 \quad (\text{if } k > 0),$$

$$\lim_{T \to \infty} \left(\frac{1}{2kT - 1}\right)\|x^*\|_2 = 0,$$

$$\lim_{T \to \infty} \left(\frac{1 - \bar{\alpha}_1}{1 + 2kT}\right)\|\mu\|_2 = 0,$$

$$\lim_{T \to \infty} (T - 1)\left(\frac{\sqrt{\bar{\alpha}_1}}{1 + 2kT}\right)\|x^*\|_2 = \left(\frac{\sqrt{\bar{\alpha}_1}}{2k}\right)\|x^*\|_2 \quad (\text{if } k > 0),$$

$$\lim_{T \to \infty} 2(T - 1)\left(\frac{\sqrt{\bar{\alpha}_1}}{1 + 2kT}\right)\|\mu\|_2 = \left(\frac{\sqrt{\bar{\alpha}_1}}{k}\right)\|\mu\|_2 \quad (\text{if } k > 0).$$

Therefore, in the limit $T \to \infty$, we have

$$\|z_0 - x^*\|_2 \leq \frac{\sqrt{\bar{\alpha}_1}}{k} \left( \|x^*\|_2 + \|\mu\|_2 \right) \text{ or,}$$

$$\|x^{\text{gen}} - x^*\|_2 \leq \frac{\sqrt{\bar{\alpha}_1}}{k} \left( \|x^*\|_2 + \|\mu\|_2 \right).$$

**Lemma 1.** *Suppose **Assumption 2** holds. Consider a $T$-step diffusion process with coefficients $\bar{\alpha}_0, \ldots, \bar{\alpha}_T$ such that $\bar{\alpha}_0 = 1$, $\bar{\alpha}_T = 0$, $\bar{\alpha}_t \in [0, 1]$. The optimal denoiser $\epsilon^*(z_t, t)$ is given by*

$$\epsilon^*(z_t, t) = -\sqrt{1 - \bar{\alpha}_t}(\sqrt{\bar{\alpha}_t}\mu - z_t).$$

*Proof.* We first observe the distribution of $z_t$.

For the diffusion forward process, we know that $z_t = \sqrt{\bar{\alpha}_t}z_0 + \sqrt{1 - \bar{\alpha}_t}\epsilon$, where $\epsilon \sim \mathcal{N}(\mathbf{0}_n, \mathbf{I}_n)$.

Note that $z_0$ is a sample from the Gaussian distribution $\mathcal{N}(\mu, \mathbf{I}_n)$.

Consequently, we note that $z_t$ is a sample from the Gaussian distribution $\mathcal{N}(\sqrt{\bar{\alpha}_t}\mu + 0_n, \bar{\alpha}_t\mathbf{I}_n + (1 - \bar{\alpha}_t)\mathbf{I}_n)$. On simplification, we note that $z_t$ is a sample from $\mathcal{N}(\sqrt{\bar{\alpha}_t}\mu, \mathbf{I}_n)$.

We denote the PDF of $z_t$'s marginal distribution as $q_t(z_t)$.

Since we are using the optimal denoiser, the reverse process PDF at $t$, induced by the optimal denoiser, $p_{*,t}(z_t)$ is the same as the forward process PDF at $t$, which is $q_t(z_t)$.

Here, note that in Sec. 2.1, we denote the reverse process PDF as $p_{\theta,t}$, where the reverse process is governed by the denoiser $\epsilon_\theta$. We replace this notation with $p_{*,t}(z_t)$ as we are using the optimal denoiser.

Therefore, the score function at $t$ is given by $\nabla_{z_t} \log p_{*,t}(z_t) = \nabla_{z_t} \log q_t(z_t)$.

The score function for the Gaussian distribution $q_t(z_t)$ with mean $\sqrt{\bar{\alpha}_t}\mu$ and covariance matrix $\mathbf{I}_n$, *i.e.*, $\nabla_{z_t}(\log q_t(z_t))$ is given by $\sqrt{\bar{\alpha}_t}\mu - z_t$.

Finally, Luo (2022) shows that for the diffusion step $t$, the optimal denoiser can be obtained from the score function using the following expression:

$$\epsilon^*(z_t, t) = -\sqrt{1 - \bar{\alpha}_t}\nabla_{z_t}\log q_t(z_t) \Rightarrow \epsilon^*(z_t, t) = -\sqrt{1 - \bar{\alpha}_t}(\sqrt{\bar{\alpha}_t}\mu - z_t). \tag{27}$$

$\square$

**Lemma 2.** *Suppose **Assumption 2** holds. Consider a $T$-step diffusion process with coefficients $\bar{\alpha}_0, \ldots, \bar{\alpha}_T$ such that $\bar{\alpha}_0 = 1$, $\bar{\alpha}_T = 0$, $\bar{\alpha}_t \in [0, 1]$. The projected posterior mean estimate, $\hat{z}_{0,\mathrm{pr}}(z_t; \epsilon_\theta)$, from the projection step in line 5 of **Algorithm 1** is given by*

$$\hat{z}_{0,\mathrm{pr}}(z_t; \epsilon_\theta) = [I + \gamma(t)A^T A]^{-1}[\mu - \bar{\alpha}_t\mu + \sqrt{\bar{\alpha}_t}z_t + \gamma(t)A^T y],$$

*where the penalty coefficient from **Algorithm 1**, $\gamma(t) > 0 \; \forall \, t \in [1, \ldots, T]$.*

*Proof.* We start with the unconstrained minimization in line 5 of Algorithm 1, given by

$$\hat{z}_{0,\mathrm{pr}}(z_t; \epsilon_\theta) = \arg\min_z \frac{1}{2} \left( \|z - \hat{z}_0(z_t; \epsilon_\theta)\|_2^2 + \gamma(t)\|y - Az\|_2^2 \right).$$

Note that we replaced the penalty function $\Pi(z)$ with $\|y - Az\|_2^2$, as we are required to generate a sample that satisfies the constraint $y = Az$.

Since the objective function is convex with respect to $z$, we obtain the global minimum by setting the gradient with respect to $z$ to 0, *i.e.,*

$$\nabla_z \left( \frac{1}{2} \left( \|z - \hat{z}_0(z_t; \epsilon_\theta)\|_2^2 + \gamma(t)\|y - Az\|_2^2 \right) \right) = 0,$$

$$\nabla_z \left( \frac{1}{2} \left( z^T z - 2z^T \hat{z}_0(z_t; \epsilon_\theta) + \hat{z}_0(z_t; \epsilon_\theta)^T \hat{z}_0(z_t; \epsilon_\theta) \right) \right) + \gamma(t)\nabla_z \left( \frac{1}{2}\|y - Az\|_2^2 \right) = 0,$$

$$z - \hat{z}_0(z_t; \epsilon_\theta) + \gamma(t)\nabla_z \left( \frac{1}{2}\|y - Az\|_2^2 \right) = 0,$$

$$z - \hat{z}_0(z_t; \epsilon_\theta) + \gamma(t)\nabla_z \left( \frac{1}{2} \left( y^T y + z^T A^T Az - 2y^T Az \right) \right) = 0,$$

$$z - \hat{z}_0(z_t; \epsilon_\theta) + \gamma(t) \left( A^T Az - A^T y \right) = 0,$$

$$\left[ \mathbf{I}_n + \gamma(t)A^T A \right] z - \left( \hat{z}_0(z_t; \epsilon_\theta) + \gamma(t)A^T y \right) = 0.$$

Solving this, we obtain the following expression for $\hat{z}_{0,\mathrm{pr}}(z_t; \epsilon_\theta)$:

$$\hat{z}_{0,\mathrm{pr}}(z_t; \epsilon_\theta) = [\mathbf{I}_n + \gamma(t)A^T A]^{-1}(\hat{z}_0(z_t; \epsilon_\theta) + \gamma(t)A^T y).$$

Note that the inverse of $\left[ \mathbf{I}_n + \gamma(t)A^T A \right]$ exists as $A^T A \succ 0$ (from **Assumption 2**) and $\gamma(t) > 0$, which ensures $\left[ \mathbf{I}_n + \gamma(t)A^T A \right] \succ 0$. Further, substituting the expression for $\hat{z}_0(z_t; \epsilon_\theta)$, we obtain

$$\hat{z}_{0,\mathrm{pr}}(z_t; \epsilon_\theta) = [\mathbf{I}_n + \gamma(t)A^T A]^{-1} \left[ \frac{z_t - \sqrt{1 - \bar{\alpha}_t}\epsilon_\theta(z_t, t)}{\sqrt{\bar{\alpha}_t}} + \gamma(t)A^T y \right].$$

Given that $P_{\mathrm{data}} = \mathcal{N}(\mu, \mathbf{I}_n)$, for the $T$-step diffusion process with coefficients $\bar{\alpha}_0, \ldots, \bar{\alpha}_T$, we use the expression for the optimal denoiser $\epsilon^*(z_t, t)$ (check Eq. 27) in place of $\epsilon_\theta(z_t, t)$ to obtain

$$\hat{z}_{0,\mathrm{pr}}(z_t; \epsilon^*) = [\mathbf{I}_n + \gamma(t)A^T A]^{-1} \left[ \frac{z_t + (1 - \bar{\alpha}_t)(\sqrt{\bar{\alpha}_t}\mu - z_t)}{\sqrt{\bar{\alpha}_t}} + \gamma(t)A^T y \right],$$

$$\hat{z}_{0,\mathrm{pr}}(z_t; \epsilon^*) = [\mathbf{I}_n + \gamma(t)A^T A]^{-1} \left[ \frac{z_t + \sqrt{\bar{\alpha}_t}\mu - z_t - \bar{\alpha}_t\sqrt{\bar{\alpha}_t}\mu + \bar{\alpha}_t z_t}{\sqrt{\bar{\alpha}_t}} + \gamma(t)A^T y \right].$$

This can be finally simplified to obtain the expression

$$\hat{z}_{0,\mathrm{pr}}(z_t; \epsilon^*) = [\mathbf{I}_n + \gamma(t)A^T A]^{-1}[\mu - \bar{\alpha}_t\mu + \sqrt{\bar{\alpha}_t}z_t + \gamma(t)A^T y].$$

$\square$

**Lemma 3.** *Suppose **Assumption 2** holds. Consider a $T$-step diffusion process with coefficients $\bar{\alpha}_0, \ldots, \bar{\alpha}_T$ such that $\bar{\alpha}_0 = 1$, $\bar{\alpha}_T = 0$, $\bar{\alpha}_t \in [0, 1]$. If $\bar{\alpha}_t < \bar{\alpha}_{t-1}$ and the penalty coefficients from **Algorithm 1** given by $\gamma(t) > 0 \,\forall\, t \in [1, T]$, the spectral norm of the matrix $K_t$, $\|K_t\|_2$, with $K_t$ as defined in Eq. 16, is less than 1.*

*Proof.* We want to show that

$$\|K_t\|_2 = \left\| \left[ \sqrt{\bar{\alpha}_{t-1}}\sqrt{\bar{\alpha}_t} \left[ \mathbf{I}_n + \gamma(t)A^T A \right]^{-1} + \sqrt{1 - \bar{\alpha}_{t-1}}\sqrt{1 - \bar{\alpha}_t}\mathbf{I}_n \right] \right\|_2 < 1.$$

The spectral norm follows the triangle inequality. Therefore, after simplifying the expression with triangle inequality, we need to show

$$\left\| \sqrt{\bar{\alpha}_{t-1}}\sqrt{\bar{\alpha}_t} \left[ \mathbf{I}_n + \gamma(t)A^T A \right]^{-1} \right\|_2 + \left\| \sqrt{1 - \bar{\alpha}_{t-1}}\sqrt{1 - \bar{\alpha}_t}\mathbf{I}_n \right\|_2 < 1.$$

We note that for $\gamma(t) > 0$, $\left[ \mathbf{I}_n + \gamma(t)A^T A \right] \succ 0$, and therefore $\left[ \mathbf{I}_n + \gamma(t)A^T A \right]^{-1} \succ 0$. Similarly, $\mathbf{I}_n \succ 0$.

Further, we use the identities that if $M \succ 0$, then $\|M\|_2 = \lambda_{\max}(M)$, $\|M^{-1}\|_2 = \frac{1}{\lambda_{\min}(M)}$, and $\|cM\|_2 = |c|\|M\|_2$.

Therefore, $\|\mathbf{I}_n\|_2 = 1$, $\| \left[ \mathbf{I}_n + \gamma(t)A^T A \right]^{-1} \|_2 = \frac{1}{\lambda_{\min}([\mathbf{I}_n + \gamma(t)A^T A])}$. Further, note that $\sqrt{\bar{\alpha}_{t-1}}\sqrt{\bar{\alpha}_t} \geq 0$ and $\sqrt{1-\bar{\alpha}_{t-1}}\sqrt{1-\bar{\alpha}_t} \geq 0$. Substituting these, the inequality simplifies to

$$\frac{\sqrt{\bar{\alpha}_{t-1}}\sqrt{\bar{\alpha}_t}}{\lambda_{\min}([\mathbf{I}_n + \gamma(t)A^T A])} + \sqrt{1-\bar{\alpha}_{t-1}}\sqrt{1-\bar{\alpha}_t} < 1.$$

Therefore, it is sufficient to show that

$$\frac{\sqrt{\bar{\alpha}_{t-1}}\sqrt{\bar{\alpha}_t}}{\lambda_{\min}([\mathbf{I}_n + \gamma(t)A^T A])} < 1 - \sqrt{1-\bar{\alpha}_{t-1}}\sqrt{1-\bar{\alpha}_t}.$$

For any diffusion process with noise coefficients $\bar{\alpha}_0, \ldots, \bar{\alpha}_T$, where $\bar{\alpha}_t > \bar{\alpha}_{t-1} \; \forall \; t \in [1, T]$, **Lemma 5** shows that $\sqrt{\bar{\alpha}_{t-1}}\sqrt{\bar{\alpha}_t} \leq 1 - \sqrt{1-\bar{\alpha}_{t-1}}\sqrt{1-\bar{\alpha}_t}$. Therefore, it is sufficient to show that $\lambda_{\min}\left( \left[ \mathbf{I}_n + \gamma(t)A^T A \right] \right) > 1$.

To proceed further, we use the Weyl's inequality Horn & Johnson (2012), which states that for any two real symmetric matrices $P \in \mathbb{R}^{n \times n}$ and $Q \in \mathbb{R}^{n \times n}$, if the eigenvalues are represented as $\lambda_{\max}(P) = \lambda_1(P) >= \lambda_2(P) \cdots >= \lambda_n(P) = \lambda_{\min}(P)$, and $\lambda_{\max}(Q) = \lambda_1(Q) >= \lambda_2(Q) \cdots >= \lambda_n(Q) = \lambda_{\min}(Q)$, then we have the following inequality:

$$\lambda_i(P) + \lambda_j(Q) \leq \lambda_{i+j-n}(P + Q). \tag{28}$$

For $i = j = n$, we have $\lambda_{\min}(P) + \lambda_{\min}(Q) \leq \lambda_{\min}(P + Q)$.

For $P = \mathbf{I}_n$ and $Q = \gamma(t)A^T A$ with $\gamma(t) > 0$, this inequality can be exploited as both these matrices are real and symmetric. Therefore, we have

$$\lambda_{\min}\left( \left[ \mathbf{I}_n + \gamma(t)A^T A \right] \right) \geq \lambda_{\min}(\mathbf{I}_n) + \lambda_{\min}(\gamma(t)A^T A), \tag{29}$$

$$\lambda_{\min}\left( \left[ \mathbf{I}_n + \gamma(t)A^T A \right] \right) \geq 1 + \gamma(t)\lambda_{\min}(A^T A). \tag{30}$$

Note that now it is sufficient to show $1 + \gamma(t)\lambda_{\min}(A^T A) > 1$. For $\gamma(t) > 0$, this inequality holds true as $\lambda_{\min}(A^T A) > 0$ ($A^T A \succ 0$). Therefore,

$$\left\| \left[ \sqrt{\bar{\alpha}_{t-1}}\sqrt{\bar{\alpha}_t} \left[ \mathbf{I}_n + \gamma(t)A^T A \right]^{-1} + \sqrt{1-\bar{\alpha}_{t-1}}\sqrt{1-\bar{\alpha}_t}\mathbf{I}_n \right] \right\|_2 < 1.$$

$\square$

**Lemma 4.** *Suppose **Assumption 2** holds. Consider a $T$-step diffusion process with coefficients $\bar{\alpha}_0, \ldots, \bar{\alpha}_T$ such that $\bar{\alpha}_0 = 1$, $\bar{\alpha}_T = 0$, $\bar{\alpha}_t \in [0, 1]$. If $\bar{\alpha}_t < \bar{\alpha}_{t-1}$ and the penalty coefficients from **Algorithm 1** given by $\gamma(t) > 0 \; \forall \; t \in [1, T]$, $\|K_1\|_2$ with $K_t$ as defined in Eq. 16 is given by*

$$\|K_1\|_2 \leq \frac{\sqrt{\bar{\alpha}_1}}{1 + \gamma(1)\lambda_{\min}(A^T A)}. \tag{31}$$

*Proof.* We want to find an upper bound for

$$\|K_t\|_2 = \left\| \left[ \sqrt{\bar{\alpha}_{t-1}}\sqrt{\bar{\alpha}_t} \left[ \mathbf{I}_n + \gamma(t)A^T A \right]^{-1} + \sqrt{1-\bar{\alpha}_{t-1}}\sqrt{1-\bar{\alpha}_t}\mathbf{I}_n \right] \right\|_2.$$

Applying the triangle inequality for spectral norm, we get

$$\|K_t\|_2 \leq \left\| \sqrt{\bar{\alpha}_{t-1}}\sqrt{\bar{\alpha}_t} \left[ \mathbf{I}_n + \gamma(t)A^T A \right]^{-1} \right\|_2 + \left\| \sqrt{1-\bar{\alpha}_{t-1}}\sqrt{1-\bar{\alpha}_t}\mathbf{I}_n \right\|_2.$$

We use the same simplifications shown in Lemma 3 to obtain

$$\|K_t\|_2 \leq \frac{\sqrt{\bar{\alpha}_{t-1}}\sqrt{\bar{\alpha}_t}}{\lambda_{\min}([\mathbf{I}_n + \gamma(t)A^T A])} + \sqrt{1-\bar{\alpha}_{t-1}}\sqrt{1-\bar{\alpha}_t}.$$

For $t = 1$, we know that $\bar{\alpha}_{t-1} = \bar{\alpha}_0 = 1$. Therefore, we obtain

$$\|K_1\|_2 \leq \frac{\sqrt{\bar{\alpha}_1}}{\lambda_{\min}([\mathbf{I}_n + \gamma(1)A^T A])}.$$

Further, the denominator can be lower bounded using Weyl's inequality, as shown in Eq. 30. Therefore, we obtain

$$\|K_1\|_2 \leq \frac{\sqrt{\bar{\alpha}_1}}{\lambda_{\min}([\mathbf{I}_n + \gamma(1)A^T A])} \leq \frac{\sqrt{\bar{\alpha}_1}}{1 + \gamma(1)\lambda_{\min}(A^T A)}.$$

Hence, we have shown that

$$\|K_1\|_2 \leq \frac{\sqrt{\bar{\alpha}_1}}{1 + \gamma(1)\lambda_{\min}(A^T A)}.$$

$\square$

**Lemma 5.** *For any $T$-step diffusion process with coefficients $\bar{\alpha}_0, \ldots, \bar{\alpha}_T$ such that $\bar{\alpha}_0 = 1$, $\bar{\alpha}_T = 0$, $\bar{\alpha}_t \in [0, 1] \ \forall t \in [1, T]$, if $\bar{\alpha}_t < \bar{\alpha}_{t-1}$, then*

$$\sqrt{\bar{\alpha}_{t-1}}\sqrt{\bar{\alpha}_t} < 1 - \sqrt{1 - \bar{\alpha}_{t-1}}\sqrt{1 - \bar{\alpha}_t}.$$

*Proof.* Squaring on both sides, we get

$$\bar{\alpha}_{t-1}\bar{\alpha}_t < 1 + (1 - \bar{\alpha}_{t-1})(1 - \bar{\alpha}_t) - 2\sqrt{1 - \bar{\alpha}_{t-1}}\sqrt{1 - \bar{\alpha}_t}.$$

After further simplification, we have to show

$$\bar{\alpha}_{t-1}\bar{\alpha}_t < (1 - \bar{\alpha}_t) + (1 - \bar{\alpha}_{t-1}) + \bar{\alpha}_{t-1}\bar{\alpha}_t - 2\sqrt{1 - \bar{\alpha}_{t-1}}\sqrt{1 - \bar{\alpha}_t},$$
$$0 < (1 - \bar{\alpha}_t) + (1 - \bar{\alpha}_{t-1}) - 2\sqrt{1 - \bar{\alpha}_{t-1}}\sqrt{1 - \bar{\alpha}_t},$$
$$0 < (\sqrt{1 - \bar{\alpha}_{t-1}} - \sqrt{1 - \bar{\alpha}_t})^2.$$

Since $\bar{\alpha}_t \neq \bar{\alpha}_{t-1}$, we know that $\sqrt{1 - \bar{\alpha}_{t-1}} \neq \sqrt{1 - \bar{\alpha}_t}$. Therefore $(\sqrt{1 - \bar{\alpha}_{t-1}} - \sqrt{1 - \bar{\alpha}_t})^2 > 0$. Therefore, we conclude that

$$\sqrt{\bar{\alpha}_{t-1}}\sqrt{\bar{\alpha}_t} < 1 - \sqrt{1 - \bar{\alpha}_{t-1}}\sqrt{1 - \bar{\alpha}_t}.$$

Note that this clearly holds for the edge case $t = 1$, where we have $\sqrt{\bar{\alpha}_1} < 1$, and for $t = T$, where we have $0 < 1 - \sqrt{1 - \bar{\alpha}_{T-1}}$. For the choices of $\bar{\alpha}_0, \ldots, \bar{\alpha}_T$, these clearly hold true. $\square$

**Lemma 6.** *Suppose **Assumption 2** holds. Consider a $T$-step diffusion process with coefficients $\bar{\alpha}_0, \ldots, \bar{\alpha}_T$ such that $\bar{\alpha}_0 = 1$, $\bar{\alpha}_T = 0$, $\bar{\alpha}_t \in [0, 1] \ \forall \ t \in [1, T]$. For the penalty coefficients from **Algorithm 1** given by $\gamma(t) > \frac{2}{\lambda_{\min}(A^T A)}$, $\|D_t\|_2$, with $D_t$ as defined in Eq. 19, is less than 1.*

*Proof.* Note that the matrix $D_t$ is given by,

$$D_t = \gamma(t)\sqrt{\bar{\alpha}_{t-1}} \left[\mathbf{I}_n + \gamma(t)A^T A\right]^{-1} A^T A - \mathbf{I}_n.$$

Using the matrix inversion identity, $(AB)^{-1} = B^{-1}A^{-1}$, we rewrite $D_t$ as follows.

$$D_t = \gamma(t)\sqrt{\bar{\alpha}_{t-1}} \left[\left(A^T A\right)^{-1} \left[\mathbf{I}_n + \gamma(t)A^T A\right]\right]^{-1} - \mathbf{I}_n.$$

$$D_t = \sqrt{\bar{\alpha}_{t-1}} \left[\frac{\left(A^T A\right)^{-1}}{\gamma(t)} \left[\mathbf{I}_n + \gamma(t)A^T A\right]\right]^{-1} - \mathbf{I}_n.$$

$$D_t = \sqrt{\bar{\alpha}_{t-1}} \left[\frac{\left(A^T A\right)^{-1}}{\gamma(t)} + \mathbf{I}_n\right]^{-1} - \mathbf{I}_n.$$

We observe that the choice of $\gamma(t)$ is greater than 0. More precisely, $\gamma(t) > \frac{2}{\lambda_{\min}(A^T A)}$. Now, if $\| - \frac{(A^T A)^{-1}}{\gamma(t)}\|_2 < 1$, then we can apply the Neumann's series for matrix inversion, which states that if $\|M\|_2 < 1$, then

$$[\mathbf{I}_n - M]^{-1} = \sum_{i=0}^{\infty} M^i. \tag{32}$$

First, note that $\|-A\|_2 = \|A\|_2$. Therefore, $\|-\frac{(A^TA)^{-1}}{\gamma(t)}\|_2 = \|\frac{(A^TA)^{-1}}{\gamma(t)}\|_2$ for $\gamma(t) > 0$. From the theorem statement, $\gamma(t) > 0$.

Additionally, we know that $\|\frac{(A^TA)^{-1}}{\gamma(t)}\|_2 = \lambda_{\max}\left(\frac{(A^TA)^{-1}}{\gamma(t)}\right) = \frac{1}{\gamma(t)\lambda_{\min}(A^TA)}$.

Therefore, it is enough to show that $\frac{1}{\gamma(t)\lambda_{\min}(A^TA)} < 1$ to apply the Neumann's series.

However, we know that $\gamma(t) > \frac{2}{\lambda_{\min}((A^TA)^{-1})}$. Therefore, we observe that $\frac{1}{\gamma(t)\lambda_{\min}(A^TA)} < \frac{1}{2} < 1$.

Thus, we have shown that $\|\frac{(A^TA)^{-1}}{\gamma(t)}\|_2 < 1$. Therefore, using Eq. 32, we get

$$\left[\mathbf{I}_n - \left(-\frac{(A^TA)^{-1}}{\gamma(t)}\right)\right]^{-1} = \sum_{i=0}^{\infty}\left(\frac{(-A^TA)^{-1}}{\gamma(t)}\right)^i = \mathbf{I}_n + \sum_{i=1}^{\infty}\left(\frac{(-1)^i(A^TA)^{-i}}{\gamma(t)^i}\right). \tag{33}$$

The last equality stems from the fact that for any matrix $M \in \mathbb{R}^{n\times n}$, $M^0 = \mathbf{I}_n$. Substituting this expression for the second term in $D_t$, we get,

$$D_t = \sqrt{\bar{\alpha}_{t-1}}\left(\sum_{i=1}^{\infty}\left(\frac{(-1)^i(A^TA)^{-i}}{\gamma(t)^i}\right)\right) + \sqrt{\bar{\alpha}_{t-1}}\mathbf{I}_n - \mathbf{I}_n.$$

On further simplification, we have

$$D_t = \sqrt{\bar{\alpha}_{t-1}}\left(\sum_{i=1}^{\infty}\left(\frac{(-1)^i(A^TA)^{-i}}{\gamma(t)^i}\right)\right) - \left(1 - \sqrt{\bar{\alpha}_{t-1}}\right)\mathbf{I}_n.$$

Computing the spectral norm and using the triangle inequality, we get

$$\|D_t\|_2 = \left\|\sqrt{\bar{\alpha}_{t-1}}\left(\sum_{i=1}^{\infty}\left(\frac{(-1)^i(A^TA)^{-i}}{\gamma(t)^i}\right)\right) - \left(1 - \sqrt{\bar{\alpha}_{t-1}}\right)\mathbf{I}_n\right\|_2,$$

$$\leq \sqrt{\bar{\alpha}_{t-1}}\left(\sum_{i=1}^{\infty}\left\|\frac{(-1)^i(A^TA)^{-i}}{\gamma(t)^i}\right\|_2\right) + \left\|(1 - \sqrt{\bar{\alpha}_{t-1}})\mathbf{I}_n\right\|_2.$$

The inequality arises from the triangle inequality for spectral norms. Note that each of the matrices within the summation is either positive definite or negative definite, and the spectral norms of all these matrices can be represented as $\left\|\frac{(A^TA)^{-i}}{\gamma(t)^i}\right\|_2$. Therefore, we get

$$\|D_t\|_2 \leq \sqrt{\bar{\alpha}_{t-1}}\left(\sum_{i=1}^{\infty}\left\|\frac{(A^TA)^{-i}}{\gamma(t)^i}\right\|_2\right) + \left(1 - \sqrt{\bar{\alpha}_{t-1}}\right).$$

Using the inequality $\|MN\|_2 \leq \|M\|_2\|N\|_2$ multiple times, we get the following:

$$\left\|\frac{(A^TA)^{-i}}{\gamma(t)^i}\right\|_2 \leq \frac{1}{\gamma(t)^i}\left(\|(A^TA)^{-1}\|_2\right)^i.$$

Additionally, for the above equation, we used $\|cM\|_2 = |c|\|M\|_2$. Here, $c$ is $\gamma(t)$, which is greater than 0. Since $A^TA \succ 0$, we have $\left\|(A^TA)^{-1}\right\|_2 = \frac{1}{\lambda_{\min}(A^TA)}$. Therefore, we have the following inequality:

$$\left\|\frac{(A^TA)^{-i}}{\gamma(t)^i}\right\|_2 \leq \frac{1}{\gamma(t)^i}\left(\frac{1}{\lambda_{\min}(A^TA)}\right)^i = \frac{1}{(\gamma(t)\lambda_{\min}(A^TA))^i}.$$

Using this to upper bound $\|D_t\|_2$, we get

$$\|D_t\|_2 \leq \sqrt{\bar{\alpha}_{t-1}}\left(\sum_{i=1}^{\infty}\left(\frac{1}{(\gamma(t)\lambda_{\min}(A^TA))^i}\right)\right) + \left(1 - \sqrt{\bar{\alpha}_{t-1}}\right).$$

Finally, the summation of an infinite geometric series of the form $a + a^2 + \ldots$, where $a < 1$ is $\frac{a}{1-a}$. Here, note that we have $\gamma(t) > \frac{1}{\lambda_{\min}(A^T A)}$. Therefore, $\frac{1}{\gamma(t)\lambda_{\min}(A^T A)} < 1$. Therefore, we have,

$$\sum_{i=1}^{\infty} \left( \frac{1}{\gamma(t)^i (\lambda_{\min}(A^T A))^i} \right) = \frac{\frac{1}{\gamma(t)\lambda_{\min}(A^T A)}}{1 - \frac{1}{\gamma(t)\lambda_{\min}(A^T A)}} = \frac{1}{\gamma(t)\lambda_{\min}(A^T A) - 1}.$$

So, we obtain

$$\|D_t\|_2 \leq \frac{\sqrt{\bar{\alpha}_{t-1}}}{\gamma(t)\lambda_{\min}(A^T A) - 1} + (1 - \sqrt{\bar{\alpha}_{t-1}}). \tag{34}$$

Now, for $\|D_t\|_2 < 1$, we need to show

$$\frac{\sqrt{\bar{\alpha}_{t-1}}}{\gamma(t)\lambda_{\min}(A^T A) - 1} + (1 - \sqrt{\bar{\alpha}_{t-1}}) < 1, \text{ or}$$

$$\frac{\sqrt{\bar{\alpha}_{t-1}}}{\gamma(t)\lambda_{\min}(A^T A) - 1} < \sqrt{\bar{\alpha}_{t-1}}.$$

This simplifies to showing $\gamma(t)\lambda_{\min}(A^T A) - 1 > 1$, which is true if $\gamma(t) > \frac{2}{\lambda_{\min}(A^T A)}$. And, from the statement of the lemma, we know that $\gamma(t) > \frac{2}{\lambda_{\min}(A^T A)}$.

Therefore, we have shown that $\|D_t\|_2 < 1$ for $\gamma(t) > \frac{2}{\lambda_{\min}(A^T A)}$. $\qquad\square$

**Lemma 7.** *Suppose **Assumption 2** holds. Consider a $T$-step diffusion process with coefficients $\bar{\alpha}_0, \ldots, \bar{\alpha}_T$ such that $\bar{\alpha}_0 = 1$, $\bar{\alpha}_T = 0$, $\bar{\alpha}_t \in [0,1] \; \forall \; t \in [0,T]$. For the penalty coefficients from **Algorithm 1** given by $\gamma(1) > \frac{2}{\lambda_{\min}(A^T A)}$, $\|D_1\|_2$, with $D_t$ as defined in Eq. 19, is upper bounded by*

$$\|D_1\|_2 \leq \frac{1}{\gamma(1)\lambda_{\min}(A^T A) - 1}.$$

*Proof.* Note that the matrix $D_t$ is given by,

$$D_t = \gamma(t)\sqrt{\bar{\alpha}_{t-1}} \left[ \mathbf{I}_n + \gamma(t)A^T A \right]^{-1} A^T A - \mathbf{I}_n.$$

From Eq. 34 in Lemma 6, we know that if $\gamma(t) > \frac{1}{\lambda_{\min}(A^T A)}$,

$$\|D_t\|_2 \leq \frac{\sqrt{\bar{\alpha}_{t-1}}}{\gamma(t)\lambda_{\min}(A^T A) - 1} + (1 - \sqrt{\bar{\alpha}_{t-1}}).$$

From the lemma, we know that $\gamma(t) > \frac{2}{\lambda_{\min}(A^T A)}$. Therefore, we use Eq. 34 and substitute for $t = 1$ and $\bar{\alpha}_0 = 1$, we get

$$\|D_1\|_2 \leq \frac{1}{\gamma(1)\lambda_{\min}(A^T A) - 1}.$$

$\qquad\square$

**Lemma 8.** *Suppose **Assumption 2** holds. Consider a $T$-step diffusion process with coefficients $\bar{\alpha}_0, \ldots, \bar{\alpha}_T$ such that $\bar{\alpha}_0 = 1$, $\bar{\alpha}_T = 0$, $\bar{\alpha}_t \in [0,1]$. If $\bar{\alpha}_t < \bar{\alpha}_{t-1} \forall \; t \in [1,T]$ with the penalty coefficients from **Algorithm 1** given by $\gamma(t) > 0$, $\|E_t\|_2 < 1$ where $E_t$ is defined as in Eq. 17.*

*Proof.* We know that the matrix $E_t$ is defined as

$$E_t = \left[ (1 - \bar{\alpha}_t)\sqrt{\bar{\alpha}_{t-1}} \left[ \mathbf{I}_n + \gamma(t)A^T A \right]^{-1} \right].$$

First, we use the identity $\|cM\|_2 = |c|\|M\|_2$, where $c$ is any real number, we need to show

$$(1 - \bar{\alpha}_t)\sqrt{\bar{\alpha}_{t-1}} \left\| \left[ \mathbf{I}_n + \gamma(t)A^T A \right]^{-1} \right\|_2 < 1.$$

Note that $(1 - \bar{\alpha}_t)\sqrt{\bar{\alpha}_{t-1}} \geq 0$. Further, for $\gamma(t) > 0$, $\left[\mathbf{I}_n + \gamma(t)A^T A\right] \succ 0$, and therefore $\left[\mathbf{I}_n + \gamma(t)A^T A\right]^{-1} \succ 0$.

We use the identity that for $M \succ 0$, $\|M^{-1}\|_2 = \frac{1}{\lambda_{\min}(M)}$.

Therefore, $\|\left[\mathbf{I}_n + \gamma(t)A^T A\right]^{-1}\|_2 = \frac{1}{\lambda_{\min}([\mathbf{I}_n + \gamma(t)A^T A])}$. We use this expression to simplify the inequality as

$$\frac{(1 - \bar{\alpha}_t)\sqrt{\bar{\alpha}_{t-1}}}{\lambda_{\min}([\mathbf{I}_n + \gamma(t)A^T A])} < 1.$$

We use to Weyl's inequality (check Eq. 30) to lower bound the denominator and thereby upper bound the left side. Therefore, it is sufficient to show

$$\frac{(1 - \bar{\alpha}_t)\sqrt{\bar{\alpha}_{t-1}}}{1 + \gamma(t)\lambda_{\min}(A^T A)} < 1.$$

We observe that the numerator $(1 - \bar{\alpha}_t)\sqrt{\bar{\alpha}_{t-1}}$ is always less than 1. However, we know that the denominator $1 + \gamma(t)\lambda_{\min}(A^T A)$ is strictly greater than 1 for $\gamma(t) > 0$ as $(A^T A)^{-1}$ exists and $\lambda_{\min}(A^T A) > 0$. Therefore, the left side is always less than 1. This leads to

$$\left\|(1 - \bar{\alpha}_t)\sqrt{\bar{\alpha}_{t-1}}\left[\mathbf{I}_n + \gamma(t)A^T A\right]^{-1}\right\|_2 < 1.$$

$\square$

**Lemma 9.** *Suppose **Assumption 2** holds. Consider a $T$-step diffusion process with coefficients $\bar{\alpha}_0, \ldots, \bar{\alpha}_T$ such that $\bar{\alpha}_0 = 1$, $\bar{\alpha}_T = 0$, $\bar{\alpha}_t \in [0, 1]$, If $\bar{\alpha}_t < \bar{\alpha}_{t-1} \forall t \in [1, T]$ with the penalty coefficients from **Algorithm 1** given by $\gamma(t) > 0$, $\|E_1\|_2$, with $E_t$ defined as in Eq. 17, is upper bounded by*

$$\sigma_{\max}(E_1) \leq \frac{1 - \bar{\alpha}_1}{1 + \gamma(1)\lambda_{\min}(A^T A)}.$$

*Proof.* We know that $E_t$ is given by

$$E_t = \sqrt{\bar{\alpha}_{t-1}}(1 - \bar{\alpha}_t)\left[\mathbf{I}_n + \gamma(t)A^T A\right]^{-1}.$$

We first substitute for $t = 1$ and $\sqrt{\bar{\alpha}_0} = 1$

$$E_1 = (1 - \bar{\alpha}_1)[\mathbf{I}_n + \gamma(1)A^T A]^{-1}.$$

We use the identity $\|cM\|_2 = |c|\|M\|_2$, where $c$ is any real number, to get

$$\|E_1\|_2 = (1 - \bar{\alpha}_1)\left\|[\mathbf{I}_n + \gamma(1)A^T A]^{-1}\right\|_2.$$

Here, note that $(1 - \bar{\alpha}_1) \geq 0$. Similar to Lemma 8, we can rewrite the spectral norm as

$$\|E_1\|_2 = \frac{1 - \bar{\alpha}_1}{\lambda_{\min}([\mathbf{I}_n + \gamma(1)A^T A])}.$$

Again, using Weyl's inequality and performing similar modifications as in Lemma 8, we obtain the following upper bound for the spectral norm

$$\|E_1\|_2 \leq \frac{1 - \bar{\alpha}_1}{1 + \gamma(1)\lambda_{\min}(A^T A)}.$$

$\square$

**Lemma 10.** *Suppose **Assumption 2** holds. Consider a $T$-step diffusion process with coefficients $\bar{\alpha}_0, \ldots, \bar{\alpha}_T$ such that $\bar{\alpha}_0 = 1$, $\bar{\alpha}_T = 0$, $\bar{\alpha}_t \in [0, 1]$. If $\bar{\alpha}_t < \bar{\alpha}_{t-1} \forall t \in [0, T]$, $\|F_t\|_2$, with $F_t$ as defined in Eq. 18, is less than 1. Additionally, $\|F_1\|_2$ is 0.*

*Proof.* Note that $F_t$ is given by the expression,

$$F_t = \sqrt{1 - \bar{\alpha}_{t-1}}\sqrt{1 - \bar{\alpha}_t}\sqrt{\bar{\alpha}_t}\mathbf{I}_n.$$

First, we use the identity $\|cM\|_2 = |c|\|M\|_2$, where $c$ is any real number. Therefore, we need to show

$$\|F_t\|_2 = \sqrt{1 - \bar{\alpha}_{t-1}}\sqrt{1 - \bar{\alpha}_t}\sqrt{\bar{\alpha}_t}\,\|\mathbf{I}_n\|_2 < 1.$$

For the given conditions on $\bar{\alpha}_0, \ldots, \bar{\alpha}_T$, we observe that at least one of the terms in $\sqrt{1 - \bar{\alpha}_{t-1}}\sqrt{1 - \bar{\alpha}_t}\sqrt{\bar{\alpha}_t}$ is always less than 1. Therefore $\|F_t\|_2 < 1$. And, since $\bar{\alpha}_0 = 1$, for $F_1$, we have $\sqrt{1 - \bar{\alpha}_0} = 0$. Therefore, $F_1$ is a null matrix and $\|F_1\|_2 = 0$. □

## B    METRICS

For the FTSD and J-FTSD metrics, we train the time series and condition encoders using the procedure given in Narasimhan et al. (2024). For FTSD, we only train the time series encoder using supervised contrastive loss to maximize the similarity of time series chunks that belong to the same sample. For J-FTSD, we perform contrastive learning training in a CLIP-like manner to maximize the similarity between time series and corresponding paired metadata, as explained in Narasimhan et al. (2024). We use Informer models as the encoders. Additionally, just as in the case of (Paul et al., 2022; Narasimhan et al., 2024), we observe that the approaches corresponding to the lowest values of FD metrics have the lowest TSTR and DTW scores and the highest SSIM scores. This further validates the correctness of the FTSD and J-FTSD metrics used for evaluation.

We sourced the implementations of DTW and SSIM from the public domain. For SSIM, we used 1D uniform filters from SciPY Virtanen et al. (2020). We set the values of $C_1$ and $C_2$ to $1^{-4}$ and $9^{-4}$.

For the constraint violation magnitude, we computed the violation for each constraint, excluding the allowable constraint violation budget.

The mean and standard deviation for the TSTR values are obtained from the results for 3 seeds.

## C    DATASETS

We compared CPS against the existing baselines for six settings - Air Quality, Air Quality Conditional, Traffic, Traffic Conditional, Stocks, and Waveforms. The training and testing splits for the Air Quality and Traffic datasets are taken from Narasimhan et al. (2024). We additionally evaluate the constrained generation approaches on the Stocks and the Waveforms datasets. We used the preprocessing scripts provided by Yoon et al. (2019) for the Stocks dataset. The waveforms dataset was synthetically generated. We generated $64,000$ sinusoidal waveforms of varying amplitudes, phases, and frequencies. The amplitude varies from $0.1$ to $1.0$. The phase varies from $0$ to $2\pi$. The frequency limits were chosen based on the Nyquist criterion. The generators and the GAN models were trained on this dataset. However, for the TSTR metrics, we created a subset of this dataset with $16,000$ samples. All the datasets except the waveforms dataset were standard normalized.

The Air Quality dataset is a multivariate dataset with six channels. The total number of train, val, and test samples are 12166, 1537, and 1525, respectively.

The Traffic dataset is univariate. The total train, val, and test samples are 1604, 200, and 201, respectively.

The Stocks dataset is a multivariate dataset with six channels. The total train, val, and test samples are 2871, 358, and 360, respectively.

The truncated form of the waveforms dataset used for evaluation consists of 13320, 1665, and 1665 train, val, and test samples, respectively.

## D    IMPLEMENTATION

In this section, we will describe the implementation details for our approach, each baseline, trained models, metrics, etc.

### D.1 DIFFUSION MODEL ARCHITECTURE

We utilize the TIME WEAVER-CSDI denoiser for all the diffusion models used in this work. The training hyperparameters and the model parameters are precisely the same as indicated in (Narasimhan et al., 2024). The total number of residual layers is 10 for all the experiments. Further, we used 200 denoising steps with a linear noise schedule for the diffusion process. All the baselines and CPS use the same base diffusion model with the TIME WEAVER-CSDI denoiser backbone.

We use 256 channels in each residual layer, with 16-dimensional vectors representing each channel. The diffusion time step input embedding is a 256-dimensional vector. Further, the metadata encoder has an embedding size of 256 for the conditional case. The metadata encoder has two attention layers with eight attention heads. All our experiments use a learning rate of $10^{-4}$. Our training procedure and the hyperparameters are precisely the same as the values in Narasimhan et al. (2024).

### D.2 CONSTRAINED POSTERIOR SAMPLING IMPLEMENTATION

For the CPS implementation, we use CVXPY Diamond & Boyd (2016). We first implement the constraint violation function with the violation threshold set to 0.005 for all the constraints except the bounds like argmax, argmin, OHLC, and the trend constraint. For example, consider the mean constraint. The constraint violation function for this constraint is implemented as $\max\left(\left|\frac{1}{L}\left(\sum_{u=1}^{L} c(u)\right) - \mu_c\right| - 0.005, 0\right)$, where $L$ is the time series horizon. We do not provide the constraint violation threshold for the bounds. Though the allowable constraint violation threshold is 0.01, we performed the projection step with a constraint violation threshold of 0.005 to ensure that the sample strictly lies within the constraint set. We use the same choice of $\gamma(t) \ \forall t \in [1, T]$ as described in Sec. 3. However, we clip the value of $\gamma(t)$ to $100,000$ after certain denoising steps, as the CVXPY solvers cannot handle extremely high values of $\gamma(t)$. We note that this clipping usually occurs after 150 denoising steps.

### D.3 BASELINE IMPLEMENTATION

This section will explain all the details about the baseline implementations. Specifically, we use two baselines - Constrained Optimization Problem (COP) and Guided DiffTime. We note that both approaches were proposed in (Coletta et al., 2024). However, the implementation of these approaches is not publicly available. Based on the details provided in (Coletta et al., 2024), we have implemented the baselines for comparison against CPS.

#### D.3.1 CONSTRAINED OPTIMIZATION PROBLEM IMPLEMENTATION

The Constrained Optimization Problem, COP, has two variants. These are referred to as COP and COP-FineTuning, respectively. In COP, we perturb a randomly selected sample from the training and validation datasets. In COP-FineTuning, we perturb the sample generated from the TIME WEAVER-CSDI diffusion model.

Note that (Coletta et al., 2024) suggests to extract statistical features to be imposed as distributional constraints. For example, Coletta et al. (2024) suggests extracting autocorrelation features for the stocks dataset. However, since it is practically impossible to list all the statistical features for each dataset to obtain the distributional constraints, Coletta et al. (2024) suggests the use of the critic function from a Wasserstein GAN (Arjovsky et al., 2017). The details of the GAN training are summarized below.

COP has two objectives - maximize the $l_2$ distance from a randomly selected sample from the training and maximize the critic value from a Wasserstein GAN.

Similarly, COP FineTuning has two objectives - minimize the $l_2$ distance from a generated sample and maximize the critic value from a Wasserstein GAN.

We optimize for these objectives while ensuring constraint satisfaction.

As suggested in (Coletta et al., 2024), we use the SLSQP solver from SciPy Virtanen et al. (2020). Unlike (Coletta et al., 2024), which performs piecewise optimization, we note that all the constraints used in our work are global. Therefore, piecewise optimization is very suboptimal. For example, it is

suboptimal to break a time series into chunks and perform optimization for each piece when the goal is to generate a sample with a specific mean value. This is also pointed out in (Coletta et al., 2024). Therefore, we perform COP for the whole time series at once. We consider two budgets - 0.005 and 0.01. This is similar to Coletta et al. (2024). However, unlike their approach, we stop with 0.01 as the allowable constraint violation in our case is 0.01 for all methods.

We used a weight of 0.1 for the critic's objective. We noticed that for different values (1.0,0.1,0.01) of this weight, there was very little change in the DTW and the SSIM metrics.

### D.3.2 CRITIC FUNCTION IMPLEMENTATION

Coletta et al. (2024) suggest using the critic function in a Wasserstein GAN Arjovsky et al. (2017) to enforce realism in the COP approach. Therefore, we used the WaveGAN Donahue et al. (2018) implementation from Alcaraz & Strodthoff (2023). The implementation from Alcaraz & Strodthoff (2023) has the gradient penalty loss, an improved training procedure to enforce the required Lipschitz continuity for the critic function. Additionally, the WaveGAN training with gradient penalty has been implemented Alcaraz & Strodthoff (2023) for generating time series samples for the ECG domain. Therefore, we use their implementation to obtain the critic function for the COP baseline. The number of parameters is adjusted such that the diffusion model and the GAN model have a comparable number of parameters.

Similar to the diffusion model, we used the same architecture and training hyperparameters for all the datasets and experimental settings. Specifically, we trained the WaveGAN model with a learning rate of $10^{-4}$ for all the datasets. The input to the generator is a 48-dimensional random vector. Additionally, we ensured that the total number of parameters was equally distributed between the generator and the discriminator to prevent either of the models from overpowering the other.

### D.3.3 GUIDED DIFFTIME IMPLEMENTATION

We use the same TIME WEAVER-CSDI denoiser as in the case of CPS. For the guidance weight, we experimented with the following weights - $(0.00001, 0.0001, 0.001, 0.01, 0.1, 1.0)$. We chose the best guidance weight based on the constraint violation rate. Note that we used the same guidance weight for all individual constraints. Using PyTorch, we implemented all the constraints mentioned in Sec. 4. Additionally, we augmented the Guided DiffTime approach with the DiffTime algorithm for fixed values. In other words, after each step of denoising followed by guidance update, we enforced the fixed value constraints, as specified in (Coletta et al., 2024). This applies to the values at argmax, argmin, 1, 24, 48, 72, and 96 timestamps.

## E DISCRIMINATIVE SCORE METRIC

In addition to the Frechet Time Series Distance (FTSD), the Joint Frechet Time Series Distance (J-FTSD), and the Train on Synthetic and Test on Real (TSTR) metrics, we provide sample quality comparison based on the Discriminative Score (DS) metric. For this metric, we train a post-hoc time series classification model to distinguish between real and generated time series samples. We use a simple 2-layer LSTM network for the classification task. DS was introduced in (Yoon et al., 2019) as a sample quality metric. Similar to the TSTR metric, we train the classifier on synthesized and real training data. We then report the classification error on the synthesized and real test data. The results are provided in Table 2. Here, note that the best-performing approach, in terms of DS, coincides with the best-performing approach in terms of other sample quality metrics, such as FTSD and TSTR.

| APPROACH | AIR QUALITY | AIR QUALITY CONDITIONAL | TRAFFIC | TRAFFIC CONDITIONAL | STOCKS | WAVEFORMS |
|---|---|---|---|---|---|---|
| GUIDED-DIFFTIME | 0.33±0.02 | 0.22±0.02 | 0.29±0.05 | 0.03±0.02 | 0.38±0.01 | 0.43±0.02 |
| COP | 0.29±0.03 | 0.28±0.01 | 0.41±0.05 | 0.41±0.02 | 0.09±0.04 | 0.44±0.02 |
| COP-FT | 0.31±0.03 | 0.03±0.01 | 0.38±0.07 | **0.01±0.01** | 0.16±0.08 | 0.41±0.03 |
| CPS | **0.06±0.01** | **0.01±0.005** | **0.02±0.01** | **0.01±0.004** | **0.006±0.004** | **0.002±0.001** |

Table 2: **CPS outperforms all the baseline approaches on the Discriminative Score (DS) metric (lower is better).** Here, we show DS averaged over 5 seeds for all the experimental setups shown in Table 1.

## F    EXTENDED RELATED WORKS

### F.1    DIFFUSION MODELS FOR TIME SERIES GENERATION

Time Series-specific tasks like forecasting (Rasul et al., 2021; Yan et al., 2021; Biloš et al., 2023) and imputation (Tashiro et al., 2021; Alcaraz & Strodthoff, 2022; Yuan & Qiao, 2024) have been addressed using conditional DMs as well as guidance-based approaches (Li et al., 2023; Yuan & Qiao, 2024). Alcaraz & Strodthoff (2023) and Narasimhan et al. (2024) have explored conditional time series generation for various domains, such as medical, energy, etc. These works aim to sample from a conditional distribution. However, there are limited prior works in the time series domain that focus on generating constrained samples.

### F.2    CONSTRAINED SAMPLE GENERATION

In many engineering applications, the sample domain can be restricted to certain manifolds. Such problem settings demand any generative modeling approach to synthesize samples that adhere to the constraints that define the manifold. Frerix et al. (2020) propose Variational Autoencoders (VAEs) with additional constraint layers added to the neural network architecture to impose linear inequality constraints of the form $Ax \leq 0$. Liu et al. (2023) and Fishman et al. (2023a;b) propose modifications to the denoising diffusion training process to restrict the generation process to the required constraint sets. More specifically, Liu et al. (2023) introduce Mirror Diffusion Models (MDMs) for convex constraint sets. MDMs are standard denoising DMs trained in the dual or the mirror space of the constraint set. Therefore, by generating in the mirror space and transforming back to the constraint set, we can generate samples from the required constraint set. Fishman et al. (2023a;b) propose a modified forward noising process, such that the intermediate noisy latents of the forward process always adhere to the constraint set. Additionally, these works introduce constraint-specific training modifications, such as clipping the score function to zero at the constraint boundaries. Overall, the constraint-specific training approach suffers from the ability to scale to new constraint sets. Additionally, the constrained time series generation problem does not assume the presence of a constrained manifold from which samples need to be generated. However, the objective is to sample from arbitrary constraint sets defined by combinations of multiple constraints such as mean, argmax, etc.

Christopher et al. (2024) propose Projected Diffusion Models (PDMs), a training-free approach for constrained generation, which involves solving a constrained optimization problem after every denoising step. The constrained optimization step projects the intermediate noisy latents of the reverse sampling process to the constraint set. This is similar to our approach, with a key difference that is highlighted in Appendix G. We compare CPS against PDM and explain the relative advantages of our approach in Appendix G.

Finally, Yuan & Qiao (2024) propose a controlled time series generation approach that is specifically designed for time series imputation. In Appendix G, we modify this approach for constrained time series generation and compare it against CPS.

## G    EXTENDED BASELINE COMPARISONS

In this section, we provide quantitative comparisons between CPS and other approaches, such as PDM (Christopher et al., 2024) and Diffusion-TS (Yuan & Qiao, 2024).

Note that the main difference between PDM and CPS is the projection step. In PDM, the noisy latent corresponding to the step $t - 1$, $z_{t-1}$, is obtained from the noisy latent corresponding to the step $t$, $z_t$, using Eq. 2. Consequently, $z_{t-1}$ is projected to the constraint set by solving a constrained optimization problem. This process is repeated for $T$ denoising steps.

In CPS, we compute the posterior mean estimate $\hat{z}_0(z_t; \epsilon_\theta)$ from $z_t$. Then, we transform $\hat{z}_0(z_t; \epsilon_\theta)$ to the projected posterior mean estimate $\hat{z}_{0,\mathrm{pr}}(z_t; \epsilon_\theta)$ using an unconstrained optimization step (line 5, Algorithm 1). Consequently, we obtain $z_{t-1}$ from $z_t$ and $\hat{z}_{0,\mathrm{pr}}(z_t; \epsilon_\theta)$ using Eq. 2. Table 3 shows the comparison between PDM and CPS for all the real-world datasets used in our experiments. We observe that both approaches provide constraint satisfaction for convex constraints. However, CPS outperforms PDM in terms of sample quality and diversity metrics. Now, we explain the reasons for the superior performance of CPS over PDM.

- **The constraint set is defined for the clean samples and not the intermediate noisy latents of a denoising process.** As the goal is to generate constrained time series samples, it is sufficient if the generated sample $z_0$ belongs to the constraint set. However, **PDM assumes the constraint set for clean samples to be the same for noisy intermediate latents.** By forcing the latents to satisfy the same constraint as $z_0$, PDM eliminates most sample paths $(z_T, \ldots, z_0)$ where $z_0$ alone eventually satisfies the required constraint. This results in poor sample diversity. CPS eliminates this problem by projecting the posterior mean estimate $\hat{z}_0(z_t; \epsilon_\theta)$ and not the noisy intermediate latents. Recall that $\hat{z}_0(z_t; \epsilon_\theta)$ is the expected clean sample with a similar noise level as the constraint set. Furthermore, the projected posterior mean estimate $\hat{z}_{0,\mathrm{pr}}(z_t; \epsilon_\theta)$ is transformed to $z_{t-1}$ using a non-markovian forward noising process. This effectively allows for sample paths where the generated sample $z_0$ alone satisfies the required constraint, and the intermediate noisy latents can be flexible.
- **PDM projection step pushes $z_{t-1}$ off the noise manifold for $t-1$.** In PDM, the projection step, when applied directly to the noisy latent $z_{t-1}$, pushes it out of the noise manifold corresponding to the diffusion step $t-1$. Consequently, a pre-trained denoiser struggles to accurately denoise the projected $z_{t-1}$ as it would be out of the training domain of the denoiser. **This effect is significantly reduced in CPS because our approach does not project $z_{t-1}$.** Instead, CPS projects the expected clean sample $\hat{z}_0(z_t; \epsilon_\theta)$. Consequently, the projected posterior mean estimate $\hat{z}_{0,\mathrm{pr}}(z_t; \epsilon_\theta)$ is transformed into $z_{t-1}$ by using a non-markovian forward noising process (Eq. 2). This ensures that $z_{t-1}$ stays very close to the noise manifold corresponding to the diffusion step $t-1$. Therefore, a pre-trained denoiser can denoise $z_{t-1}$ more accurately in our approach. This ultimately preserves the generated sample quality.

Empirically, the difference in the projection step results in the superior performance of CPS over PDM, providing $7\times$ reduction in the FTSD metric and $4\times$ reduction in the DS metric overall (check Table 3).

| DATASET | APPROACH | FTSD ($\downarrow$) | TSTR ($\downarrow$) | DS ($\downarrow$) | DTW ($\downarrow$) | SSIM ($\uparrow$) | CONSTRAINT VIOLATION RATE ($\downarrow$) | CONSTRAINT VIOLATION MAGNITUDE ($\downarrow$) |
|---|---|---|---|---|---|---|---|---|
| AIR QUALITY | PDM | 0.1503 | 0.205±0.005 | 0.254±0.014 | 2.544±1.96 | 0.342±0.148 | 0.0 | 0.0 |
| | CPS (OURS) | **0.0234** | **0.19±0.003** | **0.06±0.01** | **2.35±1.48** | **0.38±0.15** | 0.0 | 0.0 |
| STOCKS | PDM | 0.0368 | 0.044±0.001 | 0.0147±0.007 | 0.447±1.06 | 0.481±0.309 | 0.0 | 0.0 |
| | CPS (OURS) | **0.0023** | **0.041±0.001** | **0.006±0.004** | **0.20±0.71** | **0.73±0.26** | 0.0 | 0.0 |
| TRAFFIC | PDM | 0.2714 | **0.29±0.008** | 0.1313±0.053 | 3.547±1.34 | 0.249±0.192 | 0.0 | 0.0 |
| | CPS (OURS) | **0.2077** | **0.29±0.001** | **0.02±0.01** | **3.41±1.47** | **0.31±0.20** | 0.0 | 0.0 |

Table 3: **CPS outperforms Projected Diffusion Models (PDM) on sample quality and similarity metrics.** On all the real-world datasets, we note that CPS provides better sample quality metrics than PDM. The experimental setup is the same as in Table 1. Both approaches provide perfect constraint satisfaction as we deal with linear and convex constraints. However, CPS outperforms PDM in the Frechet Time Series Distance (FTSD), Train on Synthetic and Test on Real (TSTR), and the Discriminative Score (DS) metrics. Additionally, CPS provides better similarity scores.

Diffusion-TS (Yuan & Qiao, 2024) proposes a guidance-based approach for time series imputation, where the guidance is obtained from the reconstruction error of the unmasked or the known parts of the time series. We replace the reconstruction error with the constraint violation loss. Table 4 shows the quantitative comparison between CPS and Diffusion-TS for all the real-world datasets. Diffusion-TS struggles to generate samples that adhere to the constraint set. This is because, similar to the Guided-DiffTime baseline, there is no principled projection step that effectively guides the sample generation process towards the constraint set.

Furthermore, we also provide comparisons against the Loss-DiffTime baseline from Coletta et al. (2024). For a fair comparison, we use the same TIME WEAVER-CSDI backbone and train the denoiser with constraints as the condition input. The quantitative comparisons are provided in Table 5. As observed with prior approaches, in the absence of any principled projection step, the Loss-DiffTime approach fails to generate samples that adhere to hard constraints. However, due to the constraint-specific training, Loss-DiffTime performs as good as CPS in terms of sample quality and similarity.

| DATASET | APPROACH | FTSD (↓) | TSTR (↓) | DS (↓) | DTW (↓) | SSIM (↑) | CONSTRAINT VIOLATION RATE (↓) | CONSTRAINT VIOLATION MAGNITUDE (↓) |
|---|---|---|---|---|---|---|---|---|
| AIR QUALITY | DIFFUSION-TS | 0.0473 | **0.185±0.004** | **0.06±0.01** | 2.53±1.96 | **0.39±0.15** | 1.0 | 5.613 |
| | CPS (OURS) | **0.0234** | 0.19±0.003 | **0.06±0.01** | **2.35±1.48** | 0.38±0.15 | **0.0** | **0.0** |
| STOCKS | DIFFUSION-TS | 1.1268 | 0.046±0.001 | 0.19±0.02 | 7.44±6.65 | 0.21±0.19 | 1.0 | 40.5139 |
| | CPS (OURS) | **0.0023** | **0.041±0.001** | **0.006±0.004** | **0.20±0.71** | **0.73±0.26** | **0.0** | **0.0** |
| TRAFFIC | DIFFUSION-TS | 0.4918 | 0.31±0.008 | 0.171±0.017 | 3.82±1.57 | **0.37±0.19** | 1.0 | 0.9743 |
| | CPS (OURS) | **0.2077** | **0.29±0.001** | **0.02±0.01** | **3.41±1.47** | 0.31±0.20 | **0.0** | **0.0** |

Table 4: **Diffusion-TS fails to generate samples that adhere to the required constraint set.** The experimental setup is the same as in Table 1. Note that the constraint violation rate for the Diffusion-TS baseline is always 1.0. Due to the absence of principled projection steps, guidance-based approaches fail to generate constrained samples. Otherwise, note that CPS is as good as or outperforms Diffusion-TS on sample quality and similarity metrics.

| DATASET | APPROACH | FTSD (↓) | TSTR (↓) | DS (↓) | DTW (↓) | SSIM (↑) | CONSTRAINT VIOLATION RATE (↓) | CONSTRAINT VIOLATION MAGNITUDE (↓) |
|---|---|---|---|---|---|---|---|---|
| AIR QUALITY | LOSS DIFFTIME | **0.0137** | **0.187±0.003** | **0.03±0.01** | **2.18±1.48** | **0.43±0.17** | 1.0 | 9.779 |
| | CPS (OURS) | 0.0234 | 0.19±0.003 | 0.06±0.01 | 2.35±1.48 | 0.38±0.15 | **0.0** | **0.0** |
| STOCKS | LOSS DIFFTIME | 0.9897 | 0.045±0.002 | 0.379±0.015 | 7.75±6.05 | 0.23±0.17 | 1.0 | 237.492 |
| | CPS (OURS) | **0.0023** | **0.041±0.001** | **0.006±0.004** | **0.20±0.71** | **0.73±0.26** | **0.0** | **0.0** |
| TRAFFIC | LOSS DIFFTIME | 0.3653 | **0.29±0.01** | 0.113±0.039 | **3.15±1.34** | 0.29±0.22 | 1.0 | 2.993 |
| | CPS (OURS) | **0.2077** | **0.29±0.001** | **0.02±0.01** | 3.41±1.47 | **0.31±0.20** | **0.0** | **0.0** |

Table 5: **Despite constraint-specific training, Loss-DiffTime struggles to generate samples that adhere to the required constraint set.** Note that Loss-DiffTime performs better than CPS on the sample quality and similarity metrics for the air quality dataset. However, due to the absence of projection steps, Loss-DiffTime fails to generate samples that adhere to hard constraints.

# H   GENERAL CONSTRAINTS EXPERIMENTS

We extended our experimental setup to generic constraints for the stocks dataset. Specifically, we imposed the Autocorrelation Function (ACF) at a specific lag as an equality constraint with an acceptable tolerance of 0.01. ACF at a specific lag $l$ for a univariate time series X of horizon $L$ is given by,

$$ACF(X) = \frac{1}{(L-l)\sigma^2} \sum_{u=1}^{L-l} (X(u) - \mu)(X(u+l) - \mu), \tag{35}$$

where $\mu = \mathbb{E}(X)$ and $\sigma^2 = \mathbb{E}[(X - \mu)^2]$. Note that $\mu$ and $\sigma$ are not fixed. Along with the ACF equality constraint, we pose the OHLC constraint for the stocks dataset. We provide the results of this experiment in Table 6. We chose ACF as it is one of the most popularly used techniques to extract the most relevant lag features for downstream tasks like forecasting.

| APPROACH | FTSD (↓) | DTW (↓) | SSIM (↑) | CONSTRAINT VIOLATION RATE (↓) | CONSTRAINT VIOLATION MAGNITUDE (↓) |
|---|---|---|---|---|---|
| GUIDED-DIFFTIME | 1.4678 | 15.06±11.92 | 0.09±0.06 | 1.0 | 284.58 |
| COP | 2.1949 | 72.11±35.97 | 0.07±0.11 | 0.41 | 0.9045 |
| CPS (OURS) | **0.0014** | **0.11±0.10** | **0.88±0.11** | **0.29** | **0.01** |

Table 6: **CPS outperforms baselines for OHLC and autocorrelation function value constraints.** Here, we use the stocks dataset and impose the Autocorrelation Function (ACF) value for a specified lag of 12 timestamps as a constraint along with the OHLC constraint. CPS outperforms all the baselines in terms of sample quality, similarity, and constraint satisfaction metrics.

Note that out of all approaches, CPS provides the least constraint violation rate and constraint violation magnitude. Additionally, even though the projection step (line 5, Algorithm 1) does not lead

to the optimal solution (as the autocorrelation function is non-convex in the sample domain), CPS's sample quality is much better than the baselines. This is due to the iterated projection and denoising operations, which significantly reduce the adverse effects of the projection step.

## I  CHOICE OF $\gamma(t)$

$\gamma(1), \ldots, \gamma(T)$ refer to the penalty coefficients in Algorithm 1. Our choice of $\gamma(t)$ can take any functional form as long as $\gamma(t)$ is a strictly decreasing function of $t$ and $\gamma(t) \to \infty$ as $t \to 1$. This is to ensure constraint satisfaction for convex constraint sets. In practice, we clip $\gamma(t)$ to a very large value, such as $10^5$, when performing the final denoising steps. Our current choice of $\gamma(t)$ decreases exponentially with $t$. As the functional form does not matter, in practical implementation, we experimented with linearly and quadratically decreasing values of $\gamma(t)$, with a very high value ($10^5$) for $t = 1$. We noted that the choice of $\gamma(t)$ has very little effect on the sample quality of the generated samples. In Table 7, we observe that the different choices of $\gamma(t)$ have effects only on the third decimal of the FTSD metric for all the real-world datasets used in our experiments.

| CHOICE OF $\gamma(t)$ | AIR QUALITY | TRAFFIC | STOCKS |
|---|---|---|---|
| LINEAR | **0.0222** | 0.2053 | **0.0013** |
| QUADRATIC | 0.0226 | **0.2027** | 0.0016 |
| EXPONENTIAL | 0.0234 | 0.2077 | 0.0023 |

Table 7: **Different choices of $\gamma(t)$ provide similar sample quality metrics.** Here, we report the FTSD score as the sample quality metric. Note that the effect of different choices of $\gamma(t)$ is only reflected in the third decimal and is insignificant.

## J  INFERENCE TIME RESULTS

We evaluated our approach for time series samples up to 576 dimensions (e.g., the air quality and the stocks dataset). We have provided the inference time taken to generate samples with up to 66 and 450 constraints for the air quality and the stocks datasets in Table 8. First, we note that the inference latency for CPS is very similar to PDM (Christopher et al., 2024), as both approaches involve projection steps after each denoising step. We observe that for univariate datasets, like the traffic dataset, the inference latency for CPS is less than that of Guided-DiffTime. Note that Guided-DiffTime requires backpropagation through the denoiser network. However, for multivariate datasets like the air quality and the stocks dataset, the inference time for CPS is roughly $2\times$ more than the inference time for Gudided-DiffTime. However, Guided-DiffTime has poor sample quality and very low constraint satisfaction rates. For all the datasets, COP has the least inference time. However, COP also suffers heavily from poor sample quality.

| APPROACH | AIR QUALITY | TRAFFIC | STOCKS |
|---|---|---|---|
| GUIDED-DIFFTIME | 14.76±0.36 s | 11.61±0.39 s | 15.24±0.43 s |
| COP-FT | **8.5±3.72 s** | **1.27±0.45 s** | **11±4.47 s** |
| CPS (OURS) | 31.49±0.64 s | 6.99±0.52 s | 35.22±2.01 s |

Table 8: **The projection step in CPS increases the sampling time.** Here, we present the average inference time taken to generate a single sample for all the real-world datasets used in our experiments. The results are shown in seconds, and the inference time is averaged over 10 runs. Though the inference time for COP-FT is very low, the generated samples have poor sample quality.

Furthermore, we note that there are multiple ways to reduce the inference time for CPS, such as:

- Capping the number of update steps in each projection operation (line 5 of Algorithm 1) during the initial denoising steps when the signal-to-noise ratio is very low.
- The projection operation (line 5 of Algorithm 1) need not be performed for every denoising step. Consequently, we can develop principled methods to identify the denoising steps where projection is required based on constraint violation.

## K    ADDITIONAL QUALITATIVE RESULTS

In this section, we provide additional qualitative results for the real-world datasets used in our experiments.

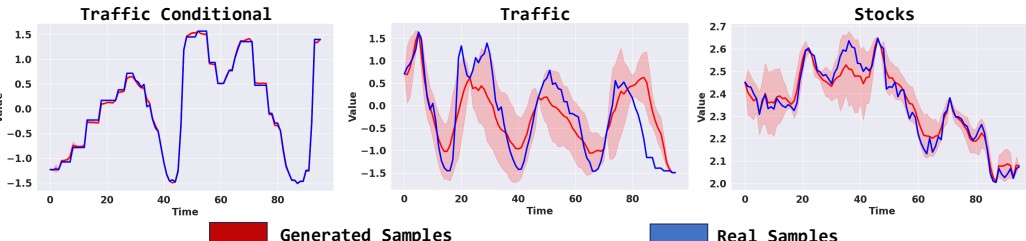

Figure 7: **CPS provides high-fidelity synthetic time series samples that adhere to the required constraints and track the real time series data.** Here, we show multiple generations (10) of the same qualitative examples shown in Fig. 6. Note that the traffic conditional setting has additional conditions or metadata as input. From Narasimhan et al. (2024), we note that metadata can be used to synthesize accurate time series. In addition to metadata, when constraints are imposed, the variance in the generated data significantly reduces (left image). However, the traffic setting without metadata (middle image) has high variance and broadly follows the trend of the ground truth time series sample. Observe that the constraint satisfaction for fixed point constraints is visible through zero variance at timestamps 1,24,48,72 and 96. The stocks setting also has no metadata input. However, due to the large number of constraints (450), the synthesized time series tracks the ground truth sample very closely (right image).

