# OpenReview forum: "Constrained Posterior Sampling: Time Series Generation with Hard Constraints"
_ICLR.cc/2025/Conference — Submitted to ICLR 2025_

### Official Review · Reviewer_HGVu · 2024-10-31

**Soundness:** 4
**Presentation:** 3
**Contribution:** 3
**Rating:** 8
**Confidence:** 4

**Summary:**

In this paper, the authors proposed a diffusion-based sampling algorithm - Constrained Posterior Sampling (CPS), to generate time series data meeting certain constraints. The basic idea is solving a regularized optimization task in every denoising step and the penalty term  will be strengthened such that the resulting data fall within the constrained set. Overall, I think this paper is well-written, has clear presentation and motivation. I will consider raising my score if the authors handle my questions properly.

**Strengths:**

**1** This paper has extensive experiments to show the effectiveness of their method.

**2** A theoretical result is provided for validating the effectiveness under a linear constraint case.

**3** Proper baselines are chosen for comparison and comprehensive literature review is provided.

**4** The proposed method basically provides some interesting ideas of handling constrained generation in diffusion models. I think the method can be extended to other scenarios.

**Weaknesses:**

**1** In each step, a constrained optimization task needs to be solved, which can be computationally intensive. Additionally, the current framework is more like as a general framework without imposing specific restrictions on the constraints. It would be beneficial to develop a universal algorithm that aims to minimize the constrained optimization for a general class of constraints, even if the optimal solution is not achieved.

**2**  The fidelity of generated data is not properly evaluated in this paper. I suggest using classification method to for fidelity evaluation. That is using a classifier to discriminate real and synthetic data and use the testing error as fidelity metric.

**Questions:**

**Q1** Page 5, Lines 240-244 "Our key intuition is to design $\gamma(t)$ as a strictly **decreasing** function of $t$ that takes small values for the initial denoising steps (t close to T) and tends to $\infty$ for the final denoising steps". Does the function form of $gamma(t)$ affect the result? Is it possible to select others?

**Q2** The main contribution is the adaptive projection step that solves the penalized optimization task. From my understanding, this is more like a soft constraint and it becomes a hard one when $t=1$. However, in practice, how to choose $T$?

**Q3** In every denoising step, the protection step solves a regularized optimization task. However, the optimization problem might be difficult if some non-linear constraints are chosen. Is it guaranteed that the optimal or local optimal solution can be obtained?

**Q4** In Theorem 2, what is design parameter k? It seems missing in the algorithm.

---

> ### Author Response · Authors · 2024-11-21
> **Rebuttal by Authors**
>
> We thank the reviewer for taking the time to provide a thoughtful and thorough review of the paper. The reviewer recognizes the generality of the approach as it can be applied to any domain. Additionally, the reviewer acknowledges the extensive comparisons with correct baselines.
>
> # Addressing the weaknesses pointed out by the reviewer.
>
> ## On the extension to a general class of constraints (Weakness 1)
> **CPS can be extended to non-convex constraint sets as well**. As pointed out by the reviewer, the projection step will not result in the global optimum. Additionally, as hinted by the reviewer, our projection step can be swapped with a constrained optimization step for a general class of constraints without any modification to the rest of the algorithm.
>
> To emphasize the ability to extend to a general class of constraints, we showcase CPS’s performance for non-convex constraints, specifically the equality constraints based on the Autocorrelation function (ACF) value for a specified lag (check Table 6 and Sec. H in the Appendix, page 31).
>
> Given a finite number of optimization steps for each projection operation, it is not possible to guarantee constraint satisfaction for all the baselines, including CPS. However, **we observed that CPS provides the best sample quality and the lowest constraint violation metrics**.
>
> **This is attributed to the iterating projection and denoising process**, which pushes the generated sample very close to the required constraint set while simultaneously minimizing the adverse effects of projection through sequential denoising.
>
> ## On the evaluation of fidelity (Weakness 2)
> We relied on the same fidelity metrics used in prior works [1,2]. However, we agree with the reviewer’s proposed evaluation metric. **In [3], this is referred to as the discriminative score (DS)**, where a 2-layer recurrent neural network (RNN) is trained to perform binary classification between the real and the generated time series. Consequently, the classification error on a heldout test set is reported as DS. **Lower values of DS represent poor classification accuracy, indicating high sample fidelity**. The DS values for all the experimental setups in Table 1 are provided in Table 2 (Sec. E, page 28) in the Appendix. **Overall, CPS conveniently outperforms the baseline approaches even with respect to DS**.

---

> > ### Author Response · Authors · 2024-11-21
> > **Rebuttal by Authors**
> >
> > # Response to the questions from the reviewer
> >
> > ## On the choice of $\gamma(t)$ (Question 1)
> >
> > **The functional form of $\gamma(t)$ generally does not matter as long as $\gamma(t)$ is strictly decreasing and tends to infinity for the final denoising step (t=1)**.
> >
> > The strictly decreasing property is required to ensure that the denoising process is progressively steered towards the constraint set as the diffusion step $t$ goes from $T$ to $1$ during the sample generation process.
> >
> > $\gamma(1) = \infty$ is required to ensure constraint satisfaction, at least for linear and convex constraint sets.
> >
> > CPS currently uses an exponentially decreasing function of $t$, based on the noise coefficients, for the penalty coefficient $\gamma(t)$.
> >
> > During implementation, we restrict $\gamma(t)$ to some large value ($10^5$) towards the final denoising steps. To test if the functional form of $\gamma(t)$ matters, we experimented with linearly and quadratically decreasing functions of $t$ and obtained very similar values for the Frechet Time Series Distance (FTSD) metric with only a difference in the third decimal. The results are provided in Table 7 (Sec. I, page 31) in the Appendix. Therefore, the functional form of $\gamma(t)$ generally does not matter, provided $\gamma(t)$ is a strictly decreasing function of $t$ and $\gamma(1) \rightarrow \infty$ for $t=1$.
> >
> > ## On the choice of $T$ (Question 2)
> > **The reviewer is correct; the projection step does not enforce hard constraints during the initial denoising steps $(t \sim T)$. It becomes a hard constraint for $t=1$.**
> >
> > The typical choice of $T$ for diffusion models in the time series domain is 50 [4]. Taking inspiration from Theorem 2, we note that the terminal error, i,e., the $l_2$ distance of the generated sample from the true solution $x^*$, depends on the initialization error for a finite $T$ (Eq. 26, Sec A.2 in the Appendix, lines 951-956, proof of Theorem 2). The initialization error exponentially decays with respect to $T$. So, it is beneficial to use larger values of T. In our experiments, we found $T=200$ steps (4X more than usual) to be good enough for all datasets.
> >
> > However, we note that there are multiple ways to reduce the inference time taken to generate samples. We refer the reviewer to Sec. J of the Appendix (page 32), where we have provided the inference time taken for sample generation for all our real-world datasets (check Table 8). Additionally, we propose multiple approaches to reduce the inference time for CPS, such as:
> > 1. Reducing the number of update steps in each projection operation (line 5 of Algorithm 1) during the initial denoising steps where the signal-to-noise ratio is very low.
> > 2. The projection operation (line 5 of Algorithm 1) need not be performed for every denoising step. Consequently, we can develop principled methods to identify the denoising steps where projection is required based on constraint violation.

---

> ### Author Response · Authors · 2024-11-21
> **Rebuttal by Authors**
>
> ## On the extension to non-linear constraints (Question 3)
>
> We agree with the reviewer that the objective function in the projection step (line 5,  Algorithm 1) is non-convex for arbitrary constraint classes. Consequently, the projection step may not result in the global optimum. Additionally, it is not possible to guarantee constraint satisfaction for non-convex constraints.
>
> We request the reviewer to check Sec. H of the Appendix, where we have added new experiments with a general class of constraints. Specifically, we imposed the Autocorrelation Function (ACF) value at a particular lag as an equality constraint for the stocks dataset.
>
> We observe that, even though the projection step does not result in the global optimum, the iterating projection and denoising process pushes the generated sample to the constraint set. Additionally, the sequential denoising process removes the adversarial artifacts caused by the projection step. **This provides very low constraint violation metrics for CPS, when compared to other baselines, while maintaining high sample quality (check Table 6 in the Appendix, page 31).**
>
> ## Clarification regarding the design parameter $k$ in the convergence analysis (Question 4)
>
> **In Theorem 2, for ease of analysis, we consider $k$ as a design parameter that affects the penalty coefficient $\gamma(t)$ in a linear fashion $\left(\gamma(t) = \frac{2k(T-t+1)}{\lambda\_\mathrm{min}(A^TA)}\right)$.** Note that we have considered $\gamma(t)$ as a linearly decreasing function of $t$.
>
> Here, we observe that for a finite number of denoising steps $T$ and a finite value of $k$, CPS does not converge to the true solution $x^*$, and the upper bound on the terminal error is given in Eq. 26 (Sec A.2 of the Appendix, lines 951-956). However, as $T$ and $k$ tend to infinity, CPS converges to $x^*$. More specifically, our proof in Sec A.2 of the Appendix implies that the way to decrease the terminal error is to have large values of $T$ with $\gamma(1) = \infty$. This coincides with our choice of $\gamma(t)$ for the final denoising steps, where $\gamma(t)$ tends to infinity.
>
> **In short, we only consider $\left(\gamma(t) = \frac{2k(T-t+1)}{\lambda\_\mathrm{min}(A^TA)}\right)$ as a linearly decreasing function of $t$ with an adjustable design parameter $k$ for ease of theoretical analysis.** $k$ is not a part of Algorithm 1.
>
> We have highlighted all the changes and additions in red in the manuscript.
>
> [1]  PSA-GAN: Progressive Self Attention GANs for Synthetic Time Series, Jeha et al.
> [2] Time Weaver: A Conditional Time Series Generation Model, Narasimhan et al.
> [3] Time-series Generative Adversarial Networks, Yoon et al.
> [4] CSDI: Conditional Score-based Diffusion Models for Probabilistic Time Series Imputation, Tashiro et al.

---

> > ### Comment · Reviewer_HGVu · 2024-11-21
> > **Response**
> >
> > I thank the authors for the quick response and clarification. I will raise my score to 8.

---

> > ### Comment · Reviewer_HGVu · 2024-11-22
> > **About the issue raised by Reviewer XXLw**
> >
> > I checked the paper [1] listed by Reviewer XXLw, finding that both methods employ a projection step for satisfying the constraint. However, the difference in your formulas is the term $\gamma(t)\prod(z)$. Can you give more details about this difference for addressing the concern of Novelty? It seems like your method is more like a soft version of projection and becomes a hard one when $\gamma(t)$ becomes infinity.
> >
> >
> >
> >
> > **References**
> >
> > [1] Constrained Synthesis with Projected Diffusion Models. NeurIPS 2024

---

> ### Author Response · Authors · 2024-11-22
> **Discussion with Reviewer HGVu**
>
> We thank the reviewer for revising the score. We are glad that your questions have been addressed. Below, we address your remaining question regarding how CPS is different from Projected Diffusion Models (PDM) [A].
>
> We note **3 crucial differences** between our CPS approach and PDM. (1) We first highlight the key difference between the **sampling procedures** in CPS and PDM.
>
> PDM Approach - PDM projects all the intermediate latents $z_T, \dots, z_1$ to the required constraint set along with $z_0$. Therefore, for every denoising step $t$, PDM operates by obtaining $z_{t-1}$ from $z_t$ and projects $z_{t-1}$ to the constraint set using a constrained optimization step.
>
> Our Approach (CPS) - CPS does not project the intermediate latents. Instead, the posterior mean estimate $\hat{z}_0$ (or the expectation of the clean sample given the noisy latent $z_t$, $E[z_0 | z_t]$) is projected to the constraint set. Therefore, for every denoising step $t$, CPS does the following operations:
>
> * Computes the posterior mean estimate $\hat{z}_0(z_t)$ from $z_t$.
> * Projects the posterior mean estimate $\hat{z}_0(z_t)$ to the constraint set using an unconstrained optimization step (line 5, Algorithm 1). The reviewer is correct; $\gamma(t) \Pi(z)$ is the penalty for constraint violation. This turns into a hard constraint when $\gamma(t)$ becomes infinity at the final denoising step.
> * Finally, the projected posterior mean estimate $\hat{z}\_{0, \mathrm{pr}}(z_t)$ is transformed to $z_{t-1}$ using a non-markovian forward noising process (Eq. 2, lines 165-167).
>
> Next, we discuss two major issues of PDM that degrade its sample quality and how our method circumvents these issues.
>
> **(2) The constraint set is defined for the clean samples and not the intermediate noisy latents of a denoising process.** As the goal is to generate constrained time series samples, it is sufficient if the generated sample $z_0$ belongs to the constraint set (i.e., terminal constraint). **However, PDM assumes the constraint set for clean samples to be the same for noisy intermediate latents.** By forcing the latents to satisfy the same terminal constraint, PDM eliminates most sample paths ($z_T, \dots, z_0$) that may eventually satisfy the terminal constraint. **This results in poor sample diversity.**
>
> **How does CPS handle this problem?** CPS eliminates this problem **by projecting the posterior mean estimate $\hat{z}_0$ and not the noisy intermediate latents.** Recall that $\hat{z}\_0$ is the expected clean sample which has a similar noise level as the constraint set. Furthermore, the projected posterior mean estimate $\hat{z}\_{0,\mathrm{pr}}(z_t)$ is transformed to $z_{t-1}$ using a non-markovian forward noising process. **This effectively allows for sample paths where the generated sample $\hat{z}_0$ alone satisfies the terminal constraint and the intermediate noisy latents can be flexible.**
>
> **(3) PDM projection step pushes $z_{t-1}$ off the noise manifold for $t-1$.** In PDM, the projection step, when applied directly to the noisy latent $z_{t-1}$ pushes it out of the noise manifold corresponding to the diffusion step $t-1$. Consequently, a pre-trained denoiser struggles to accurately denoise the projected $z_{t-1}$ as it would be out of the training domain of the denoiser.
>
> **How does CPS handle this problem?** This effect is significantly reduced in CPS because **CPS does not project $z_{t-1}$.** Instead, CPS projects the expected clean sample $\hat{z}\_0$. Consequently, the projected posterior mean estimate $\hat{z}\_{0, \mathrm{pr}}(z_t)$ is transformed into $z_{t-1}$ by using a non-markovian forward noising process (Eq. 2). This ensures that $z_{t-1}$ stays very close to the noise manifold corresponding to the diffusion step $t-1$. Therefore, a pre-trained denoiser can denoise $z_{t-1}$ more accurately in our approach. This ultimately preserves the generated sample quality.
>
> Finally, we have highlighted the main differences in the sampling process between the CPS and PDM, and how this difference affects the generated sample quality. Empirically, we show that **CPS performs much better than PDM on the real-world datasets**, providing **7x reduction in the Frechet Time Series Distance metric and 4x reduction in the Discriminative Score metric overall, as given in Table 3 (page 30).**
>
> We hope our response has clarified the differences between our algorithm and PDM [A]. We are happy to address any further questions during the discussion phase.
>
> ### References
>
> [A] Constrained Synthesis with Projected Diffusion Models, Christopher et al.

---

### Official Review · Reviewer_pCUD · 2024-10-31

**Soundness:** 4
**Presentation:** 4
**Contribution:** 4
**Rating:** 8
**Confidence:** 3

**Summary:**

The paper proposes a method called CPS to sample time series from a diffusion model with given constraints. The trick is projecting the intermediate values in the diffusion model sampling process towards the constraint set. In later iterations of the sampling process, the diffusion model corrects the projected time series so it stays close to the data generating distribution. The paper derives some theoretical properties of their sampler, including a convergence theorem in the case of Gaussian data. The paper also contains experiments on several real and toy datasets, where CPS outperforms baselines.

**Strengths:**

The paper is very well-written, easy to read, and makes clear arguments. The proposed method CPS is simple, and seems easy to apply. It may even be applicable to other domains, like tabular data, where meaningful constraints can be specified as simple inequalities. The results are very good, with CPS outperforming baseline methods in almost all experiments.

**Weaknesses:**

I'm not sure how meaningful the similarity metrics comparing the generated sample and the test sample that satisfies the all constraints is, since an optimal generator may not get the best possible score. This is because any finite set of constraints is likely to leave some freedom to generate different outputs, which would not be identical to the reference sample from the test set. This is shown in Table 1, where none of COP, COP-FT and CPS violate any constraints on several datasets, but have different values for the DTW and SSIM metrics.

A comparison with Loss DiffTime would still be interesting, since it would show how beneficial training specifically for a given set of constraints is.

**Questions:**

- In Section 3, should $\hat{z}_0$, and various related symbols, have the subscript $t$? If not, why is 0 used there?
- What does the form of the distribution that CPS samples from, given in Theorem 1, tell us?
- How many constraints are imposed in Table 1 and Figure 6?
- How badly are the generated time series in Figure 6 violating the constraints?
- How is it possible that CPS is effectively copying samples (see traffic conditional and stocks datasets in Figure 6) from the test set that the generator has presumably not seen during training? Are these time series really characterised so precisely by a relatively small number of numbers?

---

> ### Author Response · Authors · 2024-11-21
> **Rebuttal by Authors**
>
> We thank the reviewer for taking the time to provide a thoughtful and thorough review of the paper. The reviewer acknowledges the simplicity of the proposed approach and recognizes the effectiveness of CPS with respect to generating high-quality samples that adhere to hard constraints.
>
> # Addressing the weaknesses pointed out by the reviewer
>
> ## On the validity of the similarity metrics. (Weakness 1)
>
> **From our experiments, we note that higher similarity scores between the synthesized and the real test samples indicate better sample quality**. From Table 1, observe that the highest SSIM and the lowest DTW scores (best similarity metrics) correspond to the lowest TSTR and the lowest Frechet Distance metrics (best sample quality metrics). We agree with the reviewer’s point that there could be multiple generated samples that belong to the constraint set. However, we specifically design the constraint set with a large number of constraints such that only one real test sample exists per constraint set (check lines 427-429). Therefore, any generated sample with high sample quality is generally expected to be similar to the real test sample. As a result, we report the similarity metrics.
>
> Therefore, **even though COP variants have perfect constraint satisfaction, their sample quality metrics are poor, and this is also reflected in the poor similarity metrics**.
>
> ## On the comparison with Loss DiffTime. (Weakness 2)
>
> **Loss DiffTime provides better sample quality but fails to adhere to the constraint set due to the absence of principled projection steps**. We agree with the reviewer that comparing against Loss DiffTime will showcase the relative performance gap for training-free approaches like CPS. To this end, we trained the Time Weaver-CSDI models with constraints as conditional input. The results are provided in Table 5 (Sec. G, page 30) in the Appendix. Overall, Loss DiffTime fails to generate samples that belong to the required constraint set because there is no explicit way to force the generated sample to adhere to the required constraints. **Constraint-specific training only aids in generating high-quality samples**.

---

> > ### Author Response · Authors · 2024-11-21
> > **Rebuttal by Authors**
> >
> > # Response to the questions from the reviewer
> >
> > ## On the issue of the subscript $0$ for $\hat{z}_0$. (Question 1)
> > **We use the subscript $0$ because $\hat{z}_0$ represents the expectation of the clean sample**. $\hat{z}_0(z_t\; \epsilon\_\theta)$ indicates the posterior mean estimate as a function of the noisy latent $z_t$ and the noise estimate in $z_t$, $\epsilon\_\theta(z_t,t)$. Here, $\epsilon\_\theta$ represents the denoiser network. The posterior mean estimate is the expectation of the clean sample conditioned on the noisy latent $z_t$. Since it is the expectation of the clean sample, we use the subscript $0$.
> >
> > ## On the clarification regarding Theorem 1. (Question 2)
> > In the absence of constraints, during each denoising step, $z_{t-1}$ is obtained using Eq. 2 (lines 166 and 167) from $z_t$ and $\hat{z}\_0$. This process can be seen as sampling $z_{t-1}$ from the Gaussian distribution given in Eq. 3 (lines 172 - 174).
> >
> > However, in the presence of constraints, we transform $\hat{z}\_0$ to the projected posterior mean estimate $\hat{z}\_{0,\mathrm{pr}}$, and $z_{t-1}$ is obtained from $z_t$ and $\hat{z}\_{0,\mathrm{pr}}$ using Eq. 2. This indicates that during each denoising step, we sample $z_{t-1}$ from the Gaussian distribution given in Eq. 3 with $\hat{z}\_0$ replaced by  $\hat{z}\_{0,\mathrm{pr}}$. Here, $\hat{z}\_{0,\mathrm{pr}}$ is obtained from  $\hat{z}\_0$ through the projection step (line 5 of Algorithm 1).
> >
> > Further, this explanation can be observed from Eqs. 3 (lines 172 - 174) and 5 (lines 289 and 290), where the only difference between the two is that  $\hat{z}\_0$ is replaced by  $\hat{z}\_{0,\mathrm{pr}}$, precisely because of the projection step in line 5 of Algorithm 1.
> >
> > Note that the projection step is deterministic, and therefore, **each iteration in CPS can be viewed as sampling from a Dirac delta distribution centered at $\hat{z}\_{0,\mathrm{pr}}$, and subsequently sampling from the Gaussian distribution in Eq. 5**. This is exactly the outline of the proof of Theorem 1, which is explained in Sec. A.1 of the Appendix.

---

> > > ### Author Response · Authors · 2024-11-21
> > > **Rebuttal by Authors**
> > >
> > > ## On the clarification regarding Table 1 and Figure 6 (Questions 3 and 4)
> > >
> > > For all the datasets, we synthesize 96 timestamps. We use the following set of constraints for each channel - mean, mean consecutive change (mcc), argmax, argmin, value at argmax, value at argmin, values at the 5 timestamps 1, 24, 48, 72, and 96.
> > >
> > > **For the air quality dataset (6 channels), there are 66 constraints** (6 mean + 6 mcc + 6 argmax + 6 argmin + 6 value at argmax + 6 value at argmin + 5x6 value at constraints). The same extends to the conditional variants as well.
> > >
> > > **For the traffic dataset (1 channel), there are 11 constraints** (similar to the air quality dataset). The same extends to the conditional variants as well.
> > >
> > > **For the stocks dataset (6 channels), there are 450 constraints** (66 (similar to air quality) + 4x96 OHLC constraints).
> > >
> > > **For the waveforms dataset (1 channel), there are 106 constraints** (11 (similar to traffic) + 95 trend constraints between successive time stamps).
> > >
> > > **In Fig. 6, both COP variants and CPS satisfy all the constraints. However, Guided-DiffTime fails to generate samples that belong to the constraint set**. This is shown by very high values of constraint violation rate in Table 1.
> > >
> > > To visualize constraint satisfaction through Fig. 1 is difficult as we cannot show that a sample satisfies/violates a particular mean constraint. However, we request the reviewer to check Fig. 7 in the Appendix (page 32), where the fixed value constraint satisfaction is clearly observed.

---

> > > > ### Author Response · Authors · 2024-11-21
> > > > **Rebuttal by Authors**
> > > >
> > > > ## Clarification regarding the qualitative results (Question 5)
> > > >
> > > > **The high degree of similarity is attributed to the metadata or conditional input for the traffic conditional setup. Similarly, for the stocks dataset, the high degree of similarity is attributed to the large number of constraints**. The reviewer is correct; these are test samples that were never seen during the training phase.
> > > >
> > > > The high degree of similarity between the generated and the real samples in the traffic conditional dataset is because of the additional metadata or conditional input to the diffusion model. In Fig. 7 (Appendix, Page 32), we show the variance in the generated samples. In the absence of metadata, note that CPS generates constraint-specific samples with high variance. But, all generated samples track the real test sample. When both constraints and metadata (conditional input) are provided, the variance reduces significantly.
> > > >
> > > > For the stocks dataset, we impose mean, argmax, and many other constraints along with the OHLC constraint (450 constraints in total), which accurately drives the sample generation process toward the real sample from the constraint set. From Fig. 7, note that there is variance in the generated samples, but it is low, and all the generated samples accurately track the real test sample.
> > > >
> > > > We have highlighted all the changes and additions in red in the manuscript.

---

> ### Comment · Reviewer_pCUD · 2024-11-21
>
> Thank you for the response. You have addressed most of my questions and concerns well. I still have a comment on the similarity metrics, though I don't consider it a major issue.
>
> > However, we specifically design the constraint set with a large number of constraints such that only one real test sample exists per constraint set (check lines 427-429). Therefore, any generated sample with high sample quality is generally expected to be similar to the real test sample.
>
> While there may only be one real sample satisfying the constraints, there may be other possible samples in the real data distribution that also satisfy the constraints, and have non-negligible probability to appear in a dataset of the size of the synthetic dataset. Because of this, it is possible that the perfect synthetic data generator that exactly samples from the distribution of real samples satisfying the constraint would not get the best possible score.
>
> Regardless, I think this is a fairly minor issue, since your method works very well according to the other metrics which do not have this issue.

---

> > ### Author Response · Authors · 2024-11-23
> > **Discussion with Reviewer pCUD**
> >
> > We thank the reviewer for the positive feedback and are glad our response addressed most concerns. Regarding the similarity metrics (DTW and SSIM), we agree there could be cases where high-likelihood samples satisfy hard constraints but exhibit low similarity with the real test sample from the constraint set. However, the constraints we impose, such as argmax/argmin locations, values at these points, and values at fixed timestamps with equal intervals (e.g., 0, 24, 48, 72, 96), effectively capture overall trends and seasonality in the time series. When a large number of constraints (~100) are imposed, the feasible set is significantly reduced, making it reasonable to expect high-likelihood samples to share trends and seasonality with the real sample.
> >
> > Additionally, we have included the discriminative score (DS), a quality metric defined as |0.5 - classification accuracy|, using a 2-layer RNN to distinguish real and generated samples. Lower DS values indicate higher fidelity, and CPS outperforms baselines on DS, as shown in Table 2 (Sec. E, page 28). Poor similarity scores (e.g., high DTW and low SSIM) generally correlate with poor sample quality metrics (e.g., high FTSD, DS, and TSTR), supporting their practical usage.
> >
> > ### References
> >
> > [1] Time-series Generative Adversarial Networks, Yoon et al.

---

### Official Review · Reviewer_qvnv · 2024-11-01

**Soundness:** 2
**Presentation:** 4
**Contribution:** 2
**Rating:** 6
**Confidence:** 4

**Summary:**

This paper investigates generating time series data samples with specific constraints. The authors propose a training-free constrained posterior sampling method based on $z-\hat{z}_{0,pr}$ optimization problem within objective function. Furthermore, they theoretically justify the use of imposing condition as a conditional sampling. They empirically prove their idea across various settings, including different setups and datasets.

**Strengths:**

• The paper investigate the very important and practical issues in time series generation, which encompasses generating particular examples with specific condition. This is the unique properties in time series generation compared to vision tasks.

• The authors propose a convex optimization based methods which is much more effective and simple for time series than existing methods.

• The paper conducts experiments on various datasets, including well-known ones, demonstrating the effectiveness of the constrained synthesis.

• The authors establish a theoretical background of their methods.

**Weaknesses:**

Please refer to the Questions section.

**Questions:**

•	Even though they propose new method, but the reviewer think the method is ot novel. It poses regularize similar to (Coletta et al., 2024), but with convex optimization strategy.

•	The author mentioned that the guided gradient is not enough for restricting the constraints. Could you clarify the main differences of two methods? In terms of convex optimization with global optimum, two of them should be same with proper set of hyperparameters.

•	Furthermore, the authors mentioned about the constraints sets as convex. However, the objective $||z-\hat{z}_{0,pr}||^2$ that the author proposed might not be optimum for non-convex optimization.

•  The comparison includes only (Coletta et al., 2024). Could you consider additional comparison methods, such as the instance-aware guidance strategy in [1]?

•	In Theorem 2, why does $k$ approach infinity as $T$ goes to infinity? More explanation on the convergence analysis would be helpful.

•	Please correct minor errors. Such as generation(Yoon et al.) in line 140 and Recently (Coletta ~) -> use citet.

[1] DIFFUSION-TS: INTERPRETABLE DIFFUSION FOR GENERAL TIME SERIES GENERATION, ICLR 2024.

---

> ### Author Response · Authors · 2024-11-21
> **Rebuttal by Authors**
>
> We thank the reviewer for taking the time to provide a thoughtful and thorough review of the paper. The reviewer acknowledges the practical importance of addressing the constrained time series generation problem. Additionally, the reviewer recognizes that the effectiveness of CPS is shown on multiple real-world datasets.
>
> # Response to the questions from the reviewer
>
> ## On the novel contributions with respect to Coletta et al. ([1]). (Question 1)
>
> **Our main novelty, which leads to significant improvements in both sample quality and constraint satisfaction metrics, is the iterated projection and denoising process.** Conceptually, the closest approach to CPS is Guided-DiffTime. The main difference between the two approaches is how the noisy latent for the step $t-1$, $z_{t-1}$, is obtained from the noisy latent for the step $t$, $z_t$.
>
> **Guided-DiffTime approach** - Guided-DiffTime obtains the noise estimate in $z_t$ and modifies the noise estimate using a gradient update to minimize constraint violation. This is indicated by $\hat{\epsilon} \leftarrow \hat{\epsilon} - \rho\nabla_{z_t}\Pi(\hat{z}_0(z_t))$, where $\Pi(\hat{z}_0(z_t))$ is the constraint violation. **This requires a backpropagation step through the denoiser**. As explained in [1], $\rho$ is chosen through trial and error.
>
> **Our approach** - However, in CPS, we first compute the posterior mean estimate from $z_t$ (line 4, Algorithm 1), project it to the constraint set  (line 5, Algorithm 1), and transform the projected posterior mean estimate to $z_{t-1}$  (lines 7-9, Algorithm 1). **In CPS, there is no backpropagation through the denoiser**.
>
> This key modification provides the following advantages over Guided-DiffTime:
> 1. **CPS provides constraint satisfaction guarantees for convex constraints, whereas Guided-DiffTime cannot**.  In CPS, the projection operation after each denoising step (including the final step) provides constraint satisfaction guarantees for convex constraints for an appropriate step size and number of update steps in the projection operation.
> 2. As a consequence of the projection operation, **CPS can handle a large number of constraints much more effectively than Guided-DiffTime**. Guidance from different constraint violations needs to be treated accurately, resulting in a tedious hyperparameter tuning process (check lines 495-497). However, the projection step in CPS circumvents this problem, and the subsequent denoising steps remove the unnatural artifacts caused by the projection step, resulting in high sample quality.
>
> Coletta et al. also propose Constrained Optimization Problem (COP). Below, we list the conceptual and empirical advantages of CPS over COP.
> 1. **CPS does not require any external module**. Whereas, COP uses a discriminator network to enforce realism during the projection process. This requires additional training (apart from the diffusion model) and leads to complex sampling procedures.
> 2. **COP is very sensitive to the initial seed of the optimization process**. The sample quality is poor if the initial seed is far away from the constraint set. In this case, the projection operation destroys the sample quality to achieve constraint satisfaction (check lines 476-482).
> **Overall, CPS achieves much better sample quality metrics than Guided-DiffTime and COP**.

---

> > ### Author Response · Authors · 2024-11-21
> > **Rebuttal by Authors**
> >
> > ## On the insufficiency of the guidance gradient for constraint satisfaction. (Question 2)
> >
> > **In guidance-based approaches, the guidance gradient steps after each denoising step alone cannot guarantee constraint satisfaction, even for convex constraints**. Guided-DiffTime [1] modifies the noise estimate with a gradient update for every denoising step such that the denoising process is steered towards the constraint set ($\hat{\epsilon} \leftarrow \hat{\epsilon} - \rho\nabla_{x_t}\Pi(\hat{z}_0(z_t))$). Similarly, Diffusion-TS [2] employs a guidance-based approach with multiple guidance steps after each denoising step. These modifications can push the sample toward the constraint set, but not guarantee constraint satisfaction, even for convex constraint set. We have empirically shown this in Tables 1 and 4 (page 30) of our manuscript. Similar results are shown in [1] as well. **Fundamentally, these approaches lack a principled projection step that is available in both COP and CPS**.
> >
> > ## On dealing with non-convex constraints. (Question 3)
> >
> > We agree with the reviewer that the objective function in the projection step (line 5,  Algorithm 1) is non-convex for arbitrary constraint classes. For non-convex constraints, the projection step may not result in the global optimal solution. Additionally, it is not possible to guarantee constraint satisfaction for non-convex constraints. However, **our iterated projection and denoising process pushes the sample very close to the constraint set while nullifying the adverse effects of the projection step**.
> >
> > We request the reviewer to check Sec. H of the Appendix, where we have added new experiments with a general class of constraints. Specifically, we imposed the Autocorrelation Function (ACF) value at a particular lag as an equality constraint for the stocks dataset. Our experimental results in Table 6 (page 31) show the effectiveness of iterated projection and denoising steps. **CPS achieves the lowest constraint violation rate and constraint violation magnitude values while maintaining very high sample quality**.

---

> ### Author Response · Authors · 2024-11-21
> **Rebuttal by Authors**
>
> ## On the comparison with Diffusion-TS. (Question 4)
>
> **We modified the Diffusion-TS baseline for constrained time series generation. However, since it is a guidance-based approach, it fails to generate samples that adhere to hard constraints**. Although the guidance approach provided in Diffusion-TS is not for constrained sample generation, we agree with the reviewer that this approach can be modified for our problem setting. We provide the comparison against Diffusion-TS in Sec. G of the Appendix (check Table 4, page 30).
>
> However, similar to the Guided-DiffTime approach, **there is no principled projection step for constraint satisfaction**. This is shown through the high values for the constraint violation metrics.
>
> We provide an additional comparison against a very recent constrained sample generation approach, Projected Diffusion Models (PDM) [3] (check Table 3 in Sec. G of the Appendix, page 30). **Overall, we observe that CPS outperforms all the training-free methods in generating high-quality samples that adhere to the required constraints**.
>
> ## Clarification regarding the convergence analysis. (Question 5)
>
> **$k$ is independent of $T$**. In our convergence analysis, $k$ is a design parameter that affects the penalty coefficient $\gamma(t)$ in a linear fashion $\left(\gamma(t) = \frac{2k(T-t+1)}{\lambda_{\mathrm{min}}(A^TA)}\right)$, and $T$ is the total number of denoising steps. **$T$ and $k$ are independently chosen and the choice of $T$ does not affect the value of $k$**. Theorem 2 claims that, under Assumption 2, CPS arrives at the unique solution $x^*$ when $T$ and $k$ tend to infinity. For any finite $k$, the terminal error in the generated sample is upper bounded, as shown in Theorem 2.
>
> For both finite $T$ and finite $k$, the upper bound on the terminal error is shown in Eq. 26 (lines 951-956) in Sec A.2 of the Appendix.
>
> We have corrected the citation errors and updated the manuscript accordingly. We have highlighted all the changes and additions in red in the manuscript.
>
> [1] On the Constrained Time-Series Generation Problem, Coletta et al.
> [2] Diffusion-TS: Interpretable Diffusion for General Time Series Generation, Yuan et al.
> [3] Constrained Synthesis with Projected Diffusion Models, Christopher et al.

---

> > ### Comment · Reviewer_qvnv · 2024-11-23
> > **Response**
> >
> > I have carefully reviewed all the authors' rebuttals.
> >
> > I understand their points and have thoroughly read the rebuttal sections in the paper.
> >
> > The proposed method demonstrates superior performance across various metrics and scenarios. In my case, conducting new experiments on non-convex optimization helped me better understand the effectiveness of CPS.
> >
> > Given that the method is specifically designed for time series generation under specific constraints, the projection approach has the potential to be effectively utilized in advanced diffusion-based methodologies.
> >
> > As a result, I have decided to increase my evaluation score.

---

> > > ### Author Response · Authors · 2024-11-28
> > > **Thank you for increasing the score**
> > >
> > > We are glad that our rebuttal has addressed the reviewer's concerns. We sincerely appreciate the thoughtful review, which has improved the quality of our manuscript, and the decision to increase the score.

---

### Official Review · Reviewer_XXLw · 2024-11-04

**Soundness:** 3
**Presentation:** 4
**Contribution:** 2
**Rating:** 5
**Confidence:** 5

**Summary:**

The paper introduces Constrained Posterior Sampling, a diffusion-based method for generating time series that adhere to hard constraints. The approach is theoretically grounded and the empirical evaluations demonstrate CPS's effectiveness in producing high-quality, constraint-compliant time series.
However, the novelty of CPS is somewhat limited due to its conceptual similarities with existing methods (see below for extended discussion) .

**Strengths:**

- The paper provides a solid theoretical basis for CPS, including proofs of convergence and constraint satisfaction.
- Experiments on real-world datasets demonstrate CPS's ability to generate high-quality and constraint-compliant time series.
- It seems like that the framework can handle a large number of constraints, and all at sampling time.

**Weaknesses:**

- The key weakness of this approach is novelty. CPS shares significant conceptual similarities with existing methods.
In particular :
[A] also uses a projection step within diffusion processes to enforce constraints, and consider arbitrary constraint sets, while providing theoretical guarantees for convex constraint sets. The algorithm proposed in [A] is also very similar to the proposed one.
[B] implements hard constraints on the outputs of autoencoders for linear constraints, as those considered in this paper.
[C] uses “mirror mappings” which apply to convex constraint sets.
[D] further extend the classes of constraints can be handled handling constraints represented by convex polytopes.

- Given the above, the paper lacks a thorough comparison with existing methods and this omission makes it difficult to assess CPS's relative advantages or limitations.

- The projection step in CPS can be computationally intensive, especially for high-dimensional constraints, how can this be handled?

[A] Jacob K Christopher, Stephen Baek, Ferdinando Fioretto. Constrained Synthesis with Projected Diffusion Models. NeurIPS 2024 - ArXiv: https://arxiv.org/abs/2402.03559v1 (first appeared on ArXiv on Feb 2024)

[B] Thomas Frerix, Matthias Nießner, and Daniel Cremers. Homogeneous linear inequality constraints for neural network activations. In Proceedings of the IEEE/CVF Conference on Computer Vision and Pattern Recognition Workshops, pages 748–749, 2020.

[C] Guan-Horng Liu, Tianrong Chen, Evangelos Theodorou, and Molei Tao. Mirror diffusion models for constrained and watermarked generation. NeurIPS 2023. ((first appeared on ArXiv on Oct 2023)

[D] Nic Fishman, Leo Klarner, Emile Mathieu, Michael Hutchinson, and Valentin De Bortoli. Metropolis sampling for constrained diffusion models. NeurIPS 2023. ((first appeared on ArXiv on July 2023)

**Questions:**

see limitations above

---

> ### Author Response · Authors · 2024-11-21
> **Rebuttal by Authors**
>
> We thank the reviewer for taking the time to provide a thoughtful and thorough review of the paper. The reviewer acknowledges our approach’s ability to scale to a large number of constraints without any constraint-specific training.
>
> # Addressing the weaknesses indicated by the reviewer:
>
> ## On the comparison of CPS with constrained diffusion models from other domains. (Weaknesses 1 and 2)
>
> **CPS is scalable to new constraints without any training.** However, the prior works ([B],[C], and [D]) are not. [B], [C], and [D] focus on learning data distributions, where the domain of the samples is constrained to some manifold. These works introduce constraint-specific training modifications.
>
> Additionally, **CPS tackles a different problem**, where the sample domain is not restricted to any manifold, but the goal is to generate samples that adhere to arbitrary constraints. Any constraint-specific training solution hinders the ability to adapt to arbitrary constraints.
>
> We elaborate on this point further by explaining our evaluation process, which involves generating N samples where each sample adheres to a different constraint set. For example, considering the class of constraints to include mean and argmax, our evaluation process would be the following:
> 1. Generate sample 1 with the following constraints: mean = 4.5, argmax = 10,
> 2. Generate sample 2 with the following constraints: mean = 1.2, argmax = 30, and so on.
>
> Adopting the approaches in [B], [C], and [D] would involve training a model for each of the N samples.
>
> ## On the comparison with Projected Diffusion Models [A].
>
> [A] addresses the same problem as ours. **The major difference between [A] and our approach is the projection step**. In [A], the noisy latents ($z_T, \dots, z_1$) are projected using a constrained optimization approach, whereas CPS projects the posterior mean estimate ($\hat{z}_0$, check line 4, Algorithm 1) using an unconstrained optimization approach.
>
> We compared our approach (CPS) against [A], and **CPS outperforms [A] on sample quality metrics for all the real-world datasets used in our experiments**. We request the reviewer to check Table 3 in Sec. G of the Appendix for the results.
>
> We hypothesize that the projected version of the noisy latent $z_{t-1}$ has a very low likelihood when compared to the training samples that the pre-trained diffusion model has seen, and therefore, the overall denoising process results in samples with poor fidelity.
>
> On the contrary, during the sample generation process, CPS projects the posterior mean estimate (line 5, Algorithm 1) and subsequently transforms the projected posterior mean estimate to obtain $z_{t-1}$, using a non-markovian forward diffusion process (Eq. 2).  Our key intuition is that the projection step suitably modifies $z_{t-1}$ to steer the denoising process towards the constraint set and the forward process nullifies the adversarial effects of the projection step.
>
> **We have added a new section to the Appendix - Sec. F - Extended Related Works, where we explained the difference between CPS and prior works [B], [C], and [D].**
>
> ## Extension to a general class of constraints.
>
> Additionally, similar to [A], we show that **our approach is not restricted to convex constraints** and works for a general class of constraints. Specifically, we experimented with the stocks dataset to generate samples with a constraint on the Autocorrelation Function (ACF) value at a specific lag (check Sec. H of the Appendix). Even for such a general class of constraints, our approach outperforms the baselines in terms of sample quality and constraint violation metrics (check Table 6).
>
> Further, generating time series samples with such arbitrary constraints has significant practical applications in terms of stress-testing and synthesizing samples for novel what-if scenarios. The prior state-of-the-art work in this domain, [E], only considers simple bounds, argmax, and fixed value constraints. The evaluation was performed on a single dataset. More importantly, [E] never considers the combination of multiple constraint classes, a key area where all the baselines fail to perform well. In our work, we have significantly improved the comparison in terms of the combination of multiple constraint classes, as well as datasets (6 datasets).
>
> [A] Constrained Synthesis with Projected Diffusion Models, Christopher et al.
> [B] Homogeneous linear inequality constraints for neural network activations, Frerix et al.
> [C] Mirror diffusion models for constrained and watermarked generation, Liu et al.
> [D] Metropolis sampling for constrained diffusion models, Fishman et al.
> [E] On the Constrained Time-Series Generation Problem, Coletta et al.

---

> ### Author Response · Authors · 2024-11-21
> **Rebuttal by Authors**
>
> ## On handling high-dimensional constraints. (Weakness 3)
>
> CPS is equivalent to [A] in terms of computation, as both approaches require a projection step after every denoising step. We evaluated our approach for time series samples up to **576 dimensions** (e.g., the air quality and the stocks dataset). We have provided the inference time taken to generate samples with up to **66 and 450 constraints** for the air quality and the stocks datasets, respectively (check Table 8 in Sec. J of the Appendix). In comparison to the baseline approaches, CPS is only slightly more in terms of inference time while providing high sample quality and perfect constraint satisfaction.
>
> Furthermore, we note that there are multiple ways to reduce the inference time, such as:
> 1. Reducing the number of update steps in each projection operation (line 5 of Algorithm 1) during the initial denoising steps where the signal-to-noise ratio is very low.
> 2. The projection operation (line 5 of Algorithm 1) need not be performed for every denoising step. Consequently, we can develop principled methods to identify the denoising steps where projection is required based on constraint violation.
>
> We have highlighted all the changes and additions in red in the manuscript.
>
> [A] Constrained Synthesis with Projected Diffusion Models, Christopher et al.

---

> > ### Comment · Reviewer_XXLw · 2024-11-21
> >
> > Many thanks for your response. I still see this work as having limited contribution when placed in relation to [A]. As the authors themselves mentioned,
> > > [A] addresses the same problem as ours.
> > and
> > > CPS is equivalent to [A] in terms of computation, as both approaches require a projection step after every denoising step
> >
> > I believe the key novelty and significant of this work still needs to be clarified.

---

> > > ### Author Response · Authors · 2024-11-22
> > > **Discussion with Reviewer XXLw**
> > >
> > > We thank the reviewer for the quick response. Below, we address your remaining questions regarding how CPS is different from Projected Diffusion Models (PDM) [A].
> > >
> > > We note **3 crucial differences** between our CPS approach and PDM. (1) We first highlight the key difference between the **sampling procedures** in CPS and PDM.
> > >
> > > PDM Approach - PDM projects all the intermediate latents $z_T, \dots, z_1$ to the required constraint set along with $z_0$. Therefore, for every denoising step $t$, PDM operates by obtaining $z_{t-1}$ from $z_t$ and projects $z_{t-1}$ to the constraint set.
> > >
> > > Our Approach (CPS) - CPS does not project the intermediate latents. Instead, the posterior mean estimate $\hat{z}_0$ (or the expectation of the clean sample given the noisy latent $z_t$, $E[z_0 | z_t]$) is projected to the constraint set. Therefore, for every denoising step $t$, CPS does the following operations:
> > >
> > > * Computes the posterior mean estimate $\hat{z}_0(z_t)$ from $z_t$.
> > > * Projects the posterior mean estimate $\hat{z}_0(z_t)$ to the constraint set.
> > > * Finally, the projected posterior mean estimate $\hat{z}\_{0, \mathrm{pr}}(z_t)$ is transformed to $z_{t-1}$ using a non-markovian forward noising process (Eq. 2, lines 165-167).
> > >
> > > Next, we discuss two major issues of PDM that degrade its sample quality and how our method circumvents these issues.
> > >
> > > **(2) The constraint set is defined for the clean samples and not the intermediate noisy latents of a denoising process.** As the goal is to generate constrained time series samples, it is sufficient if the generated sample $z_0$ belongs to the constraint set (i.e., terminal constraint). **However, PDM assumes the constraint set for clean samples to be the same for noisy intermediate latents.** By forcing the latents to satisfy the same terminal constraint, PDM eliminates most sample paths ($z_T, \dots, z_0$) that may eventually satisfy the terminal constraint. **This results in poor sample diversity.**
> > >
> > > **How does CPS handle this problem?** CPS eliminates this problem **by projecting the posterior mean estimate $\hat{z}_0$ and not the noisy intermediate latents.** Recall that $\hat{z}\_0$ is the expected clean sample which has a similar noise level as the constraint set. Furthermore, the projected posterior mean estimate $\hat{z}\_{0,\mathrm{pr}}(z_t)$ is transformed to $z_{t-1}$ using a non-markovian forward noising process. **This effectively allows for sample paths where the generated sample $\hat{z}_0$ alone satisfies the terminal constraint and the intermediate noisy latents can be flexible.**
> > >
> > > **(3) PDM projection step pushes $z_{t-1}$ off the noise manifold for $t-1$.** In PDM, the projection step, when applied directly to the noisy latent $z_{t-1}$ pushes it out of the noise manifold corresponding to the diffusion step $t-1$. Consequently, a pre-trained denoiser struggles to accurately denoise the projected $z_{t-1}$ as it would be out of the training domain of the denoiser.
> > >
> > > **How does CPS handle this problem?** This effect is significantly reduced in CPS because **CPS does not project $z_{t-1}$.** Instead, CPS projects the expected clean sample $\hat{z}\_0$. Consequently, the projected posterior mean estimate $\hat{z}\_{0, \mathrm{pr}}(z_t)$ is transformed into $z_{t-1}$ by using a non-markovian forward noising process (Eq. 2). This ensures that $z_{t-1}$ stays very close to the noise manifold corresponding to the diffusion step $t-1$. Therefore, a pre-trained denoiser can denoise $z_{t-1}$ more accurately in our approach. This ultimately preserves the generated sample quality.
> > >
> > > Finally, we have highlighted the main differences in the sampling process between the CPS and PDM, and how this difference affects the generated sample quality. Empirically, we show that **CPS performs much better than PDM on the real-world datasets**, providing **7x reduction in the Frechet Time Series Distance metric and 4x reduction in the Discriminative Score metric overall, as given in Table 3 (page 30).**
> > >
> > > We hope our response has clarified your concerns regarding the differences between our algorithm and PDM [A]. We are happy to address any further questions during the discussion phase. If our clarifications and the newly added results meet your expectations, we kindly request you to consider revising your score.
> > >
> > > ### References
> > >
> > > [A] Constrained Synthesis with Projected Diffusion Models, Christopher et al.

---

> > > > ### Comment · Reviewer_XXLw · 2024-11-23
> > > >
> > > > Thank you for the extended explanation. After more consideration, I have decided to increase my score. I still believe that novelty is a bit limited in the face of previous work in this area, but there is indeed a significant contribution in the proposed approach.
> > > > I encourage the authors to position their work fairly in the current literature, and include the explanations provided above to differentiate against these previous work.

---

> > > > > ### Author Response · Authors · 2024-11-28
> > > > > **Discussion with Reviewer XXLw**
> > > > >
> > > > > We thank the reviewer for the positive feedback and for increasing the score. As the reviewer pointed out, our approach is significantly different from PDM [1] and is more effective in terms of generating high-quality samples.
> > > > >
> > > > > As suggested by the reviewer, we made the following modifications to **position our work fairly in the current literature and highlight the differences with PDM [1]**:
> > > > >
> > > > > * We have now added the prior constrained diffusion approaches in Section 2.1 (lines 179-187) of the revised draft.
> > > > >
> > > > > * Appendix F.2 (lines 1579-1594) includes a detailed discussion on training-based methods for constraint satisfaction and their limitations in scalability.
> > > > >
> > > > > * In Appendix G (lines 1619-1644), we have explained the key differences between a leading method PDM [1] and our approach CPS, highlighting the importance of CPS in superior sample quality and diversity.
> > > > >
> > > > > * For clarity, we have now added references to Appendix G in the main paper (lines 373-376 and lines 516-518).
> > > > >
> > > > > We hope the above discussion and additional results in the revised draft successfully address the reviewer’s concerns. We would be glad to address any additional points in the remaining time of the discussion phase.
> > > > >
> > > > > ### References
> > > > >
> > > > > [1]  Constrained Synthesis with Projected Diffusion Models, Christopher et al.

---

### Meta-Review · Area_Chair_PYLY · 2024-12-19

**Metareview:**

This paper considers the constrained sampling problem from a diffusion model. It has two orthogonal components, the development of the constrained posterior sampling (CPS) algorithm and its application to time series. The proposed CPS algorithm leverages proximal optimization algorithms with a carefully designed varying objective to gradually impose the hard constraints. One major weakness pointed out by the reviewers is the lack of thorough comparisons to existing methods. The authors have added comparison experiments to several methods including PDM, Diffusion-TS and Loss-DiffTime. Still, some baselines are missing. Constrained sampling can be viewed as a special case of conditional sampling and inverse problem [1]. For instance, Stochastic Control Guidance (SCG) [2] can also be used to achieve constrained sampling. In addition to experimental comparison, the paper can also benefit from a detailed comparison in methodology level to highlight its contribution. Finally, the position of the paper can be improved. It seems CPS is not restricted to time series generation. If it has superior performance in general, experiments should be conducted in other domains as well to make the paper stronger. If it only works for time series, detailed discussion should be included to explain the reasons. Overall, this is a borderline paper. The authors are encouraged to prepare a comprehensive revision based on these comments.
[1] A Survey on Diffusion Models for Inverse Problems
[2] Symbolic Music Generation with Non-Differentiable Rule Guided Diffusion

**Additional Comments On Reviewer Discussion:**

The reviewers raises some questions on the results and the presentation of the paper. The authors reply by modifying the paper, adding experiments in the paper, and adding clarifications in the response. Some reviewers are not fully convinced and believe the paper is not ready for publication at ICLR.

---

### Decision · Program_Chairs · 2025-01-22

Reject